# Lower Bounds of Uniform Stability in Gradient-Based Bilevel Algorithms for Hyperparameter Optimization

**Rongzhen Wang**[1,2], **Chenyu Zheng**[1,2], **Guoqiang Wu**[3],
**Xu Min**[4], **Xiaolu Zhang**[4], **Jun Zhou**[4], **Chongxuan Li**[1,2]*
[1] Gaoling School of Artificial Intelligence, Renmin University of China
[2] Beijing Key Laboratory of Big Data Management and Analysis Methods
[3] School of Software, Shandong University [4] Ant Group
{wangrz,cyzheng,chongxuanli}@ruc.edu.cn; guoqiangwu@sdu.edu.cn;
minxu.mx@antgroup.com; {yueyin.zxl,jun.zhoujun}@antfin.com

## Abstract

Gradient-based bilevel programming leverages unrolling differentiation (UD) or implicit function theorem (IFT) to solve hyperparameter optimization (HO) problems, and is proven effective and scalable in practice. To understand the generalization behavior, existing works establish upper bounds on the uniform stability of these algorithms, while their tightness is still unclear. To this end, this paper attempts to establish stability lower bounds for UD-based and IFT-based algorithms. A central technical challenge arises from the dependency of each outer-level update on the concurrent stage of inner optimization in bilevel programming. To address this problem, we introduce lower-bounded expansion properties to characterize the instability in update rules which can serve as general tools for lower-bound analysis. These properties guarantee the hyperparameter divergence at the outer level and the Lipschitz constant of inner output at the inner level in the context of HO. Guided by these insights, we construct a quadratic example that yields tight lower bounds for the UD-based algorithm and meaningful bounds for a representative IFT-based algorithm. Our tight result indicates that uniform stability has reached its limit in stability analysis for the UD-based algorithm.

## 1 Introduction

Hyperparameters significantly influence the convergence behavior of learning algorithms as well as the efficiency and generalization performance of the trained model [1, 2, 3]. *Hyperparameter optimization* (HO) algorithms aim to find the best hyperparameters (associated with the optimized model parameters) on a validation set. Classical approaches for HO include grid search [4], random search [5], Bayesian optimization [6, 7, 8], and evolutionary algorithms [9, 10], which often suffer from the problem of scaling up. Recently, gradient-based methods have achieved excellent empirical performance in high-dimensional HO problems [11, 3, 12].

In gradient-based methods, HO is formulated as a bilevel programming problem. The inner level seeks the best model parameters on the training set given current hyperparameters. In the outer level, hyperparameters are optimized with gradient descent. However, the gradient is difficult to compute as it requires differentiating the optimized model parameters w.r.t. the hyperparameters. Two mainstream strategies have been developed to obtain this Jacobian by explicitly *unrolling differentiation* (UD) [13, 2, 14] or approximately applying the *implicit function theorem* (IFT) [1, 15, 16, 12].

---

*Correspondence to Chongxuan Li.

To investigate the underlying reason for their success, existing work establishes generalization upper bounds based on algorithmic stability [17, 18]. In particular, [17] presents a generalization framework associated with a notion of uniform stability for general bilevel programming in HO and stability upper bounds for the UD-based algorithm. Despite their efforts, such algorithms have not been fully understood and one of the key unsolved problems is whether existing stability analyses are tight.

To this end, this paper establishes lower bounds on the stability of gradient-based bilevel programming algorithms for HO. Technically, we begin by introducing *lower-bounded expansion properties* which inherently characterize the instability in general update rules including stochastic gradient descent (SGD) as detailed in Section 4. Our expansion properties, to a certain degree, mirror those introduced by [19], with the distinction being our emphasis on lower bounds rather than upper bounds. This approach not only enables a comparative analysis with upper bounds to evaluate their alignment (i.e., tightness) but also lays down a conceptual framework for analyzing the lower bounds of algorithmic stability, generally applicable to both single-level SGD and various bilevel algorithms.

Building upon these properties, we explore the stability of the UD-based algorithm in Section 5. We first present a recursive stability lower bound that aligns with the existing upper bound at the outer level given the expansion properties of the compound validation loss, followed by an analysis of the Lipschitz constant of the inner output to maximize those expansion coefficients. Guided by these theoretical insights, we construct a quadratic example that yields a tight lower bound for the UD-based algorithm with constant step sizes and a nearly tight lower bound with linearly decreasing step sizes with respect to key factors. Meaningful bounds for a representative IFT-based algorithm are also provided in Appendix C based on its essential connection to UD-based methods. We highlight that the example is carefully designed to obtain explicit stability lower bounds by overcoming the challenges posed by the intricate behavior of the bilevel algorithms, i.e. the dependence of each outer-level update on the current turn of inner optimization.

We outline our contributions as follows: (1) We introduce lower-bounded expansion properties that can serve as general tools for analyzing lower bounds of the stability in gradient descent. (2) To our knowledge, we present the first lower bounds of uniform stability for both the UD-based and representative IFT-based algorithms, facing the challenge posed by the intricate formulation of the outer update in bilevel optimization. (3) Our lower bounds match existing upper bounds for the UD-based algorithm, verifying that uniform stability has reached its limit in characterizing the generalization of the UD-based algorithm. Detailed results are summarized in Table 1.

## 2   Related work

Algorithmic stability [20, 21] measures the change in the model output when a single training example is replaced. It is shown to be sufficient and necessary for learnability in certain cases [22]. Stability-based generalization analysis of an algorithm typically consists of three key elements: a notion of stability, a stability-based generalization bound, and a stability analysis depending on the algorithm. Below, we introduce the related work based on these three elements.

**Algorithmic stability.** [23] introduce uniform stability, which characterizes the worst-case change of loss and presents a stability-based generalization bound with high probability. Notable efforts [24, 25, 26, 27] have been made to obtain sharper bounds for uniformly stable algorithms in general. Besides the uniform stability, various notions of stability that characterize the average change [22], local change [28], or change in the hypothesis [29, 30] are investigated for fine-grained analyses.

**Stability of stochastic gradient descent (SGD).** SGD has been one of the workhorses in deep learning and therefore attracted much attention. To unravel the mystery behind its success, [19] analyze its (randomized) uniform stability on (strongly) convex and nonconvex losses. [31] analyze the uniform stability of (S)GD for nonsmooth convex losses and provide sharp upper and lower expectation bounds. [32, 30] consider the on-average stability for SGD and build data-dependent generalization bounds to explain the effectiveness of practical techniques like proper initialization.

Recent work establishes lower bounds on the uniform stability of SGD and investigates the tightness of corresponding upper bounds. [33] proves a general minimax optimal lower bound for stability generalization error together with optimization error on convex and smooth losses. [31] finds the general technique proposed by [33] is sub-optimal in convex but nonsmooth cases, and provides

sharper lower bounds by constructing a special class of loss functions. [34] adopts a similar approach by construction to present lower bounds for smooth and potentially nonconvex loss functions.

In this paper, we focus on the smooth and nonconvex cases in HO and provide stability lower bounds by construction as [31] and [34]. Compared with [34], we consider a more complicated and nontrivial bilevel optimization problem, where the interaction between inner and outer processes brings a significant impact on the analysis. A detailed comparison is provided in Section 5.3 and Appendix E.

**Stability for bilevel programming.** [17] extend the notion of uniform stability to HO and analyze stability upper bounds of the UD-based algorithm, while the tightness of their result is largely open. Recently, it has been extended to the analysis of implicit gradient algorithms [18]. This paper provides the first lower bounds, generally applying to two main categories of gradient-based HO methods. There are other bilevel optimization algorithms [12, 35, 36, 37, 38, 39, 15] and settings [2, 16, 3, 40, 41, 42, 43] where our framework can potentially be extended in future work.

# 3 Problem formulation

## 3.1 Elementary notations and definitions

**Scalar, vector, and matrix.** We employ lowercase letters (e.g., $a$), lowercase boldface letters (e.g., $\boldsymbol{a}$), and uppercase boldface letters (e.g., $\boldsymbol{A}$) to denote scalars, vectors, and matrices, respectively. For a vector $\boldsymbol{a}$, $\|\cdot\|$ denotes its Euclidean norm. For a matrix $\boldsymbol{A}$, $\|\cdot\|$ denotes its spectral norm. Additionally, let $\boldsymbol{a}_1 \stackrel{\circ}{=} \boldsymbol{a}_2$ denote $\boldsymbol{a}_1$ and $\boldsymbol{a}_2$ are collinear by a non-negative factor, namely, $\exists a \geq 0$ s.t. $\boldsymbol{a}_1 = a\boldsymbol{a}_2$.

**Loss function.** A differentiable function $\ell : \Omega \to \mathbb{R}$ is $L$-Lipschitz continuous if $\forall \boldsymbol{u}, \boldsymbol{v} \in \Omega$, $\|\ell(\boldsymbol{u}) - \ell(\boldsymbol{v})\| \leq L\|\boldsymbol{u} - \boldsymbol{v}\|$. It is $\gamma$-smooth if $\forall \boldsymbol{u}, \boldsymbol{v} \in \Omega$, $\|\nabla\ell(\boldsymbol{u}) - \nabla\ell(\boldsymbol{v})\| \leq \gamma\|\boldsymbol{u} - \boldsymbol{v}\|$.

**Twin datasets.** A pair of datasets are considered *twin datasets* if they differ in only a single data point, denoted by $S \simeq \tilde{S}$. Throughout this paper, we use a tilde symbol to distinguish their corresponding notions, e.g., examples $\boldsymbol{z}$ and $\tilde{\boldsymbol{z}}$, output parameters $\boldsymbol{w}$ and $\tilde{\boldsymbol{w}}$.

**Asymptotic notations.** Denote with $a_n \lesssim b_n$ that $a_n$ is bounded above by $b_n$ up to a constant factor for sufficiently large $n$, and conversely by $a_n \gtrsim b_n$. We say $a_n \asymp b_n$ if $a_n \lesssim b_n$ and $a_n \gtrsim b_n$.

## 3.2 HO as bilevel programming

Denote the testing, validation, and training distributions on the data space $\mathcal{Z}$ by $\mathcal{D}^{\text{test}}$, $\mathcal{D}^{\text{val}}$ and $\mathcal{D}^{\text{tr}}$, and corresponding losses by $\ell^{\text{test}}$, $\ell^{\text{val}}$ and $\ell^{\text{tr}}$. Since the validation phase is generally regarded as a rehearsal for testing, $\mathcal{D}^{\text{val}}$ and $\ell^{\text{val}}$ are commonly assumed to be consistent with $\mathcal{D}^{\text{test}}$ and $\ell^{\text{test}}$.

Given a validation set $S^{\text{val}} = \{\boldsymbol{z}_i^{\text{val}}\}_{i=1}^m \stackrel{\text{i.i.d.}}{\sim} (\mathcal{D}^{\text{val}})^m$ and a training set $S^{\text{tr}} = \{\boldsymbol{z}_j^{\text{tr}}\}_{j=1}^n \stackrel{\text{i.i.d.}}{\sim} (\mathcal{D}^{\text{tr}})^n$, HO algorithms seek the best-performing hyperparameter-parameter pair on $S^{\text{val}}$. Denote by $\boldsymbol{\lambda}$ the hyperparameter in space $\Lambda$, $\boldsymbol{\theta}$ the (model) parameter in space $\Theta$. This process can be formulated as the following bilevel problem:

$$\hat{\boldsymbol{\lambda}} \approx \underset{\boldsymbol{\lambda} \in \Lambda}{\arg\min} \frac{1}{m} \sum_{i=1}^m \ell^{\text{val}}(\boldsymbol{\lambda}, \hat{\boldsymbol{\theta}}(\boldsymbol{\lambda}); \boldsymbol{z}_i^{\text{val}}), \text{ where } \hat{\boldsymbol{\theta}}(\boldsymbol{\lambda}) \approx \underset{\boldsymbol{\theta} \in \Theta}{\arg\min} \frac{1}{n} \sum_{j=1}^n \ell^{\text{tr}}(\boldsymbol{\lambda}, \boldsymbol{\theta}; \boldsymbol{z}_j^{\text{tr}}).$$

Here, $\hat{\boldsymbol{\theta}}(\boldsymbol{\lambda})$ is selected by its training performance under the given $\boldsymbol{\lambda}$ and $\ell^{\text{val}}(\boldsymbol{\lambda}, \hat{\boldsymbol{\theta}}(\boldsymbol{\lambda}); \boldsymbol{z})$ can be rewritten as a *compound validation loss* $\mathcal{L}(\boldsymbol{\lambda}; \boldsymbol{z})$ considering $\hat{\boldsymbol{\theta}}(\boldsymbol{\lambda})$ a function of $\boldsymbol{\lambda}$.

Various methods are proposed to solve this nested problem, among which gradient-based algorithms have recently achieved success in scalability [11, 3, 12]. As shown in Algorithm 1, gradient-based methods utilize SGD as the optimizer at both levels, where the primary challenge lies in the calculation of the gradient of the compound validation loss, called *hypergradient*,

$$\nabla_{\boldsymbol{\lambda}} \mathcal{L}(\boldsymbol{\lambda}; \boldsymbol{z}) = \nabla_{\boldsymbol{\lambda}} \ell^{\text{val}}(\boldsymbol{\lambda}, \boldsymbol{\theta}_K(\boldsymbol{\lambda}); \boldsymbol{z}) + \underbrace{\nabla_{\boldsymbol{\lambda}} \boldsymbol{\theta}_K(\boldsymbol{\lambda})}_{\text{inner Jacobian}} \nabla_{\boldsymbol{\theta}} \ell^{\text{val}}(\boldsymbol{\lambda}, \boldsymbol{\theta}_K(\boldsymbol{\lambda}); \boldsymbol{z}), \tag{1}$$

where the *inner Jacobian* involves differentiating through the inner-level optimization. To this end, the UD-based methods obtain the exact inner Jacobian by directly unrolling the inner differentiation:[2]

$$\nabla_{\boldsymbol{\lambda}}\boldsymbol{\theta}_K(\boldsymbol{\lambda}) = -\sum_{k=0}^{K-1}\eta_{k+1}\nabla_{\boldsymbol{\theta}\boldsymbol{\lambda}}^2\ell^{\mathrm{tr}}(\boldsymbol{\lambda},\boldsymbol{\theta}_k)\prod_{i=k+1}^{K}(\boldsymbol{I}-\eta_{i+1}\nabla_{\boldsymbol{\theta}\boldsymbol{\theta}}^2\ell^{\mathrm{tr}}(\boldsymbol{\lambda},\boldsymbol{\theta}_i)). \quad (2)$$

While representative IFT-based methods leverage the implicit function theorem and Neumann series to obtain an alternative estimation [12]:

$$\widehat{\nabla_{\boldsymbol{\lambda}}\boldsymbol{\theta}_K}(\boldsymbol{\lambda}) = -\eta_K\nabla_{\boldsymbol{\theta}\boldsymbol{\lambda}}^2\ell^{\mathrm{tr}}(\boldsymbol{\lambda},\boldsymbol{\theta}_K)\sum_{k=0}^{K-1}\Big[\boldsymbol{I}-\eta_K\nabla_{\boldsymbol{\theta}\boldsymbol{\theta}}^2\ell^{\mathrm{tr}}(\boldsymbol{\lambda},\boldsymbol{\theta}_K)\Big]^k. \quad (3)$$

Please refer to Algorithm 1 for the whole process. Notably, this paper adopts a common theoretical assumption [19, 17] of constant inner step sizes and decreasing outer step sizes.[3]

---

**Algorithm 1** Gradient-based bilevel HO

1: **Input:** Initialization $\boldsymbol{\lambda}_0$ and $\boldsymbol{\theta}_0$; training set $S^{\mathrm{tr}}$ and validation set $S^{\mathrm{val}}$; step size scheme $\alpha$ and $\eta$
2: **Output:** The hyperparameter $\boldsymbol{\lambda}_T$ and hypothesis $\boldsymbol{\theta}_K$
3: **for** $t = 1$ **to** $T$ **do**
4:     **for** $k = 1$ **to** $K$ **do**
5:         uniformly sampling $j_k$ from $[n]$
6:         $\boldsymbol{\theta}_k \leftarrow \boldsymbol{\theta}_{k-1} - \eta_k\nabla_{\boldsymbol{\theta}}\ell^{\mathrm{tr}}(\boldsymbol{\lambda}_{t-1},\boldsymbol{\theta}_{k-1};\boldsymbol{z}_{j_k}^{\mathrm{tr}})$
7:     **end for**
8:     uniformly sampling $i_t$ from $[m]$
9:     $\boldsymbol{g} \leftarrow \nabla\mathcal{L}(\boldsymbol{\lambda}_{t-1};\boldsymbol{z}_{i_t}^{\mathrm{val}})$     ▷ UD-based algorithm in Eq. (2), IFT-based algorithm in Eq. (3)
10:     $\boldsymbol{\lambda}_t \leftarrow \boldsymbol{\lambda}_{t-1} - \alpha_t\boldsymbol{g}$
11: **end for**
12: **return** $\boldsymbol{\lambda}_T$ and $\boldsymbol{\theta}_K$

---

### 3.3 Generalization and stability of HO

The generalization behavior of HO algorithms characterizes the selected model's potential performance on the unseen test data. Specifically, denoting the hyperparameter output by a stochastic HO algorithm $\mathcal{A}$ as $\mathcal{A}(S^{\mathrm{val}}, S^{\mathrm{tr}})$, we are interested in the difference between its expected testing risk and empirical validation risk, namely *generalization error* defined as

$$\epsilon_{\mathrm{gen}} := \mathbb{E}_{\mathcal{A},S^{\mathrm{val}},S^{\mathrm{tr}}}\left[\mathbb{E}_{\boldsymbol{z}\sim\mathcal{D}^{\mathrm{test}}}[\mathcal{L}(\mathcal{A}(S^{\mathrm{val}},S^{\mathrm{tr}});\boldsymbol{z})] - \frac{1}{m}\sum_{i=1}^{m}\mathcal{L}(\mathcal{A}(S^{\mathrm{val}},S^{\mathrm{tr}});\boldsymbol{z}_i^{\mathrm{val}})\right]. \quad (4)$$

Stability-based generalization theory turns this problem into measuring the algorithmic robustness. We define the notion of *uniform argument stability* for HO algorithms, which captures the variation in algorithm outputs when replacing a single validation point.[4]

**Definition 3.1** (Uniformly argument stability on validation). A stochastic HO algorithm $\mathcal{A}$ is $\epsilon_{\mathrm{arg}}$-uniformly argument stable on validation where

$$\epsilon_{\mathrm{arg}} := \sup_{S^{\mathrm{val}}\simeq\tilde{S}^{\mathrm{val}}\in\mathcal{Z}^m, S^{\mathrm{tr}}\in\mathcal{Z}^n}\mathbb{E}_{\mathcal{A}}[\|\mathcal{A}(S^{\mathrm{val}},S^{\mathrm{tr}}) - \mathcal{A}(\tilde{S}^{\mathrm{val}},S^{\mathrm{tr}})\|]. \quad (5)$$

Our analysis mainly leverages this notion following [31], as it is the key measure for stability bounds under the Lipschitz continuous condition. $\epsilon_{\mathrm{arg}}$ differs from the uniform stability $\epsilon_{\mathrm{stab}}$ defined in [17, Definition 1] only by a Lipschitz constant $L$ (i.e., $\epsilon_{\mathrm{stab}} \leq L\epsilon_{\mathrm{arg}}$), and our results for $\epsilon_{\mathrm{stab}}$ are also

---

[2]For formula neatness, we set an unused $\eta_{K+1} = 0$ as a placeholder, and similarly $\alpha_{T+1} = 0$ in Eq. (6).
[3]Namely, $\eta_k = \eta$ and $\alpha_t \leq \frac{c}{t}$, with a constant $c > 0$. The decreasing step size is also widely adopted in the optimization convergence analysis works, such as SGD [44, 45], AdaGrad [46], Adam [47, 48].
[4]Definition 3.1 considers perturbing the validation set instead of the training set. The relevant discussion is provided in Appendix G.1

provided in Theorem 5.6 and Theorem C.7 for direct comparison with former works. Existing stability-based generalization bound [17, Theorem 1] shows that uniform stability guarantees generalization in expectation for HO algorithms that $\epsilon_{\text{gen}} \leq \epsilon_{\text{stab}}$.

Former work [17] constructs the first stability upper bound for UD-based HO algorithms. This result is fundamentally based on an upper-bounded recurrence relation of the distance between the outputs respectively optimized on twin validation sets, denoted by $\delta_t := \|\boldsymbol{\lambda}_t - \tilde{\boldsymbol{\lambda}}_t\|$ at the $t$-th step.

**Theorem 3.2** (Recursion upper bound for UD, Theorems 2 and 3, [17]). *Suppose the compound validation loss $\mathcal{L}(\cdot; \boldsymbol{z})$ is $L$-Lipschitz continuous and $\gamma$-smooth for all $\boldsymbol{z} \in \mathcal{Z}$, and the training loss $\ell^{\text{tr}}(\boldsymbol{\lambda}, \cdot; \boldsymbol{z})$ is $\gamma^{\text{tr}}$-smooth for all $\boldsymbol{\lambda} \in \Lambda$ and $\boldsymbol{z} \in \mathcal{Z}$. Then for all $1 \leq t \leq T$, $\mathbb{E}_{\mathcal{A}}[\delta_t] \leq \left[1 + (1 - 1/m)\alpha_t \gamma\right]\mathbb{E}_{\mathcal{A}}[\delta_{t-1}] + \frac{2\alpha_t L}{m}$, where $L \lesssim (1 + \eta \gamma^{\text{tr}})^K, \gamma \lesssim (1 + \eta \gamma^{\text{tr}})^{2K}$.*

Unrolling this recursion, we directly get the stability upper bound in recursion form:

$$\epsilon_{\text{arg}} \leq \sum_{t=1}^{T} \prod_{s=t+1}^{T+1} \left(1 + \alpha_s(1 - 1/m)\gamma\right)\frac{2\alpha_t L}{m}. \tag{6}$$

As this result does not explicitly display its order w.r.t. $T$ under decreasing step sizes $\alpha_t \leq c/t$, [17] further deforms Eq. (6) with the bounded loss condition to obtain $\epsilon_{\text{stab}} \lesssim T^{\frac{(1-1/m)\gamma c}{(1-1/m)\gamma c + 1}}/m$.

[18] analyzes a specific IFT-based algorithm, which, under certain assumptions, achieves a similar result of $\epsilon_{\text{stab}} \lesssim T^q/m$ with $q < 1$. Though these stability upper bounds have been established, their tightness is rarely explored, and the stability of IFT-based algorithms remains largely open. Therefore, this paper takes a first step towards establishing stability lower bounds (namely, how unstable an algorithm can be) for both UD-based and IFT-based HO algorithms.

# 4 Expansion properties of update rules

This paper endeavors to establish tight lower bounds for uniform (argument) stability as defined in Definition 3.1, which is fundamentally the supremum of the output divergence. For iterative algorithms, this divergence accumulates recursively across the whole optimization process. Therefore, we first introduce *lower-bounded expansion properties* in Section 4.1 to characterize update rules that will induce guaranteed divergence at each iteration. This is followed by an analysis in Section 4.2 on how the objective functions within SGD need to be structured to satisfy these properties. We will see in Section 5 that, for gradient-based HO algorithms, these properties jointly lead to a lower-bounded divergence recursion given the outer-level update properties in Theorem 5.1 and a lower-bounded Lipschitz constant of the inner output given the inner-level update properties in Theorem 5.2.

Our expansion properties correspond, to some extent, with those presented by [19] and the key difference lies in our focus on lower bounds rather than upper bounds. This approach not only facilitates comparisons with upper bounds to discuss their alignments (i.e. tightness) but also provides a general framework for analyzing the lower bounds of algorithmic stability.

## 4.1 Lower-bounded expansion properties of general iterative algorithms

Let $\boldsymbol{w}$ be a general notation for parameters (or hyperparameters) in space $\Omega$. An update rule is a function $G : \Omega \to \Omega$ that maps $\boldsymbol{w}$ to its next state $G(\boldsymbol{w})$, and an iterative algorithm is composed of a series of consecutive update rules. We denote two sequences of update rules by $\{G_t\}_{t=1}^{T}$ and $\{G'_t\}_{t=1}^{T}$, and the corresponding outputs by $\{\boldsymbol{w}_t\}_{t=1}^{T}$ and $\{\boldsymbol{w}'_t\}_{t=1}^{T}$.

Intrinsically, the divergence between $\boldsymbol{w}_t$ and $\boldsymbol{w}'_t$ dynamically evolves across the entire process, driven by two factors: disparity in current update rules, and difference in current parameters resulting from prior updates. Our goal is to systematically analyze how variations between two update sequences lead to substantial divergence in outputs. In the following, we introduce Definition 4.1 and Definition 4.2 correspondingly to characterize properties of update rules leading to increasing divergence.

**Definition 4.1** ($\sigma$-divergent). Two update rules $G$ and $G'$ are $\sigma$-*divergent along $\boldsymbol{v}$* if for all $\boldsymbol{w} \in \Omega$,
$$G(\boldsymbol{w}) - G'(\boldsymbol{w}) \doteq \boldsymbol{v}, \|G(\boldsymbol{w}) - G'(\boldsymbol{w})\| \geq \sigma.$$

**Definition 4.2** ($\rho$-growing). An update rule $G$ is $\rho$-*growing along $\boldsymbol{v}$* if for all $\boldsymbol{w}, \boldsymbol{w}' \in \Omega$ such that $\boldsymbol{w} - \boldsymbol{w}'$ parallel with $\boldsymbol{v}$,
$$G(\boldsymbol{w}) - G(\boldsymbol{w}') \stackrel{\circ}{=} \boldsymbol{w} - \boldsymbol{w}', \|G(\boldsymbol{w}) - G(\boldsymbol{w}')\| \geq \rho\|\boldsymbol{w} - \boldsymbol{w}'\|.$$

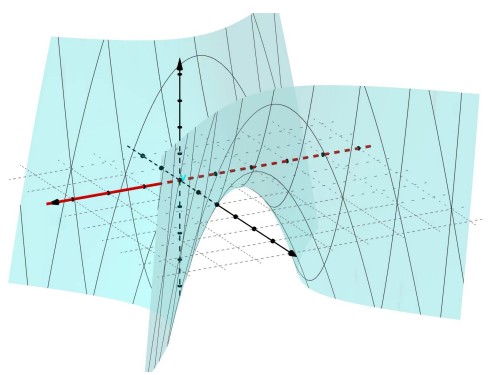

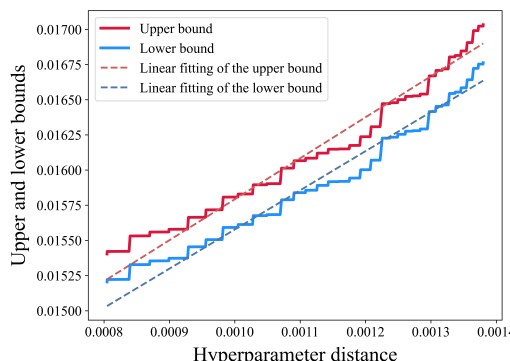

Figure 1: The loss surface of a two-dimensional example: $\ell(\boldsymbol{\theta}) = \frac{1}{2}\boldsymbol{w}^\top \boldsymbol{A}\boldsymbol{w} - \boldsymbol{w}$ where $\boldsymbol{A} = \begin{bmatrix} -1 & 0 \\ 0 & 1 \end{bmatrix}$. The direction $\boldsymbol{v}_1 = \begin{bmatrix} 1 \\ 0 \end{bmatrix}$ is highlighted in red, along which $\ell$ exhibits expansive property.

Figure 2: Practical output distances vs. theoretical bounds in Theorem 5.5. We implement UD-based Algorithm 1 on Example 5.3. The output hyperparameter distances with increasing $T$ are plotted on the horizontal axis. The upper/lower bounds with corresponding $T$ are plotted on the vertical axis. The linear trends suggest these three values are of almost the same order w.r.t. $T$.

Intuitively, $\sigma$-divergent update rules produce sufficiently divergent output parameters and a $\rho$-growing update rule scales the divergence between parameters with a sufficiently large factor. The direction $\boldsymbol{v}$ is chosen as the most expansive direction, as detailed with a concrete example in Section 5.4.

## 4.2  Lower-bounded expansion properties of SGD

One-step SGD can be generally formulated as $G_{\ell,\alpha}(\boldsymbol{w}) = \boldsymbol{w} - \alpha\nabla\ell(\boldsymbol{w})$, where the loss function directly impacts this gradient-based update rule. We now define the $\mu$-expansive property for the loss function which leads to the growing property of SGD.

**Definition 4.3** ($\mu$-expansive)**.** A differentiable function $\ell : \Omega \to \mathbb{R}$ is $\mu$-expansive along $\boldsymbol{v}$ if for all $\boldsymbol{w}, \boldsymbol{w}' \in \Omega$ that $\boldsymbol{w} - \boldsymbol{w}'$ parallel with $\boldsymbol{v}$, there exists $\mu_{\boldsymbol{w},\boldsymbol{w}'} \geq \mu$ such that

$$\nabla\ell(\boldsymbol{w}) - \nabla\ell(\boldsymbol{w}') = -\mu_{\boldsymbol{w},\boldsymbol{w}'}(\boldsymbol{w} - \boldsymbol{w}').$$

This paper mainly focuses on the case when $\mu > 0$ where the loss function is nonconvex. We have $\mu \leq 0$ for the convex case. When $\mu > 0$, Definition 4.3 connects to $\mu$-strongly concavity.[5] These concepts are equivalent in the one-dimensional case. In general, $\mu$-strongly concavity imposes uniform curvature in all directions, while $\mu$-expansiveness restricts concavity in only one direction with an additional restriction for the colinearity of $\nabla\ell(\boldsymbol{w}) - \nabla\ell(\boldsymbol{w}')$ and $-(\boldsymbol{w} - \boldsymbol{w}')$. See Appendix G.2 for details. We illustrate a simple loss function in Fig. 1 that satisfies Definition 4.3.

Notably, the directional restrictions on the update rules in Definitions 4.1 to 4.3 simplify the lower-bound calculations as it enables us to focus only on the norm of the divergence at each step and get rid of directional variation, which aids in a clearer understanding of divergence dynamics. As a first attempt to establish stability lower bounds for bilevel optimization problems, our work leaves open whether these conditions can be relaxed. A potential approach might involve requiring the divergence to exhibit a specific directional component rather than strict alignment as in the current definitions.

The following lemma shows that the expansiveness of the loss function can induce the growing property of SGD.

**Lemma 4.4** (Growing property of SGD with expansive loss function, proof in Appendix B.2)**.** *Suppose $\ell$ is $\mu$-expansive along $\boldsymbol{v}$ and $1 + \alpha\mu \geq 0$, then $G_{\ell,\alpha}$ is $(1 + \alpha\mu)$-growing along $\boldsymbol{v}$.*

---

[5]Namely, $\forall \boldsymbol{w}, \boldsymbol{w}' \in \Omega, \langle \nabla\ell(\boldsymbol{w}) - \nabla\ell(\boldsymbol{w}'), \boldsymbol{w} - \boldsymbol{w}' \rangle \leq -\mu\|\boldsymbol{w} - \boldsymbol{w}'\|^2$.

# 5 Lower bounds on uniform stability in HO

Based on tools introduced in Section 4, this section proceeds to precisely characterize the stability of gradient-based bilevel HO algorithms. In Section 5.1, we provide a lower-bounded recursion of hyperparameter divergence that aligns with Theorem 3.2 given the expansion properties of the outer optimization, followed by a lower bound of Lipschitz constant of the inner output given the expansion properties of the inner optimization in Section 5.2. These findings pose insights in the construction of Example 5.3 at both the inner and outer levels to maximize the instability of HO algorithms. This quadratic example produces a tight lower bound for UD-based algorithms, detailed in Section 5.4, and meaningful bounds for IFT-based algorithms, provided in Appendix C.

## 5.1 Stability lower bound given outer-level expansion properties

We first establish a uniform argument stability lower bound by considering the outer level of the bilevel programming as a single-level optimization problem w.r.t. the hyperparameters. This approach takes the compound validation loss $\mathcal{L}$ as a whole, temporarily disregarding the dependence of this loss on the inner-level solution and the inner Jacobian.

Suppose $S^{\mathrm{val}}$ and $\tilde{S}^{\mathrm{val}}$ are twin validation sets differing only on the $i$-th entry, and denote the sequences of update rules on $S^{\mathrm{val}}$ and $\tilde{S}^{\mathrm{val}}$ as $\{G_{\boldsymbol{z}_{i_t},\alpha_t}\}_{t=1}^T$ and $\{G_{\tilde{\boldsymbol{z}}_{i_t},\alpha_t}\}_{t=1}^T$ [6] respectively. According to Definition 3.1, the uniform argument stability is lower-bounded by the hyparameter distance after $T$ steps, which primarily depends on the divergent property of update rules on different examples and the expansiveness of the compound validation loss. By characterizing these properties and utilizing Lemmas 4.4 and B.2, we obtain a lower bound of the stability in the following Theorem 5.1.

**Theorem 5.1** (Lower bound given outer-level expansion properties, proof in Appendix B.3). *Suppose there exists a nonzero vector $\boldsymbol{v}$ along which $G_{\boldsymbol{z}_i,\alpha_t}$ and $G_{\tilde{\boldsymbol{z}}_i,\alpha_t}$ are $2\alpha_t L'$-divergent and $\mathcal{L}(\cdot;\boldsymbol{z})$ is $\gamma'$-expansive for all $\boldsymbol{z} \in S^{\mathrm{val}}$. Then we have $\mathbb{E}_{\mathcal{A}}[\delta_t] \geq \left[1 + \alpha_t(1 - \frac{1}{m})\gamma'\right]\mathbb{E}_{\mathcal{A}}[\delta_{t-1}] + \frac{2\alpha_t L'}{m}$ and*

$$
\epsilon_{\mathrm{arg}} \geq \sum_{t=1}^{T} \prod_{s=t+1}^{T+1} \left(1 + \alpha_s(1 - 1/m)\gamma'\right)\frac{2\alpha_t L'}{m}.
$$

Theorem 5.1 echos the upper bound formulation in Eq. (6). The distinctions arise solely in the smooth/expansive coefficients $\gamma/\gamma'$ and the continuous/divergent coefficients $L/L'$. Consequently, the alignment of these two bounds (i.e., their tightness) hinges on the values of these coefficients. As detailed later in Section 5.2, we delve deeper into the coefficients present in our lower bound by unfolding the bilevel problem, focusing on the solution of the inner level and its Jacobian.

Further, Theorem 5.1 not only applies to all HO algorithms that employs outer-level SGD but also to single-level SGD. In the context of single-level SGD, the expansion properties can be directly inferred from the loss function, as elaborated in Appendix E.

## 5.2 Lipschitz lower bound given inner-level expansion properties

Based on Theorem 5.1, our next step towards building stability lower bounds is analyzing the expansive coefficient $\gamma'$ and divergent coefficient $L'$ of outer-level optimization where the hypergradient is used for update. As can be observed in Eq. (1), the inner Jacobian $\nabla_{\boldsymbol{\lambda}}\boldsymbol{\theta}_K(\boldsymbol{\lambda})$ is a key bridge between the inner and outer level that significantly influence the hypergradient. Here, we measure the lower bound of its maximum volume, i.e., the Lipschitz continuity coefficient of $\boldsymbol{\theta}_K(\boldsymbol{\lambda})$ regarding $\boldsymbol{\lambda}$ denoted by $L^{\boldsymbol{\theta}_K}$, to provide a guarantee for the effect of the hypergradient.

Denote corresponding inner update rules as $G_{\boldsymbol{\lambda},\eta}$ and $G_{\boldsymbol{\lambda}',\eta}$ given hyperparameters $\boldsymbol{\lambda}$ and $\boldsymbol{\lambda}'$. Applying them consecutively for $K$ times, we get two sequences of inner updates. Theorem 5.2 presents how the expansion properties of the inner problem characterize the lower bound of $L^{\boldsymbol{\theta}_K}$.

**Theorem 5.2** (Lower bound of Lipschitz of the inner-level solution, proof in Appendix B.4). *Given any two hyperparameters $\boldsymbol{\lambda}, \boldsymbol{\lambda}' \in \Lambda$, suppose there exists a nonzero vector $\boldsymbol{v}$ along which $G_{\boldsymbol{\lambda},\eta}$ and*

---

[6]We slightly abuse the notation in the subscript of the update rule as in Section 4.2, since the loss function in this contest can be solely distinguished by the selected sample. We use a similar simplification for the inner update rules in the next section.

$G_{\boldsymbol{\lambda}',\eta}$ are $\|\boldsymbol{\lambda} - \boldsymbol{\lambda}'\|\sigma^{\mathrm{tr}}$-divergent and $\ell^{\mathrm{tr}}(\boldsymbol{\lambda}, \cdot; \boldsymbol{z})$ is $\mu^{\mathrm{tr}}$-expansive for all $\boldsymbol{z} \in S^{\mathrm{tr}}$ with $\mu^{\mathrm{tr}} > 0$. Then we have

$$L^{\boldsymbol{\theta}_K} \geq \frac{\sigma^{\mathrm{tr}}}{\eta\mu^{\mathrm{tr}}}[(1 + \eta\mu^{\mathrm{tr}})^K - 1].$$

Omitting constants that depend on $\eta$, $\sigma^{\mathrm{tr}}$, and $\mu^{\mathrm{tr}}$, we get $L^{\boldsymbol{\theta}_K} \gtrsim (1 + \eta\mu^{\mathrm{tr}})^K$.

It is worth mentioning that our lower bound for $L^{\boldsymbol{\theta}_K}$ is matched with its upper bound in [17] (see in Theorem 3 and Proposition 2), which prepares us to obtain a tight lower bound.

## 5.3 An example with maximal simplification

Motivated by Theorems 5.1 and 5.2, the following example is carefully constructed, exhibiting all expansive and divergent properties as required by these theorems to establish tight lower bounds on uniform argument stability of gradient-based HO algorithms.

**Example 5.3.** We introduce an HO problem as follows. The validation loss and training loss are given by:

$$\ell^{\mathrm{val}}(\boldsymbol{\lambda}, \boldsymbol{\theta}; \boldsymbol{z}) = \ell^{\mathrm{tr}}(\boldsymbol{\lambda}, \boldsymbol{\theta}; \boldsymbol{z}) = \frac{1}{2}\boldsymbol{\theta}^\top \boldsymbol{A}\boldsymbol{\theta} + \boldsymbol{\lambda}^\top \boldsymbol{\theta} - y\boldsymbol{x}^\top \boldsymbol{\theta},$$

where $\boldsymbol{A} \in \mathbb{R}^{d \times d}$ is symmetric. Denote the eigenvalues of $\boldsymbol{A}$ as $\gamma_1 \leq \cdots \leq \gamma_d$. Let $\gamma_1 < 0$ and $|\gamma_1| \geq |\gamma_d|$, and $\boldsymbol{v}_1$ be a unit eigenvector for $\gamma_1$. Let $S^{\mathrm{val}}$ and $\tilde{S}^{\mathrm{val}}$ be a pair of twin validation datasets differing at the $i$-th example where

$$\boldsymbol{z}_i = (\boldsymbol{x}_i, y_i) = (\boldsymbol{v}_1, 1), \tilde{\boldsymbol{z}}_i = (\tilde{\boldsymbol{x}}_i, \tilde{y}_i) = (-\boldsymbol{v}_1, 1).$$

In this example, $\boldsymbol{A}$ determines the convexity of the problem. Throughout the main text, we consider the most common nonconvex case where $\boldsymbol{A}$ is indefinite and symmetric. See Appendix D for the results of (strongly) convex losses.

Notably, our example satisfies Assumption B.1 adopted for establishing the stability upper bounds where the loss functions are Lipschitz continuous and smooth. Hereafter we denote $\mathcal{L}(\cdot; \boldsymbol{z})$ as $L$-Lipschitz continuous and $\gamma$-smooth, and $\ell^{\mathrm{tr}}(\boldsymbol{\lambda}, \cdot; \boldsymbol{z})$ as $\gamma^{\mathrm{tr}}$-smooth, where $\gamma^{\mathrm{tr}} = |\gamma_1|$.

Example 5.3 is constructed adhering to the principle of maximal simplification. Specifically, the quadratic form is essential for inducing nonconvexity. The second bilinear cross term represents the simplest scenario for interaction between hyperparameters and parameters, ensuring a non-zero inner Jacobian. The final term provides a connection for parameters and data. $\ell^{\mathrm{val}}$ and $\ell^{\mathrm{tr}}$ are set to be identical here for simplicity, and our results do not fundamentally depend on their consistency.

We emphasize the role of the eigenvector (i.e., $\boldsymbol{v}_1$) which corresponds to the smallest eigenvalue. It represents the least convex direction, thereby offering the greatest expansiveness of the loss (see Fig. 1), and both the inner and outer optimizations attain the highest level of divergence and expansiveness in this direction. Consequently, in Example 5.3, the distinct samples in $S^{\mathrm{val}}$ and $\tilde{S}^{\mathrm{val}}$ are set to align reversely with $\boldsymbol{v}_1$ to make the HO algorithms unstable.

*Remark.* The constructed example is required to meet two essential criteria: first, it must reveal the instability inherent in the algorithms; second, it must allow precise calculation of the smoothness coefficient $\gamma$ and the expansion coefficient $\mu$ for the compound validation loss to verify the alignment between lower and upper bounds. Simultaneously satisfying these two requirements is challenging for bilevel algorithms. In Appendix G.3, we provide a ridge regression example to illustrate how the bilevel structure complicates the analysis of stability lower bounds.

## 5.4 Lower bounds of UD-based algorithms

The following proposition shows that Example 5.3 induces the expansion of UD-based algorithms.

**Proposition 5.4** (Expansion properties of UD-based algorithms, proof in Appendix B.5). *Suppose we solve Example 5.3 by UD-based Algorithm 1 with constant inner step size $\eta$ where $1 - \eta\gamma_d \geq 0$ and outer step size $\alpha_t$. Then (1) the outer update rules $G_{\boldsymbol{z}_i, \alpha_t}$ and $G_{\tilde{\boldsymbol{z}}_i, \alpha_t}$ are $2\alpha_t L'$- divergent along $\boldsymbol{v}_1$, and (2) the composite validation loss $\mathcal{L}(\cdot; \boldsymbol{z})$ is $\gamma'$-expansive along $\boldsymbol{v}_1$ for all $\boldsymbol{z} \in S^{\mathrm{val}}$, where*

$$L \asymp L' \asymp (1 + \eta\gamma^{\mathrm{tr}})^K, \gamma = \gamma' \asymp (1 + \eta\gamma^{\mathrm{tr}})^{2K}.$$

Combining the lower bound in Theorem 5.1 with the upper bound in Eq. (6), we instantly get

$$\sum_{t=1}^{T} \prod_{s=t+1}^{T} \left(1 + \alpha_s(1 - 1/m)\gamma\right)\frac{2\alpha_t L'}{m} \leq \epsilon_{\text{arg}} \leq \sum_{t=1}^{T} \prod_{s=t+1}^{T} \left(1 + \alpha_s(1 - 1/m)\gamma\right)\frac{2\alpha_t L}{m}, \quad (7)$$

where the bounds are in the same order w.r.t. $T$, $K$ and $m$. These matching bounds in recursion form verify the **tightness** of the existing upper bound [17].

Specifically, for constant step sizes, i.e., $\alpha_t = c$ for all $t$, Eq. (7) explicitly reveals the scale of $\epsilon_{\text{arg}}$ regarding $T$: $\epsilon_{\text{arg}} \asymp \left(1 + c(1 - 1/m)\gamma\right)^T/m$. However, for linearly decreasing step sizes $\alpha_t \leq c/t$, additional scaling steps [7] are necessary and the deformed result is provided below.

**Theorem 5.5** (Uniform argument stability of UD-based algorithms, proof in Appendix B.6). *Solving Example 5.3 by UD-based Algorithm 1 with constant inner step size $\eta$ where $1 - \eta\gamma_d \geq 0$ and decreasing outer step sizes $\alpha_t = c/t$ with $c$ as a positive constant has uniform argument stability that*

$$\frac{T^{\ln\left(1+(1-\frac{1}{m})c\gamma'\right)}}{m} \lesssim \epsilon_{\text{arg}} \lesssim \frac{T^{(1-\frac{1}{m})c\gamma}}{m},$$

*where $\gamma = \gamma' \asymp (1 + \eta\gamma^{\text{tr}})^{2K}$ as in Proposition 5.4.*

The scaling steps unavoidably create a discrepancy between the deformed lower and upper bounds, while their quotient, $T^{(1-\frac{1}{m})c\gamma-\ln\left(1+(1-\frac{1}{m})c\gamma'\right)}$, is small given a small $c$ (e.g., 0.01). We compare the practical output hyperparameter distances and the theoretical bounds in Fig. 2.

Notably, the upper bound in our result is not contradictory to the existing upper bound of $\epsilon_{\text{arg}} \lesssim T^{\frac{(1-1/m)\gamma c}{(1-1/m)\gamma c+1}}/m$ in [17] because we remove the bounded loss assumption, i.e., $\exists a, b \in \mathbb{R}$ s.t. $\mathcal{L} \in [a, b]$. This modification is necessary to fairly compare the upper and lower bounds. Detailed discussion is provided in Appendix E.3.

Based on the results of uniform argument stability, we can further obtain similar results of uniform stability by introducing additional assumptions as below.

**Theorem 5.6** (Uniform stability of UD-based algorithms, proof in Appendix B.7). *Following the same condition as in Theorem 5.5, and additionally, if the initial points $\boldsymbol{\theta}_0 = \mathbf{0}, \boldsymbol{\lambda}_0 = \mathbf{0}$, and $\boldsymbol{v}_1 \perp \boldsymbol{x}_j^{\text{val}}$ for any $j \in [m]\backslash i$ and $\boldsymbol{v}_1 \perp \boldsymbol{x}_j^{\text{tr}}$ for any $j \in [n]$, then Algorithm 1 has uniform stability that $\frac{T^{\ln\left(1+(1-\frac{1}{m})c\gamma'\right)}}{m} \lesssim \epsilon_{\text{stab}} \lesssim \frac{T^{(1-\frac{1}{m})c\gamma}}{m}$, where $\gamma = \gamma' \asymp (1 + \eta\gamma^{\text{tr}})^{2K}$ as in Proposition 5.4.*

Technically, we adopt these additional assumptions following [34] to simplify the formulation of $\mathcal{L}(\boldsymbol{\lambda}_T, \boldsymbol{z}) - \mathcal{L}(\boldsymbol{\lambda}'_T, \boldsymbol{z})$ by eliminating the quadratic term and reducing it to be colinear with $\boldsymbol{\lambda}_T - \boldsymbol{\lambda}'_T$. By doing so, a clear relation can be established between the loss divergence and the hyperparameter divergence, which leads to a transfer from uniform argument stability to uniform stability.

*Remark.* For now, we have characterized the stability error as an upper bound on the generalization error. Let us now examine how this stability-based generalization bound informs the allocation of data between the validation and training sets. Suppose we have a total of $N$ data points, with $m = aN$ assigned to the validation set $S^{\text{val}}$ and $n = (1-a)N$ assigned to the training set $S^{\text{tr}}$, where $a \in (0, 1)$. The expected population risk can be decomposed into the generalization error and the empirical validation risk as follows:

$$\mathbb{E}_{\mathcal{A},S^{\text{val}},S^{\text{tr}},\boldsymbol{z}^{\text{test}}} \left[\mathcal{L}(\mathcal{A}(S^{\text{val}}, S^{\text{tr}}); \boldsymbol{z}^{\text{test}})\right] = \underbrace{\epsilon_{\text{gen}}}_{(I)} + \underbrace{\mathbb{E}_{\mathcal{A},S^{\text{val}},S^{\text{tr}}} \left[\frac{1}{m}\sum_{i=1}^{m} \mathcal{L}(\mathcal{A}(S^{\text{val}}, S^{\text{tr}}); \boldsymbol{z}_i^{\text{val}})\right]}_{(II)}.$$

On one hand, the generalization bound $\epsilon_{gen} \leq \epsilon_{stab} = \Theta(1/aN)$ (as in Eq. (7)) suggests that $a$ should be sufficiently large to keep term (I) small. On the other hand, $a$ should also be sufficiently small to get a low validation risk in term (II), since a larger training set generally improves validation performance. Thus, selecting $a$ involves a trade-off to optimize the overall population risk.

---

[7]For instance, $1 + x \leq e^x$ is used in [19].

# 6 Conclusion and discussion

This paper establishes novel lower bounds of the uniform stability for various HO algorithms and shows the existing upper bound in UD-based algorithms is tight. This result indicates that the notion of uniform stability has reached its limit in stability analysis for the UD-based algorithm. The lower-bounded expansion properties proposed in this paper can serve as general tools for analyzing lower bounds of stability. This paper applies them to both single-level and bilevel optimization. We also discuss in detail potential extensions of our analysis framework on establishing average stability lower bounds and generalization lower bounds in Appendix H.

**Limitations and social impacts.** This paper is constrained in the scope of smooth loss functions, while non-smooth scenarios [31] remain open. Moreover, a uniform stability lower bound does not directly imply a generalization lower bound. This gap exists as algorithmic stability is inherently introduced as a theoretical tool for analyzing the generalization upper bound. Alternative approaches might include directly deriving a generalization lower bound with examples considering the data distribution. This paper is a purely theoretical work, we have not identified any direct, significant societal impacts that must be emphasized.

## Acknowledgments and Disclosure of Funding

This work was supported by Beijing Natural Science Foundation (L247030); NSF of China (Nos. 62076145, 62206159); Beijing Nova Program (No. 20230484416); Major Innovation & Planning Interdisciplinary Platform for the "Double-First Class" Initiative, Renmin University of China; the Fundamental Research Funds for the Central Universities, and the Research Funds of Renmin University of China (22XNKJ13); the Natural Science Foundation of Shandong Province (Nos. ZR2022QF117), the Fundamental Research Funds of Shandong University; and the Ant Group Research Fund. The work was partially done at the Engineering Research Center of Next-Generation Intelligent Search and Recommendation, Ministry of Education. G. Wu was also sponsored by the TaiShan Scholars Program.

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

# A  Overview of main results

## A.1  Overview of main contributions

Table 1: Overview of main contributions. The results presented here are derived without the bounded loss assumption for fair comparison. Deformed bounds are derived under decreasing step size $\alpha_t \leq c/t$ where $c > 0$ is a constant.

| | Our contributions | Comparable results |
|---|---|---|
| **Expansion properties** | $\sigma$-divergent, $\rho$-growing (Ours) | $\sigma$-bounded, $\eta$-expansive [19] |
| **UD-based algorithm** | Recursive lower bound: $\sum_{t=1}^{T}\prod_{s=t+1}^{T}(1+\alpha_s(1-1/m)\gamma')\frac{2\alpha_t L'}{m}$, where $\gamma' = \gamma \asymp (1+\eta\gamma^{tr})^{2K}$, $L' \asymp (1+\eta\gamma^{tr})^{K}$ (Ours) | Recursive upper bound: $\sum_{t=1}^{T}\prod_{s=t+1}^{T}(1+\alpha_s(1-1/m)\gamma)\frac{2\alpha_t L}{m}$, where $\gamma \lesssim (1+\eta\gamma^{tr})^{2K}$, $L \lesssim (1+\eta\gamma^{tr})^{K}$ [17] |
| | Deformed lower bound: $\underset{\sim}{\gtrsim} \frac{T^{\ln(1+(1-1/m)c\gamma')}}{m}$, where $\gamma' = \gamma \asymp (1+\eta\gamma^{tr})^{2K}$ (Ours) | Deformed upper bound: $\underset{\sim}{\lesssim} \frac{T^{(1-1/m)c\gamma}}{m}$, where $\gamma \lesssim (1+\eta\gamma^{tr})^{2K}$ (Ours) |
| **IFT-based algorithm** | Recursive lower bound: $\sum_{t=1}^{T}\prod_{s=t+1}^{T}(1+\alpha_s(1-1/m)\gamma')\frac{2\alpha_t L'}{m}$, where $\gamma' = \gamma \asymp (1+\eta\gamma^{tr})^{2K}$, $L' \asymp (1+\eta\gamma^{tr})^{K}$ (Ours) | Recursive upper bound: $\sum_{t=1}^{T}\prod_{s=t+1}^{T}(1+\alpha_s(1-1/m)\gamma)\frac{2\alpha_t L}{m}$, where $\gamma \lesssim K(1+\eta\gamma^{tr})^{2K}$, $L \lesssim (1+\eta\gamma^{tr})^{K}$ (Ours) |
| | Deformed lower bound $\underset{\sim}{\lesssim} \frac{T^{\ln(1+(1-1/m)c\gamma')}}{m}$, where $\gamma' \gtrsim (1+\eta\gamma^{tr})^{2K}$ (Ours) | Deformed upper bound $\underset{\sim}{\lesssim} \frac{T^{(1-1/m)c\gamma}}{m}$, where $\gamma \lesssim K(1+\eta\gamma^{tr})^{2K}$ (Ours) |

## A.2 Dependent graph of main results

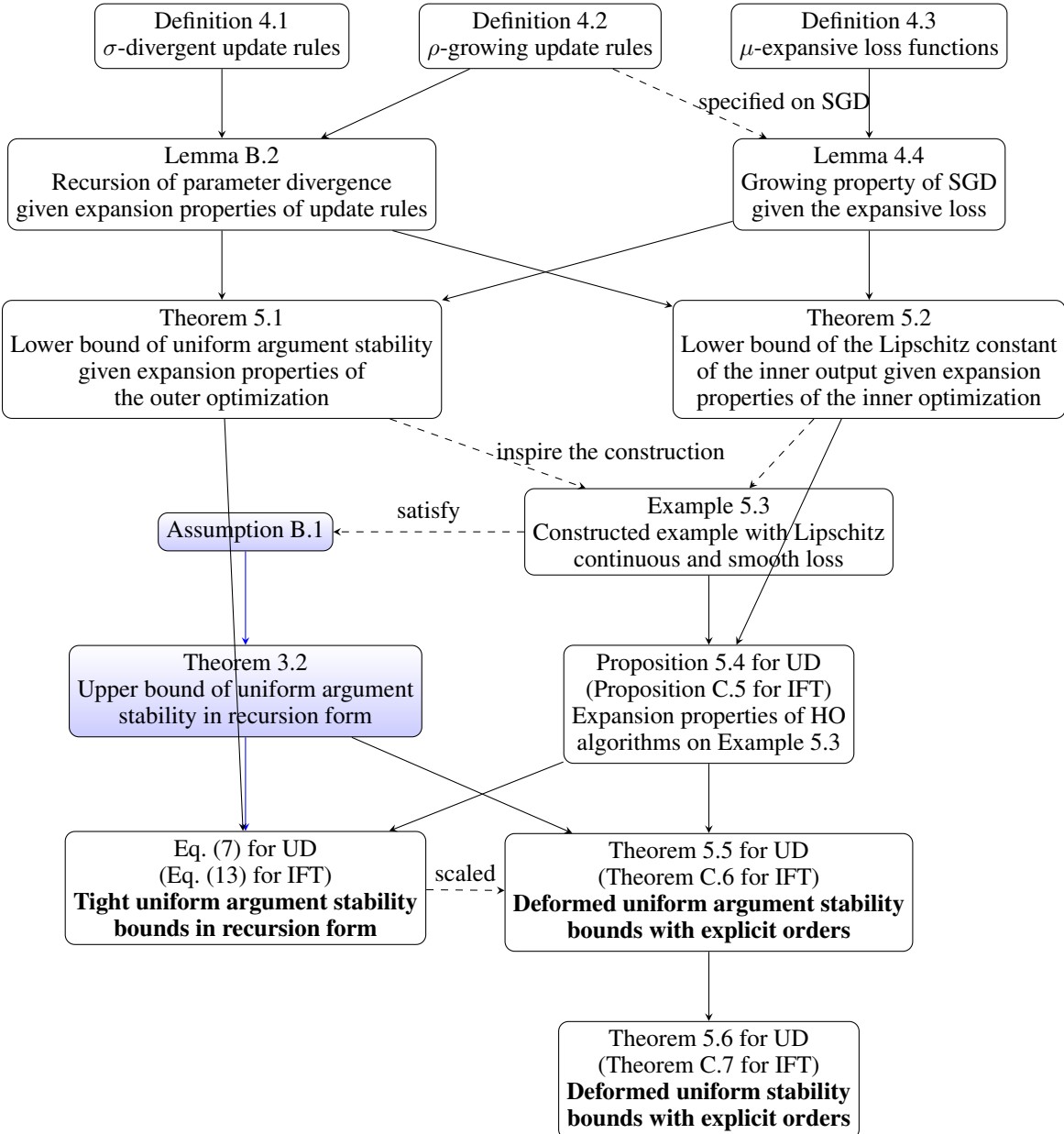

Figure 3: Dependent graph of our main results. The blue node denotes previous results and others are our contributions. The solid line represents direct proof dependency. The dashed line is annotated with text therein.

# B  Proofs of main theoretical results

## B.1  General assumptions

We first list some assumptions in the derivation for upper bounds [17], which are common theoretical conditions for an HO problem. We follow these assumptions throughout Section 5. The constructed Example 5.3 also satisfies these assumptions.

**Assumption B.1.** Let $\Omega$ be an open set including $\Lambda \times \Theta \times \mathcal{Z}$, we assume that

1. $\Lambda$ and $\Theta$ are compact and convex with non-empty interiors, and $\mathcal{Z}$ is compact,

2. $\ell^{\mathrm{val}}(\boldsymbol{\lambda}, \boldsymbol{\theta}; \boldsymbol{z}) \in C^2(\Omega)$, that is, $\ell^{\mathrm{val}}$ is second order continuously differentiable on $\Omega$,

3. $\ell^{\mathrm{tr}}(\boldsymbol{\lambda}, \boldsymbol{\theta}; \boldsymbol{z}) \in C^3(\Omega)$, that is, $\varphi_i$ is third order continuously differentiable on $\Omega$,

4. $\ell^{\mathrm{tr}}(\boldsymbol{\lambda}, \boldsymbol{\theta}; \boldsymbol{z})$ is $\gamma^{\mathrm{tr}}$-smooth as a function of $\boldsymbol{\theta}$ for all $z \in Z$ and $\lambda \in \Lambda$ (the first and third points imply such a constant $\gamma^{\mathrm{tr}}$ exists).

## B.2 Proof of Lemma 4.4

Lemma 4.4: Assume $\ell$ is $\mu$-expansive on $\boldsymbol{v}$ and $1 + \alpha\mu \geq 0$, then $G_{\ell,\alpha}$ is $(1 + \alpha\mu)$-growing on $\boldsymbol{v}$.

*Proof.* Recalling Definition 4.3, for any $\boldsymbol{w} - \boldsymbol{w}'$ colinear with $\boldsymbol{v}$ we have

$$
\begin{aligned}
& G_{\ell,\alpha}(\boldsymbol{w}) - G_{\ell,\alpha}(\boldsymbol{w}') \\
={} & \boldsymbol{w} - \boldsymbol{w}' - \alpha\big(\nabla\ell(\boldsymbol{w}) - \nabla\ell(\boldsymbol{w}')\big) \\
={} & \boldsymbol{w} - \boldsymbol{w}' + \alpha a(\boldsymbol{w} - \boldsymbol{w}') \\
={} & (1 + \alpha\mu_{\boldsymbol{w},\boldsymbol{w}'})(\boldsymbol{w} - \boldsymbol{w}'),
\end{aligned}
$$

where $\mu_{\boldsymbol{w},\boldsymbol{w}'} \geq \mu$ and thus $1 + \alpha\mu_{\boldsymbol{w},\boldsymbol{w}'} \geq 1 + \alpha\mu \geq 0$ by assumption. Therefore, we have $G_{\ell,\alpha}(\boldsymbol{w}) - G_{\ell,\alpha}(\boldsymbol{w}') \overset{\circ}{=} \boldsymbol{w} - \boldsymbol{w}'$ and $\|G_{\ell,\alpha}(\boldsymbol{w}) - G_{\ell,\alpha}(\boldsymbol{w}')\| \geq (1 + \alpha\mu)\|\boldsymbol{w} - \boldsymbol{w}'\|$, which implies $G_{\ell,\alpha}$ is $(1 + \alpha\mu)$-growing on $\boldsymbol{v}$ according to Definition 4.2. $\qquad\square$

## B.3 Proof of Theorem 5.1

As the divergence of each step is entwined with prior results and shapes subsequent evolution, we first provide the following recursion for the parameter distance using the expansion properties.

**Lemma B.2** (Recursion of parameter divergence). *Let the initial points be $\boldsymbol{w}_0 = \boldsymbol{w}_0' \in \Omega$. Suppose there exists a nonzero vector $\boldsymbol{v}$ along which, for all $1 \leq t \leq T$, $G_t \neq G_t'$ are $\sigma$-divergent and $G_t$ are $\rho$-growing. Then we have $\boldsymbol{w}_t - \boldsymbol{w}_t' \overset{\circ}{=} \boldsymbol{v}$ for all $1 \leq t \leq T$ and recursively,*

$$
\|\boldsymbol{w}_0 - \boldsymbol{w}_0'\| = 0, \|\boldsymbol{w}_t - \boldsymbol{w}_t'\| \geq \begin{cases} \rho\|\boldsymbol{w}_{t-1} - \boldsymbol{w}_{t-1}'\| + \sigma, & G_t \neq G_t', \\ \rho\|\boldsymbol{w}_{t-1} - \boldsymbol{w}_{t-1}'\|, & G_t = G_t', \end{cases} \quad t \geq 1.
$$

*Proof.* Without loss of generality, assume $\boldsymbol{v}_1$ is a unit vector (i.e., $\|\boldsymbol{v}_1\| = 1$). At the initial point, we have $\boldsymbol{w}_0 - \boldsymbol{w}_0' = \boldsymbol{0} \overset{\circ}{=} \boldsymbol{v}$. According to Definition 4.1 and Definition 4.2, if $\boldsymbol{w}_{t-1} - \boldsymbol{w}_{t-1}' \overset{\circ}{=} \boldsymbol{v}$, then

$$
\begin{aligned}
\boldsymbol{w}_t - \boldsymbol{w}_t' ={} & G_t(\boldsymbol{w}_{t-1}) - G_t'(\boldsymbol{w}_{t-1}') \\
={} & \begin{cases} G_t(\boldsymbol{w}_{t-1}) - G_t(\boldsymbol{w}_{t-1}') + G_t(\boldsymbol{w}_{t-1}') - G_t'(\boldsymbol{w}_{t-1}') & G_t \neq G_t' \\ G_t(\boldsymbol{w}_{t-1}) - G_t(\boldsymbol{w}_{t-1}') & G_t = G_t' \end{cases} \\
={} & \begin{cases} \rho_t(\boldsymbol{w}_{t-1} - \boldsymbol{w}_{t-1}') + \sigma_t \boldsymbol{v}, & G_t \neq G_t', \\ \rho_t(\boldsymbol{w}_{t-1} - \boldsymbol{w}_{t-1}'), & G_t = G_t', \end{cases} \\
={} & \begin{cases} (\rho_t\|\boldsymbol{w}_{t-1} - \boldsymbol{w}_{t-1}'\| + \sigma_t)\boldsymbol{v}, & G_t \neq G_t', \\ \rho_t\|\boldsymbol{w}_{t-1} - \boldsymbol{w}_{t-1}'\|\boldsymbol{v}, & G_t = G_t', \end{cases}
\end{aligned}
$$

where $\rho_t \geq \rho$ and $\sigma_t \geq \sigma$. Thus, we have the above recurrence relation for parameter distance, and all subsequent parameter divergence will be in the direction of $\boldsymbol{v}$. $\qquad\square$

Now we are prepared to prove Theorem 5.1: Suppose there exists a nonzero vector $\boldsymbol{v}$ along which $G_{\boldsymbol{z}_i,\alpha_t}$ and $G_{\tilde{\boldsymbol{z}}_i,\alpha_t}$ are $2\alpha_t L'$-divergent and $\mathcal{L}(\cdot; \boldsymbol{z})$ is $\gamma'$-expansive for all $\boldsymbol{z} \in S^{\mathrm{val}}$. Then we have

$$
\epsilon_{\mathrm{arg}} \geq \sum_{t=1}^{T} \prod_{s=t+1}^{T} \left(1 + \alpha_s(1 - 1/m)\gamma'\right)\frac{2\alpha_t L'}{m}.
$$

*Proof.* Using Lemma 4.4, we have $G_{\boldsymbol{z}_{i_t},\alpha_t}$ and $G_{\tilde{\boldsymbol{z}}_{i_t},\alpha_t}$ are $(1 + \alpha_t\gamma')$-growing for all $1 \leq t \leq T$. Denote $\delta_t = \|\boldsymbol{\lambda}_t - \tilde{\boldsymbol{\lambda}}_t\|$ for each step $t$. As Algorithm 1 is initialized with the same starting point,

we know that $\boldsymbol{\lambda}_0 = \tilde{\boldsymbol{\lambda}}_0$ and thus $\delta_0 = 0$. For all $1 \le t \le T$, there is a probability of $1 - \frac{1}{m}$ to select the same examples and $\frac{1}{m}$ otherwise. Consequently, by the law of total probability, we have the recurrence relation

$$\mathbb{E}_{\mathcal{A}}[\delta_t] \ge \left(1 - \frac{1}{m}\right) \mathbb{E}_{\mathcal{A}}[(1 + \alpha_t \gamma')\delta_{t-1}] + \frac{1}{m}\mathbb{E}_{\mathcal{A}}[\delta_{t-1} + 2\alpha_t L'] \quad \text{(Lemma B.2, the law of total probability)}$$

$$= \left[1 + \alpha_t(1 - \frac{1}{m})\gamma'\right]\mathbb{E}_{\mathcal{A}}[\delta_{t-1}] + \frac{2\alpha_t L'}{m} \qquad\qquad \text{(linearity of expectation),}$$

By unwinding the recurrence from $T$ to 1, for all $S^{\mathrm{val}} \simeq \tilde{S}^{\mathrm{val}} \in \mathcal{Z}^m, S^{\mathrm{tr}} \in \mathcal{Z}^n$ we have

$$\mathbb{E}_{\mathcal{A}}[\delta_T] \ge \sum_{t=1}^{T} \prod_{s=t+1}^{T} \left(1 + \alpha_s(1 - 1/m)\gamma'\right)\frac{2\alpha_t L'}{m}, \qquad\qquad (8)$$

which implies

$$\epsilon_{\mathrm{arg}} \ge \sum_{t=1}^{T} \prod_{s=t+1}^{T} \left(1 + \alpha_s(1 - 1/m)\gamma'\right)\frac{2\alpha_t L'}{m}.$$

$\square$

## B.4   Proof of Theorem 5.2

Here we prove a more general version of Theorem 5.2 in the main paper by additionally considering the cases where $\mu^{\mathrm{tr}} \le 0$. Theorem 5.2 in the main paper can be simply derived by letting $\mu^{\mathrm{tr}} > 0$.

**Theorem B.3** (Lower bound of Lipschitz of the inner-level solution, generalized Theorem 5.2). *Given any two hyperparameters $\boldsymbol{\lambda}, \boldsymbol{\lambda}' \in \Lambda$, suppose there exists a nonzero vector $\boldsymbol{v}$ along which $G_{\boldsymbol{\lambda},\eta}$ and $G_{\boldsymbol{\lambda}',\eta}$ are $\|\boldsymbol{\lambda} - \boldsymbol{\lambda}'\|\sigma^{\mathrm{tr}}$-divergent and $\ell^{\mathrm{tr}}(\boldsymbol{\lambda}, \cdot; \boldsymbol{z})$ is $\mu^{\mathrm{tr}}$-expansive for all $\boldsymbol{z} \in S^{\mathrm{tr}}$ where $1 + \eta\mu^{\mathrm{tr}} \ge 0$. Then we have*

$$L^{\boldsymbol{\theta}_K} = \begin{cases} \frac{\sigma^{\mathrm{tr}}}{\eta\mu^{\mathrm{tr}}}[(1 + \eta\mu^{\mathrm{tr}})^K - 1], & \mu^{\mathrm{tr}} > 0, \\ \sigma^{\mathrm{tr}}K, & \mu^{\mathrm{tr}} = 0, \\ \frac{\sigma^{\mathrm{tr}}}{\eta|\mu^{\mathrm{tr}}|}[1 - (1 - \eta|\mu^{\mathrm{tr}}|)^K], & \mu^{\mathrm{tr}} < 0. \end{cases}$$

*Omitting constants that depend on $\eta$, $\sigma^{\mathrm{tr}}$, and $\mu^{\mathrm{tr}}$, we get $L^{\boldsymbol{\theta}_K} \gtrsim \begin{cases} (1 + \eta\mu^{\mathrm{tr}})^K, & \mu^{\mathrm{tr}} > 0, \\ K, & \mu^{\mathrm{tr}} = 0, \\ 1, & \mu^{\mathrm{tr}} < 0. \end{cases}$*

*Proof.* First, for any $\boldsymbol{\lambda}$ and $\boldsymbol{\lambda}'$, we establish a lower bound of $\|\boldsymbol{\theta}_K(\boldsymbol{\lambda}) - \boldsymbol{\theta}_K(\boldsymbol{\lambda}')\|$ in a recursion way. Using Lemma 4.4, we have $G_{\boldsymbol{\lambda},\eta}$ and $G_{\boldsymbol{\lambda}',\eta}$ are $(1 + \eta\mu^{\mathrm{tr}})$-growing. For any inner step $1 \le k \le K$, we have

$$\|\boldsymbol{\theta}_k(\boldsymbol{\lambda}) - \boldsymbol{\theta}_k(\boldsymbol{\lambda}')\| = \|G_{\boldsymbol{\lambda},\eta}(\boldsymbol{\theta}_{k-1}(\boldsymbol{\lambda})) - G_{\boldsymbol{\lambda}',\eta}(\boldsymbol{\theta}_{k-1}(\boldsymbol{\lambda}'))\|$$

$$= \|G_{\boldsymbol{\lambda},\eta}(\boldsymbol{\theta}_{k-1}(\boldsymbol{\lambda})) - G_{\boldsymbol{\lambda},\eta}(\boldsymbol{\theta}_{k-1}(\boldsymbol{\lambda}')) + G_{\boldsymbol{\lambda},\eta}(\boldsymbol{\theta}_{k-1}(\boldsymbol{\lambda}')) - G_{\boldsymbol{\lambda}',\eta}(\boldsymbol{\theta}_{k-1}(\boldsymbol{\lambda}'))\|$$

$$\ge \left|(1 + \eta\mu^{\mathrm{tr}})\|\boldsymbol{\theta}_{k-1}(\boldsymbol{\lambda}) - \boldsymbol{\theta}_{k-1}(\boldsymbol{\lambda}')\| + \sigma^{\mathrm{tr}}\|\boldsymbol{\lambda} - \boldsymbol{\lambda}'\|\|\boldsymbol{v}\| \right| \qquad \text{(Lemma B.2)}$$

$$= \left|(1 + \eta\mu^{\mathrm{tr}})\|\boldsymbol{\theta}_{k-1}(\boldsymbol{\lambda}) - \boldsymbol{\theta}_{k-1}(\boldsymbol{\lambda}')\| + \sigma^{\mathrm{tr}}\|\boldsymbol{\lambda} - \boldsymbol{\lambda}'\| \right| \qquad (\|\boldsymbol{v}\| = 1)$$

$$= (1 + \eta\mu^{\mathrm{tr}})\|\boldsymbol{\theta}_{k-1}(\boldsymbol{\lambda}) - \boldsymbol{\theta}_{k-1}(\boldsymbol{\lambda}')\| + \sigma^{\mathrm{tr}}\|\boldsymbol{\lambda} - \boldsymbol{\lambda}'\|. \qquad (1 + \eta\mu^{\mathrm{tr}} \ge 0)$$

Using the fact that the algorithm is initialized with the same starting point and unwinding the above recurrence from $K$ to 1 , we obtain

$$\|\boldsymbol{\theta}_K(\boldsymbol{\lambda}) - \boldsymbol{\theta}_K(\boldsymbol{\lambda}')\| \ge \sum_{k=0}^{K-1} (1 + \eta\mu^{\mathrm{tr}})^k \sigma^{\mathrm{tr}}\|\boldsymbol{\lambda} - \boldsymbol{\lambda}'\|,$$

which implies that for any $\boldsymbol{\lambda} \in \Lambda$. According to the mean value theorem for vector valued multivariable function, there exists a $\boldsymbol{c}$ on line segment determined by $\boldsymbol{\lambda}$ and $\boldsymbol{\lambda}'$ such that $\|\nabla\boldsymbol{\theta}_K(\boldsymbol{c})(\boldsymbol{\lambda} - \boldsymbol{\lambda}')\| \ge$

$\|\boldsymbol{\theta}_K(\boldsymbol{\lambda}) - \boldsymbol{\theta}_K(\boldsymbol{\lambda}')\|$, and for triangle inequality, we have $\|\nabla\boldsymbol{\theta}_K(\boldsymbol{c})\|\|\boldsymbol{\lambda} - \boldsymbol{\lambda}'\| \geq \|\nabla\boldsymbol{\theta}_K(\boldsymbol{c})(\boldsymbol{\lambda} - \boldsymbol{\lambda}')\|$. Therefore, by the definition of Lipschitz continuity, it holds that

$$L^{\boldsymbol{\theta}_K} \geq \|\nabla\boldsymbol{\theta}_K(\boldsymbol{c})\| \geq \frac{\|\boldsymbol{\theta}_K(\boldsymbol{\lambda}) - \boldsymbol{\theta}_K(\boldsymbol{\lambda}')\|}{\|\boldsymbol{\lambda} - \boldsymbol{\lambda}'\|} \geq \sigma^{\text{tr}}\sum_{k=0}^{K-1}(1 + \eta\mu^{\text{tr}})^k = \begin{cases} \sigma^{\text{tr}}\frac{(1+\eta\mu^{\text{tr}})^K - 1}{\eta\mu^{\text{tr}}}, & \mu^{\text{tr}} > 0, \\ \sigma^{\text{tr}}K, & \mu^{\text{tr}} = 0, \\ \sigma^{\text{tr}}\frac{1-(1-\eta|\mu^{\text{tr}}|)^K}{\eta|\mu^{\text{tr}}|}, & \mu^{\text{tr}} < 0, \end{cases}$$

which completes the proof. $\qquad\square$

### B.5 Proof of Proposition 5.4

Before deriving Proposition 5.4, we present a technical lemma as follows.

**Lemma B.4.** *Suppose that $\boldsymbol{A} \in \mathbb{R}^{d\times d}$ is a symmetric matrix. We denote $\boldsymbol{v}_1, \ldots, \boldsymbol{v}_d$ the orthogonal unit eigenvectors of $\boldsymbol{A}$ and $\gamma_1 \leq \cdots \leq \gamma_d$ the corresponding eigenvalues, where we assume that $1 - \eta\gamma_d \geq 0$. Then it holds that*

$$\left\| \eta\sum_{k=0}^{K-1}(\boldsymbol{I} - \eta\boldsymbol{A})^k\left(2\boldsymbol{I} - \eta\boldsymbol{A}\sum_{k=0}^{K-1}(\boldsymbol{I} - \eta\boldsymbol{A})^k\right) \right\| = \eta\sum_{k=0}^{K-1}(1 - \eta\gamma_1)^k\left(2 - \eta\gamma_1\sum_{k=0}^{K-1}(1 - \eta\gamma_1)^k\right)$$

$$\asymp \begin{cases} (1 - \eta\gamma_1)^{2K}, & \gamma_1 < 0, \\ K, & \gamma_1 = 0, \\ 1, & \gamma_1 > 0. \end{cases}$$

*Proof.* For simplicity, we denote $\eta\sum_{k=0}^{K-1}(\boldsymbol{I} - \eta\boldsymbol{A})^k\left(2\boldsymbol{I} - \eta\boldsymbol{A}\sum_{k=0}^{K-1}(\boldsymbol{I} - \eta\boldsymbol{A})^k\right)$ by $\boldsymbol{C}$, then $\boldsymbol{C}$ is symmetric and has eigenvectors $\boldsymbol{v}_1, \ldots, \boldsymbol{v}_d$ as well. Based on the symmetric of $\boldsymbol{C}$, $\|\boldsymbol{C}\|$ equals to its maximum absolute eigenvalue, which can be expressed as

$$\|\boldsymbol{C}\| = \sup_i\|\|\boldsymbol{C}\boldsymbol{v}_i\|\|$$

$$= \sup_i\left| \eta\sum_{k=0}^{K-1}(1 - \eta\gamma_i)^k\left(2 - \eta\gamma_i\sum_{k=0}^{K-1}(1 - \eta\gamma_i)^k\right) \right|$$

$$= \sup_i \begin{cases} 2\eta K & \gamma_i = 0, \\ \left|\eta\sum_{k=0}^{K-1}(1 - \eta\gamma_i)^k\left(2 - \eta\gamma_i\frac{1-(1-\eta\gamma_i)^K}{1-(1-\eta\gamma_i)}\right)\right| & \gamma_i \neq 0 \end{cases}$$

$$= \sup_i \begin{cases} 2\eta K & \gamma_i = 0, \\ \eta\sum_{k=0}^{K-1}(1 - \eta\gamma_i)^k\left(1 + (1 - \eta\gamma_i)^K\right) & \gamma_i \neq 0. \end{cases}$$

The last equation for $\gamma_i \neq 0$ holds for $1 - \eta\gamma_i \geq 1 - \eta\gamma_d \geq 0$.

We define $h(\gamma) := \eta\sum_{k=0}^{K-1}(1 - \eta\gamma)^k\left(1 + (1 - \eta\gamma)^K\right)$, which is decreasing on $(-\infty, \gamma_d]$, achieving the maximum at the smallest eigenvalue $\gamma_1$. Therefore,

$$\|\boldsymbol{C}\| = \eta\sum_{k=0}^{K-1}(1 - \eta\gamma_1)^k\left(2 - \eta\gamma_1\sum_{k=0}^{K-1}(1 - \eta\gamma_1)^k\right) \asymp \begin{cases} (1 - \eta\gamma_1)^{2K}, & \gamma_1 < 0, \\ K, & \gamma_1 = 0, \\ 1, & \gamma_1 > 0. \end{cases}$$

$\qquad\square$

Now, we are ready to prove Proposition 5.4. Here we prove a more general version of Proposition 5.4 in the main paper by additionally considering the cases where $\gamma_1 \geq 0$. Proposition 5.4 in the main paper can be simply derived by letting $\gamma_1 < 0$.

**Proposition B.5** (Expansion properties of UD-based algorithms, generalized Proposition 5.4). *Suppose we solve Example 5.3 by UD-based Algorithm 1 with constant inner step size $\eta$ where $1 - \eta\gamma_d \geq 0$*

*and outer step size $\alpha_t$. Then (1) the outer update rules $G_{\boldsymbol{z}_i,\alpha_t}$ and $G_{\tilde{\boldsymbol{z}}_i,\alpha_t}$ are $2\alpha_t L'$- divergent along $\boldsymbol{v}_1$, and (2) the composite validation loss $\mathcal{L}(\cdot;\boldsymbol{z})$ is $\gamma'$-expansive along $\boldsymbol{v}_1$ for all $\boldsymbol{z} \in S^{\mathrm{val}}$, where*

$$L \asymp L' \asymp \begin{cases} (1+\eta\gamma^{\mathrm{tr}})^{2K}, & \gamma_1 < 0, \\ K, & \gamma_1 = 0, \\ 1, & \gamma_1 > 0, \end{cases} \quad and \quad \gamma = \gamma' \asymp \begin{cases} (1+\eta\gamma^{\mathrm{tr}})^{2K}, & \gamma_1 < 0, \\ K, & \gamma_1 = 0, \\ 1, & \gamma_1 > 0. \end{cases}$$

*Proof.* As Example 5.3 satisfies Assumption B.1, we have $L' \leq L \lesssim (1+\eta\gamma^{\mathrm{tr}})^K$ by Theorem 3 in [17]. We are going to verify that $L' \gtrsim (1+\eta\gamma^{\mathrm{tr}})^K$ and $\gamma = \gamma' \gtrsim (1+\eta\gamma^{\mathrm{tr}})^{2K}$ in the following.

Given a hyperparameter $\boldsymbol{\lambda}$, a constant step size $\eta$ and a initial point $\boldsymbol{\theta}_0$, at each step $1 \leq k \leq K$, we have an inner update

$$\begin{aligned} G_{\boldsymbol{\lambda},\eta}(\boldsymbol{\theta}_{k-1}) &= \boldsymbol{\theta}_{k-1} - \eta\nabla_{\boldsymbol{\theta}}\ell^{\mathrm{tr}}(\boldsymbol{\lambda},\boldsymbol{\theta}_{k-1};\boldsymbol{z}_{j_k}^{\mathrm{tr}}) \\ &= \boldsymbol{\theta}_{k-1} - \eta\nabla_{\boldsymbol{\theta}}\left[\frac{1}{2}\boldsymbol{\theta}_{k-1}^\top \boldsymbol{A}\boldsymbol{\theta}_{k-1} + \boldsymbol{\lambda}^\top\boldsymbol{\theta}_{k-1} - y_{j_k}^{\mathrm{tr}}\boldsymbol{x}_{j_k}^{\mathrm{tr}\top}\boldsymbol{\theta}_{k-1}\right] \\ &= \boldsymbol{\theta}_{k-1} - \eta(\boldsymbol{A}\boldsymbol{\theta}_{k-1} + \boldsymbol{\lambda} - y_{j_k}^{\mathrm{tr}}\boldsymbol{x}_{j_k}^{\mathrm{tr}}) \\ &= (\boldsymbol{I} - \eta\boldsymbol{A})\boldsymbol{\theta}_{k-1} - \eta(\boldsymbol{\lambda} - y_{j_k}^{\mathrm{tr}}\boldsymbol{x}_{j_k}^{\mathrm{tr}}), \end{aligned}$$

where $j_k$ is uniformly sampled from $[n]$. Recursively, we get

$$\begin{aligned} \boldsymbol{\theta}_K(\boldsymbol{\lambda}) &= G_{\boldsymbol{\lambda},\eta}\Big(G_{\boldsymbol{\lambda},\eta}(\ldots G_{\boldsymbol{\lambda},\eta}(\boldsymbol{\theta}_0))\Big) \\ &= (\boldsymbol{I} - \eta\boldsymbol{A})^K\boldsymbol{\theta}_0 - \eta\sum_{k=0}^{K-1}(\boldsymbol{I} - \eta\boldsymbol{A})^k(\boldsymbol{\lambda} - y_{j_k}^{\mathrm{tr}}\boldsymbol{x}_{j_k}^{\mathrm{tr}}), \end{aligned}$$

so that

$$\nabla_{\boldsymbol{\lambda}}\boldsymbol{\theta}_K(\boldsymbol{\lambda}) = -\eta\sum_{k=0}^{K-1}(\boldsymbol{I} - \eta\boldsymbol{A})^k. \tag{9}$$

As $L'$ and $\gamma'$ describe the expansion properties of SGD on $\mathcal{L}$, we first investigate the gradient of the compound validation loss

$$\nabla\mathcal{L}(\boldsymbol{\lambda};\boldsymbol{z}) = \boldsymbol{\theta}_K(\boldsymbol{\lambda}) + \nabla_{\boldsymbol{\lambda}}\boldsymbol{\theta}_K(\boldsymbol{\lambda})^\top\big(\boldsymbol{A}\boldsymbol{\theta}_K(\boldsymbol{\lambda}) + \boldsymbol{\lambda} - y\boldsymbol{x}\big).$$

For all $\boldsymbol{\lambda} \in \Lambda$, the outer update divergence when SGD picks the distinct examples is

$$\begin{aligned} G_{\boldsymbol{z}_i,\alpha_t}(\boldsymbol{\lambda}) - G_{\tilde{\boldsymbol{z}}_i,\alpha_t}(\boldsymbol{\lambda}) &= \alpha_t\big(\nabla\mathcal{L}(\boldsymbol{\lambda};\boldsymbol{z}_i) - \nabla\mathcal{L}(\boldsymbol{\lambda};\tilde{\boldsymbol{z}}_i)\big) \\ &= \alpha_t\nabla_{\boldsymbol{\lambda}}\boldsymbol{\theta}_K(\boldsymbol{\lambda})^\top\big(-y_i\boldsymbol{x}_i - (-\tilde{y}_i\tilde{\boldsymbol{x}}_i)\big) \\ &= -2\alpha_t\nabla_{\boldsymbol{\lambda}}\boldsymbol{\theta}_K(\boldsymbol{\lambda})^\top\boldsymbol{v}_1 \qquad (y_i = \tilde{y}_i = 1, \boldsymbol{x}_i = \boldsymbol{v}_1, \tilde{\boldsymbol{x}}_i = -\boldsymbol{v}_1) \\ &= 2\alpha_t\eta\sum_{k=0}^{K-1}(\boldsymbol{I} - \eta\boldsymbol{A})^k\boldsymbol{v}_1 \qquad (\boldsymbol{A} \text{ is symmetric}) \\ &= 2\alpha_t\eta\sum_{k=0}^{K-1}(1 - \eta\gamma_1)^k\boldsymbol{v}_1 \qquad (\boldsymbol{A}\boldsymbol{v}_1 = \gamma_1\boldsymbol{v}_1) \\ &= 2\alpha_t\eta\sum_{k=0}^{K-1}(1 + \eta\gamma^{\mathrm{tr}})^k\boldsymbol{v}_1. \qquad (\gamma^{\mathrm{tr}} = |\gamma_1| = -\gamma_1) \end{aligned}$$

Recalling Definition 4.1, we have $G_{\boldsymbol{z}_i,\alpha_t}$ and $G_{\boldsymbol{z}_i',\alpha_t}$ are $\alpha_t L'$-divergent along $\boldsymbol{v}_1$, where

$$L' = \eta\sum_{k=0}^{K-1}(1 + \eta\gamma^{\mathrm{tr}})^k = \begin{cases} \big[(1+\eta\gamma^{\mathrm{tr}})^K - 1\big]/\gamma^{\mathrm{tr}} \asymp (1+\eta\gamma^{\mathrm{tr}})^K, & \gamma_1 < 0, \\ \eta K \asymp K, & \gamma_1 = 0, \\ \big[1 - (1+\eta\gamma^{\mathrm{tr}})^K\big]/\gamma^{\mathrm{tr}}, \asymp 1 & \gamma_1 > 0. \end{cases} \tag{10}$$

For the case that $\gamma_1 < 0$, we have $L' \asymp (1+\eta\gamma^{\mathrm{tr}})^K$.

Next, we are going to clarify that $\mathcal{L}(\boldsymbol{\lambda})$ is $\gamma'$-expansive along $\boldsymbol{v}_1$, and $\gamma'$ equals to the smooth constant $\gamma$ of $\mathcal{L}(\boldsymbol{\lambda})$. As $\mathcal{L}$ is twice differentiable, according to the definition of smoothness, we have

$$
\begin{aligned}
\gamma &= \sup_{\boldsymbol{\lambda} \in \Lambda} \|\nabla_{\boldsymbol{\lambda}}^2 \mathcal{L}(\boldsymbol{\lambda}; \boldsymbol{z})\| \\
&= \|\nabla_{\boldsymbol{\lambda}} \boldsymbol{\theta}_K(\boldsymbol{\lambda}) + \nabla_{\boldsymbol{\lambda}} \boldsymbol{\theta}_K(\boldsymbol{\lambda})^\top (\boldsymbol{A} \nabla_{\boldsymbol{\lambda}} \boldsymbol{\theta}_K(\boldsymbol{\lambda}) + \boldsymbol{I})\| && (\nabla_{\boldsymbol{\lambda}}^2 \boldsymbol{\theta}_K(\boldsymbol{\lambda}) = \boldsymbol{0}) \\
&= \left\| -\eta \sum_{k=0}^{K-1} (\boldsymbol{I} - \eta \boldsymbol{A})^k \left( 2\boldsymbol{I} - \eta \boldsymbol{A} \sum_{k=0}^{K-1} (\boldsymbol{I} - \eta \boldsymbol{A})^k \right) \right\| \\
&= \eta \sum_{k=0}^{K-1} (1 - \eta \gamma_1)^k \left( 2 - \eta \gamma_1 \sum_{k=0}^{K-1} (1 - \eta \gamma_1)^k \right) && (\text{Lemma B.4}) \\
&= \eta \sum_{k=0}^{K-1} (1 + \eta \gamma^{\mathrm{tr}})^k \left( 2 + \eta \gamma^{\mathrm{tr}} \sum_{k=0}^{K-1} (1 + \eta \gamma^{\mathrm{tr}})^k \right) && (11) \\
&\asymp \begin{cases} (1 + \eta \gamma^{\mathrm{tr}})^{2K}, & \gamma_1 < 0, \\ K, & \gamma_1 = 0, \\ 1, & \gamma_1 > 0. \end{cases}
\end{aligned}
$$

Eq. (11) holds for $\gamma^{\mathrm{tr}} = |\gamma_1| = -\gamma_1$. For all $\boldsymbol{\lambda}, \boldsymbol{\lambda}' \in \Lambda$, such that $\boldsymbol{\lambda} - \boldsymbol{\lambda}' = a\boldsymbol{v}_1 \overset{\circ}{=} \boldsymbol{v}_1$ where $a \in \mathbb{R}_+$, we have

$$
\begin{aligned}
\nabla \mathcal{L}(\boldsymbol{\lambda}; \boldsymbol{z}) - \nabla \mathcal{L}(\boldsymbol{\lambda}'; \boldsymbol{z}) &= \boldsymbol{\theta}_K(\boldsymbol{\lambda}) - \boldsymbol{\theta}_K(\boldsymbol{\lambda}') \\
&\quad + \nabla_{\boldsymbol{\lambda}} \boldsymbol{\theta}_K(\boldsymbol{\lambda})^\top (\boldsymbol{A} \boldsymbol{\theta}_K(\boldsymbol{\lambda}) + \boldsymbol{\lambda} - y\boldsymbol{x}) - \nabla_{\boldsymbol{\lambda}'} \boldsymbol{\theta}_K(\boldsymbol{\lambda}')^\top (\boldsymbol{A} \boldsymbol{\theta}_K(\boldsymbol{\lambda}') + \boldsymbol{\lambda}' - y\boldsymbol{x}) \\
&= -\eta \sum_{k=0}^{K-1} (\boldsymbol{I} - \eta \boldsymbol{A})^k (\boldsymbol{\lambda} - \boldsymbol{\lambda}') - \eta \sum_{k=0}^{K-1} (\boldsymbol{I} - \eta \boldsymbol{A})^k \left( -\boldsymbol{A} \eta \sum_{k=0}^{K-1} (\boldsymbol{I} - \eta \boldsymbol{A})^k + \boldsymbol{I} \right) (\boldsymbol{\lambda} - \boldsymbol{\lambda}') \\
&= -\eta \sum_{k=0}^{K-1} (\boldsymbol{I} - \eta \boldsymbol{A})^k \left( 2\boldsymbol{I} - \eta \boldsymbol{A} \sum_{k=0}^{K-1} (\boldsymbol{I} - \eta \boldsymbol{A})^k \right) (\boldsymbol{\lambda} - \boldsymbol{\lambda}') \\
&= -\eta \sum_{k=0}^{K-1} (\boldsymbol{I} - \eta \boldsymbol{A})^k \left( 2\boldsymbol{I} - \eta \boldsymbol{A} \sum_{k=0}^{K-1} (\boldsymbol{I} - \eta \boldsymbol{A})^k \right) a\boldsymbol{v}_1 \\
&= -\eta \sum_{k=0}^{K-1} (1 - \eta \gamma_1)^k \left( 2 - \eta \gamma_1 \sum_{k=0}^{K-1} (1 - \eta \gamma_1)^k \right) a\boldsymbol{v}_1 \\
&= -\eta \sum_{k=0}^{K-1} (1 + \eta \gamma^{\mathrm{tr}})^k \left( 2 + \eta \gamma^{\mathrm{tr}} \sum_{k=0}^{K-1} (1 + \eta \gamma^{\mathrm{tr}})^k \right) (\boldsymbol{\lambda} - \boldsymbol{\lambda}'), \\
&:= -\gamma'(\boldsymbol{\lambda} - \boldsymbol{\lambda}').
\end{aligned}
$$

According to Definition 4.3, this implies $\mathcal{L}(\boldsymbol{\lambda})$ is $\gamma'$-expansive along $\boldsymbol{v}_1$. Therefore

$$
\gamma' = \gamma \asymp \begin{cases} (1 + \eta \gamma^{\mathrm{tr}})^{2K}, & \gamma_1 < 0, \\ K, & \gamma_1 = 0, \\ 1, & \gamma_1 > 0. \end{cases}
$$

For the case that $\gamma_1 < 0$, we obtain that $\gamma' = \gamma \asymp (1 + \eta \gamma^{\mathrm{tr}})^{2K}$.

$\square$

## B.6  Proof of Theorem 5.5

Here we prove a more general version of Theorem 5.5 in the main paper where $\gamma_1 < 0$ by additionally considering the cases where $\gamma_1 \geq 0$.

**Theorem B.6** (Uniform argument stability of UD algorithm, generalized Theorem 5.5). *Solving Example 5.3 by UD-based Algorithm 1 with constant inner step size $\eta$ where $1 - \eta\gamma_d \geq 0$ and decreasing outer step sizes $\alpha_t = c/t$ with $c$ as a positive constant has uniform argument stability that*

$$\frac{T^{\ln\left(1+(1-\frac{1}{m})c\gamma'\right)}}{m} \lesssim \epsilon_{\mathrm{arg}} \lesssim \frac{T^{(1-\frac{1}{m})c\gamma}}{m},$$

*where $\gamma = \gamma' \asymp \begin{cases} (1-\eta\gamma_1)^{2K}, & \gamma_1 < 0, \\ K, & \gamma_1 = 0, \quad \text{as in Proposition B.5.} \\ 1, & \gamma_1 > 0. \end{cases}$*

*Proof.* We first derive the left side of the result (i.e., the lower bound). The derivation is built upon the resursion form of the lower bound in Theorem 5.1 and utilizes a scaling operation that when $r = \frac{\ln\left(1+(1-1/m)c\gamma'\right)}{(1-1/m)c\gamma'}$, it holds that $1 + x \geq \exp(rx)$ for any $x \in \left\{(1-1/m)c\gamma'/t | 1 \leq t \leq T\right\}$. According to Theorem 5.1, we have

$$
\begin{aligned}
\epsilon_{\mathrm{arg}} &\geq \sum_{t=1}^{T} \prod_{s=t+1}^{T} \left(1 + \alpha_s(1-1/m)\gamma'\right)\frac{2\alpha_t L'}{m} \\
&\geq \sum_{t=1}^{T} \prod_{s=t+1}^{T} \exp\left[r\left(1-\frac{1}{m}\right)\alpha_s\gamma'\right]\frac{2\alpha_t L'}{m} \\
&= \sum_{t=1}^{T-1} \prod_{s=t+1}^{T} \exp\left[r\left(1-\frac{1}{m}\right)\frac{c\gamma'}{s}\right]\frac{2cL'}{tm} + \frac{2cL'}{Tm} && (\alpha_t = c/t) \\
&= \sum_{t=1}^{T-1} \exp\left[r\left(1-\frac{1}{m}\right)c\gamma' \sum_{s=t+1}^{T}\frac{1}{s}\right]\frac{2cL'}{tm} + \frac{2cL'}{Tm} \\
&\geq \sum_{t=1}^{T-1} \exp\left[r\left(1-\frac{1}{m}\right)c\gamma' \ln\frac{T+1}{t+1}\right]\frac{2cL'}{tm} + \frac{2cL'}{Tm} && (\forall t_2 > t_1 > 0, \sum_{t=t_1}^{t_2}\frac{1}{t} \geq \ln\frac{t_2+1}{t_1})) \\
&= \sum_{t=1}^{T} \left(\frac{T+1}{t+1}\right)^{r\left(1-1/m\right)c\gamma'}\frac{2cL'}{tm} \\
&\geq \frac{2cL'}{m}(T+1)^{r\left(1-1/m\right)c\gamma'} \sum_{t=2}^{T+1} t^{-r\left(1-1/m\right)c\gamma'-1} \\
&\geq \frac{2cL'}{m}(T+1)^{r\left(1-1/m\right)c\gamma'} \int_{2}^{T+2} t^{-r\left(1-1/m\right)c\gamma'-1}dt \\
& && (\forall a > 0, \sum_{t=2}^{T+1} t^{-a-1} \geq \int_{2}^{T+2} t^{-a-1}dt) \\
&= \frac{2cL'}{m}(T+1)^{r\left(1-1/m\right)c\gamma'} \left[\frac{2^{-r\left(1-1/m\right)c\gamma'} - (T+2)^{-r\left(1-1/m\right)c\gamma'}}{r\left(1-1/m\right)c\gamma'}\right] \\
&= \frac{2L'}{r(m-1)\gamma'}(T+1)^{r\left(1-1/m\right)c\gamma'} \left[2^{-r\left(1-1/m\right)c\gamma'} - (T+2)^{-r\left(1-1/m\right)c\gamma'}\right] \\
&\geq \frac{2cL'}{m\ln\left(1+(1-1/m)c\gamma'\right)}\left[\left(\frac{T+1}{2}\right)^{\ln\left(1+(1-1/m)c\gamma'\right)} - 1\right]. && (r = \frac{\ln\left(1+(1-1/m)c\gamma'\right)}{(1-1/m)c\gamma'})
\end{aligned}
$$

Then, we continue to derive the right side of the result (i.e., the upper bound). Based on Eq. (6), we have

$$\epsilon_{\text{arg}} \leq \sum_{t=1}^{T} \prod_{s=t+1}^{T} \left(1 + \alpha_s(1 - 1/m)\gamma\right)\frac{2\alpha_t L}{m}$$

$$\leq \sum_{t=1}^{T-1} \prod_{s=t+1}^{T} \exp\left[\left(1 - \frac{1}{m}\right)\frac{\gamma c}{s}\right]\frac{2cL}{tm} + \frac{2cL}{Tm}$$

$$= \sum_{t=1}^{T-1} \exp\left[\left(1 - \frac{1}{m}\right)\gamma c \sum_{s=t+1}^{T} \frac{1}{s}\right]\frac{2cL}{tm} + \frac{2cL}{Tm}$$

$$\leq \sum_{t=1}^{T-1} \exp\left[\left(1 - \frac{1}{m}\right)\gamma c \ln\frac{T}{t}\right]\frac{2cL}{tm} + \frac{2cL}{Tm} \qquad (\forall t_2 > t_1 > 1, \sum_{t=t_1}^{t_2}\frac{1}{t} \leq \ln\frac{t_2}{t_1-1}))$$

$$= \sum_{t=1}^{T} \left(\frac{T}{t}\right)^{(1-1/m)\gamma c}\frac{2cL}{tm}$$

$$= \frac{2cL}{m}T^{(1-1/m)\gamma c}\sum_{t=1}^{T} t^{-(1-1/m)\gamma c-1}$$

$$\leq \frac{2cL}{m}T^{(1-1/m)\gamma c}\left(1 + \int_{1}^{T} t^{-(1-1/m)\gamma c-1}dt\right)$$

$$\qquad (\forall a > 0, \sum_{t=1}^{T} t^{-a-1} \leq 1 + \int_{1}^{T} t^{-a-1}dt)$$

$$= \frac{2cL}{m(1-1/m)c\gamma}\left[\left(1 + (1-1/m)\gamma c\right)T^{(1-1/m)\gamma c} - 1\right]$$

$$= \frac{2L}{(m-1)\gamma}\left[\left(1 + (1-1/m)\gamma c\right)T^{(1-1/m)\gamma c} - 1\right].$$

Therefore, it holds that

$$\frac{2cL'\left[\left(\frac{T+1}{2}\right)^{\ln\left(1+(1-1/m)c\gamma'\right)} - 1\right]}{m\ln\left(1 + (1-1/m)c\gamma'\right)} \leq \epsilon_{\text{arg}} \leq \frac{2L\left[\left(1 + (1-1/m)\gamma c\right)T^{(1-1/m)\gamma c} - 1\right]}{(m-1)\gamma}. \quad (12)$$

Omitting the constants depending on $c$, $\gamma$, and $L$, $\gamma\prime$ and $L'$, we have $\frac{T^{\ln\left(1+(1-\frac{1}{m})c\gamma'\right)}}{m} \lesssim \epsilon_{\text{arg}} \lesssim \frac{T^{(1-\frac{1}{m})c\gamma}}{m}$, which completes the proof.

$\square$

## B.7 Proof of Theorem 5.6

Theorem 5.6: Following the same condition as in Theorem 5.5, and additionally, if the initial points $\boldsymbol{\theta}_0 = \mathbf{0}, \boldsymbol{\lambda}_0 = \mathbf{0}$, and $\boldsymbol{v}_1 \perp \boldsymbol{x}_j^{\text{val}}$ for any $j \in [m]\backslash i$ and $\boldsymbol{v}_1 \perp \boldsymbol{x}_j^{\text{tr}}$ for any $j \in [n]$, then the order of uniform stability $\epsilon_{\text{stab}}$ w.r.t. $T$ satisfies $\frac{T^{\ln\left(1+(1-\frac{1}{m})c\gamma'\right)}}{m} \lesssim \epsilon_{\text{stab}} \lesssim \frac{T^{(1-\frac{1}{m})c\gamma}}{m}$, where $\gamma = \gamma' \asymp (1 + \eta\gamma^{\text{tr}})^{2K}$ as in Proposition 5.4.

*Proof.* In the following, we show that $\epsilon_{\text{stab}}$ explicitly shows the same order as $\epsilon_{\text{arg}}$ in Theorem 5.5 with additional assumptions for Example 5.3. For the upper bound, it is easy to get with Lipschitz condition that

$$\epsilon_{\text{stab}} = \sup_{S^{\text{val}}\simeq\tilde{S}^{\text{val}}\in\mathcal{Z}^m, S^{\text{tr}}\in\mathcal{Z}^n, \boldsymbol{z}\in\mathcal{Z}} \mathbb{E}_{\mathcal{A}}[\|\mathcal{L}(\mathcal{A}(S^{\text{val}}, S^{\text{tr}}); \boldsymbol{z}) - \mathcal{L}(\mathcal{A}(\tilde{S}^{\text{val}}, S^{\text{tr}}); \boldsymbol{z})\|]$$

$$\leq \sup_{S^{\text{val}}\simeq\tilde{S}^{\text{val}}\in\mathcal{Z}^m, S^{\text{tr}}\in\mathcal{Z}^n} \mathbb{E}_{\mathcal{A}}[L\|\mathcal{A}(S^{\text{val}}, S^{\text{tr}}) - \mathcal{A}(\tilde{S}^{\text{val}}, S^{\text{tr}})\|]$$

$$= L\epsilon_{\text{arg}},$$

and according to Theorem 5.5, we have $\epsilon_{\text{stab}} \lesssim \frac{T^{(1-\frac{1}{m})c\gamma}}{m}$.

To obtain the lower bound, we need to explicitly derive the optimization process of $\mathcal{A}(S^{\text{val}}, S^{\text{tr}})$ and $\mathcal{A}(\tilde{S}^{\text{val}}, S^{\text{tr}})$ (i.e., $\boldsymbol{\lambda}_T$ and $\tilde{\boldsymbol{\lambda}}_T$), and corresponding loss values.

From the proof of Proposition 5.4, we know that

$$\boldsymbol{\theta}_K(\boldsymbol{\lambda}) = G_{\boldsymbol{\lambda},\eta}\Big(G_{\boldsymbol{\lambda},\eta}\big(\dots G_{\boldsymbol{\lambda},\eta}(\boldsymbol{\theta}_0)\big)\Big)$$

$$= (\boldsymbol{I} - \eta\boldsymbol{A})^K \boldsymbol{\theta}_0 - \eta \sum_{k=0}^{K-1} (\boldsymbol{I} - \eta\boldsymbol{A})^k (\boldsymbol{\lambda} - y_{j_k}^{\text{tr}} \boldsymbol{x}_{j_k}^{\text{tr}})$$

$$= -\eta \sum_{k=0}^{K-1} (\boldsymbol{I} - \eta\boldsymbol{A})^k (\boldsymbol{\lambda} - y_{j_k}^{\text{tr}} \boldsymbol{x}_{j_k}^{\text{tr}}). \qquad (\boldsymbol{\theta}_0 = \boldsymbol{0})$$

$$= -\eta \sum_{k=0}^{K-1} (\boldsymbol{I} - \eta\boldsymbol{A})^k \boldsymbol{\lambda} + \eta \sum_{k=0}^{K-1} (\boldsymbol{I} - \eta\boldsymbol{A})^k y_{j_k}^{\text{tr}} \boldsymbol{x}_{j_k}^{\text{tr}}$$

$$:= -\boldsymbol{B}_K \boldsymbol{\lambda} + \boldsymbol{b}_K^{\text{tr}},$$

where symmetric matrix $\boldsymbol{B}_K := \eta \sum_{k=0}^{K-1} (\boldsymbol{I} - \eta\boldsymbol{A})^k$ and vector $\boldsymbol{b}_K^{\text{tr}} := \eta \sum_{k=0}^{K-1} (\boldsymbol{I} - \eta\boldsymbol{A})^k y_{j_k}^{\text{tr}} \boldsymbol{x}_{j_k}^{\text{tr}}$.

Building upon $\boldsymbol{\theta}_K(\boldsymbol{\lambda})$, we can derive $\mathcal{L}(\boldsymbol{\lambda}, \boldsymbol{z})$ as

$$\mathcal{L}(\boldsymbol{\lambda}, \boldsymbol{z}) = \frac{1}{2}\boldsymbol{\theta}_K^\top(\boldsymbol{\lambda})\boldsymbol{A}\boldsymbol{\theta}_K(\boldsymbol{\lambda}) + \boldsymbol{\lambda}^\top \boldsymbol{\theta}_K(\boldsymbol{\lambda}) - y\boldsymbol{x}^\top \boldsymbol{\theta}_K(\boldsymbol{\lambda})$$

$$= \frac{1}{2}(-\boldsymbol{B}_K\boldsymbol{\lambda} + \boldsymbol{b}_K^{\text{tr}})^\top \boldsymbol{A}(-\boldsymbol{B}_K\boldsymbol{\lambda} + \boldsymbol{b}_K^{\text{tr}}) + \boldsymbol{\lambda}^\top(-\boldsymbol{B}_K\boldsymbol{\lambda} + \boldsymbol{b}_K^{\text{tr}}) - y\boldsymbol{x}^\top(-\boldsymbol{B}_K\boldsymbol{\lambda} + \boldsymbol{b}_K^{\text{tr}})$$

$$= \frac{1}{2}\boldsymbol{\lambda}^\top(\boldsymbol{B}_K\boldsymbol{A}\boldsymbol{B}_K - 2\boldsymbol{B}_K)\boldsymbol{\lambda} + (-\boldsymbol{b}_K^{\text{tr}\top}\boldsymbol{A}\boldsymbol{B}_K + \boldsymbol{b}_K^{\text{tr}\top} + y\boldsymbol{x}^\top\boldsymbol{B}_K)\boldsymbol{\lambda} + \frac{1}{2}\boldsymbol{b}_K^{\text{tr}\top}\boldsymbol{A}\boldsymbol{b}_K^{\text{tr}} - y\boldsymbol{x}^\top\boldsymbol{b}_K^{\text{tr}}$$

$$= \frac{1}{2}\boldsymbol{\lambda}^\top\boldsymbol{B}_K(\boldsymbol{A}\boldsymbol{B}_K - 2\boldsymbol{I})\boldsymbol{\lambda} + (-\boldsymbol{b}_K^{\text{tr}\top}\boldsymbol{A}\boldsymbol{B}_K + \boldsymbol{b}_K^{\text{tr}\top} + y\boldsymbol{x}^\top\boldsymbol{B}_K)\boldsymbol{\lambda} + \frac{1}{2}\boldsymbol{b}_K^{\text{tr}\top}\boldsymbol{A}\boldsymbol{b}_K^{\text{tr}} - y\boldsymbol{x}^\top\boldsymbol{b}_K^{\text{tr}},$$

whose gradient is

$$\nabla_{\boldsymbol{\lambda}}\mathcal{L}(\boldsymbol{\lambda}, \boldsymbol{z}) = \boldsymbol{B}_K(\boldsymbol{A}\boldsymbol{B}_K - 2\boldsymbol{I})\boldsymbol{\lambda} + (-\boldsymbol{B}_K\boldsymbol{A}\boldsymbol{b}_K^{\text{tr}} + \boldsymbol{b}_K^{\text{tr}} + y\boldsymbol{B}_K\boldsymbol{x})$$

$$= \boldsymbol{B}_K(\boldsymbol{A}\boldsymbol{B}_K - 2\boldsymbol{I})\boldsymbol{\lambda} + y\boldsymbol{B}_K\boldsymbol{x} + \boldsymbol{b}_K^{\text{tr}} - \boldsymbol{B}_K\boldsymbol{A}\boldsymbol{b}_K^{\text{tr}}.$$

Then, the update rule of $\boldsymbol{\lambda}$ can be expressed as

$$\boldsymbol{\lambda}_t = \boldsymbol{\lambda}_{t-1} - \alpha_t \nabla_{\boldsymbol{\lambda}}\mathcal{L}(\boldsymbol{\lambda}_{t-1}, \boldsymbol{z}_{i_t})$$

$$= \boldsymbol{\lambda}_{t-1} - \alpha_t\Big[\boldsymbol{B}_K(\boldsymbol{A}\boldsymbol{B}_K - 2\boldsymbol{I})\boldsymbol{\lambda}_{t-1} + y_{i_t}\boldsymbol{B}_K\boldsymbol{x}_{i_t} + \boldsymbol{b}_K^{\text{tr}} - \boldsymbol{B}_K\boldsymbol{A}\boldsymbol{b}_K^{\text{tr}}\Big]$$

$$= \Big[\boldsymbol{I} - \alpha_t\boldsymbol{B}_K(\boldsymbol{A}\boldsymbol{B}_K - 2\boldsymbol{I})\Big]\boldsymbol{\lambda}_{t-1} - \alpha_t y_{i_t}\boldsymbol{B}_K\boldsymbol{x}_{i_t} - \alpha_t(\boldsymbol{b}_K^{\text{tr}} - \boldsymbol{B}_K\boldsymbol{A}\boldsymbol{b}_K^{\text{tr}}).$$

Now, by unwinding the recurrence from $T$ to 1 with $\boldsymbol{\lambda}_0 = \boldsymbol{0}$, we can obtain

$$\boldsymbol{\lambda}_T = \prod_{t=1}^{T}\big[\boldsymbol{I} - \alpha_t\boldsymbol{B}_K(\boldsymbol{A}\boldsymbol{B}_K - 2\boldsymbol{I})\big]\boldsymbol{\lambda}_0 + \sum_{t=1}^{T}\prod_{s=t+1}^{T}\big[\boldsymbol{I} - \alpha_s\boldsymbol{B}_K(\boldsymbol{A}\boldsymbol{B}_K - 2\boldsymbol{I})\big](-\alpha_t y_{i_t}\boldsymbol{B}_K\boldsymbol{x}_{i_t})$$

$$+ \sum_{t=1}^{T}\prod_{s=t+1}^{T}\big[\boldsymbol{I} - \alpha_s\boldsymbol{B}_K(\boldsymbol{A}\boldsymbol{B}_K - 2\boldsymbol{I})\big](-\alpha_t(\boldsymbol{b}_K^{\text{tr}} - \boldsymbol{B}_K\boldsymbol{A}\boldsymbol{b}_K^{\text{tr}}))$$

$$= \underbrace{\sum_{t=1}^{T}\prod_{s=t+1}^{T}\big[\boldsymbol{I} - \alpha_s\boldsymbol{B}_K(\boldsymbol{A}\boldsymbol{B}_K - 2\boldsymbol{I})\big](-\alpha_t y_{i_t}\boldsymbol{B}_K\boldsymbol{x}_{i_t})}_{\boldsymbol{r}}$$

$$+ \underbrace{\sum_{t=1}^{T}\prod_{s=t+1}^{T}\big[\boldsymbol{I} - \alpha_s\boldsymbol{B}_K(\boldsymbol{A}\boldsymbol{B}_K - 2\boldsymbol{I})\big](-\alpha_t(\boldsymbol{b}_K^{\text{tr}} - \boldsymbol{B}_K\boldsymbol{A}\boldsymbol{b}_K^{\text{tr}}))}_{\boldsymbol{a}_1}. \qquad (\boldsymbol{\lambda}_0 = \boldsymbol{0})$$

Recall that $S^{\text{val}}$ and $\tilde{S}^{\text{val}}$ only differ in the $i$-th entry where $\boldsymbol{z}_i = (\boldsymbol{x}_i, y_i) = (\boldsymbol{v}_1, 1)$, $\tilde{\boldsymbol{z}}_i = (\tilde{\boldsymbol{x}}_i, \tilde{y}_i) = (-\boldsymbol{v}_1, 1)$. Denote $\mathbb{1}[\cdot]$ as the indicator function. We simplify the term $\boldsymbol{r}$ as follows:

$$
\boldsymbol{r} = \sum_{t=1}^{T} \prod_{s=t+1}^{T} \left[\boldsymbol{I} - \alpha_s \boldsymbol{B}_K(\boldsymbol{A}\boldsymbol{B}_K - 2\boldsymbol{I})\right] \left(-\alpha_t \boldsymbol{B}_K \sum_{j=1}^{m} y_j \boldsymbol{x}_j \mathbb{1}[i_t = j]\right)
$$

$$
= \sum_{j=1}^{m} \sum_{t=1}^{T} \mathbb{1}[i_t = j] \prod_{s=t+1}^{T} \left[\boldsymbol{I} - \alpha_s \boldsymbol{B}_K(\boldsymbol{A}\boldsymbol{B}_K - 2\boldsymbol{I})\right] \left(-\alpha_t \boldsymbol{B}_K y_j \boldsymbol{x}_j\right)
$$

$$
= \sum_{t=1}^{T} \mathbb{1}[i_t = i] \prod_{s=t+1}^{T} \left[\boldsymbol{I} - \alpha_s \boldsymbol{B}_K(\boldsymbol{A}\boldsymbol{B}_K - 2\boldsymbol{I})\right] \left(-\alpha_t \boldsymbol{B}_K y_i \boldsymbol{x}_i\right)
$$

$$
+ \sum_{j \neq i}^{m} \sum_{t=1}^{T} \mathbb{1}[i_t = j] \prod_{s=t+1}^{T} \left[\boldsymbol{I} - \alpha_s \boldsymbol{B}_K(\boldsymbol{A}\boldsymbol{B}_K - 2\boldsymbol{I})\right] \left(-\alpha_t \boldsymbol{B}_K y_j \boldsymbol{x}_j\right)
$$

$$
= \sum_{t=1}^{T} \mathbb{1}[i_t = i] \prod_{s=t+1}^{T} \left[\boldsymbol{I} - \alpha_s \boldsymbol{B}_K(\boldsymbol{A}\boldsymbol{B}_K - 2\boldsymbol{I})\right] \left(-\alpha_t \boldsymbol{B}_K \boldsymbol{v}_1\right)
$$

$$
+ \sum_{j \neq i}^{m} \sum_{t=1}^{T} \mathbb{1}[i_t = j] \prod_{s=t+1}^{T} \left[\boldsymbol{I} - \alpha_s \boldsymbol{B}_K(\boldsymbol{A}\boldsymbol{B}_K - 2\boldsymbol{I})\right] \left(-\alpha_t \boldsymbol{B}_K y_j \boldsymbol{x}_j\right)
$$

$$
= \sum_{t=1}^{T} \mathbb{1}[i_t = i] \prod_{s=t+1}^{T} \left[\boldsymbol{I} - \alpha_s \eta \sum_{k=0}^{K-1}(\boldsymbol{I} - \eta\boldsymbol{A})^k(\boldsymbol{A}\eta \sum_{k=0}^{K-1}(\boldsymbol{I} - \eta\boldsymbol{A})^k - 2\boldsymbol{I})\right] \left(-\alpha_t \eta \sum_{k=0}^{K-1}(\boldsymbol{I} - \eta\boldsymbol{A})^k \boldsymbol{v}_1\right)
$$

$$
+ \sum_{j \neq i}^{m} \sum_{t=1}^{T} \mathbb{1}[i_t = j] \prod_{s=t+1}^{T} \left[\boldsymbol{I} - \alpha_s \boldsymbol{B}_K(\boldsymbol{A}\boldsymbol{B}_K - 2\boldsymbol{I})\right] \left(-\alpha_t \boldsymbol{B}_K y_j \boldsymbol{x}_j\right)
$$

$$
= \sum_{t=1}^{T} \mathbb{1}[i_t = i] \prod_{s=t+1}^{T} \left[1 - \alpha_s \eta \sum_{k=0}^{K-1}(1 - \eta\gamma_1)^k(\gamma_1 \eta \sum_{k=0}^{K-1}(1 - \eta\gamma_1)^k - 2)\right] \left(-\alpha_t \eta \sum_{k=0}^{K-1}(1 - \eta\gamma_1)^k\right) \boldsymbol{v}_1
$$

$$
+ \sum_{j \neq i}^{m} \sum_{t=1}^{T} \mathbb{1}[i_t = j] \prod_{s=t+1}^{T} \left[\boldsymbol{I} - \alpha_s \boldsymbol{B}_K(\boldsymbol{A}\boldsymbol{B}_K - 2\boldsymbol{I})\right] \left(-\alpha_t \boldsymbol{B}_K y_j \boldsymbol{x}_j\right)
$$

$$
= - \underbrace{\sum_{t=1}^{T} \mathbb{1}[i = i_t] \prod_{s=t+1}^{T} \left[1 - \alpha_s L'(\gamma_1 L' - 2)\right] \left(\alpha_t L'\right) \boldsymbol{v}_1}_{\boldsymbol{b} := \tau \boldsymbol{v}_1}
$$

$$
+ \underbrace{\sum_{j \neq i}^{m} \sum_{t=1}^{T} \mathbb{1}[i_t = j] \prod_{s=t+1}^{T} \left[\boldsymbol{I} - \alpha_s \boldsymbol{B}_K(\boldsymbol{A}\boldsymbol{B}_K - 2\boldsymbol{I})\right] \left(-\alpha_t \boldsymbol{B}_K y_j \boldsymbol{x}_j\right)}_{\boldsymbol{a}_2},
$$

where $L' = \eta \sum_{k=0}^{K-1}(1 - \eta\gamma_1)^k$ as in Proposition B.5. We further define $\boldsymbol{a} := \boldsymbol{a}_1 + \boldsymbol{a}_2$, and then $\boldsymbol{\lambda}_T = \boldsymbol{a}_1 + \boldsymbol{a}_2 - \boldsymbol{b} = \boldsymbol{a} - \boldsymbol{b}$. Follow the same process of derivation, we have $\tilde{\boldsymbol{\lambda}}_T = \boldsymbol{a} + \boldsymbol{b}$ where the opposite symbol for $b$ arise from $\boldsymbol{x}_i = \boldsymbol{v}_1$ while $\tilde{\boldsymbol{x}}_i = \boldsymbol{v}_1$.

Recall that we have assumed that $\boldsymbol{v}_1 \perp \boldsymbol{x}_k^{\text{tr}}$ for any $k \in [n]$ and $\boldsymbol{v}_1 \perp \boldsymbol{x}_j^{\text{val}}$ for any $j \in [m]$ that $j \neq i$, thus $\boldsymbol{v}_1^\top \boldsymbol{b}_K^{\text{tr}} = 0$ and $\boldsymbol{v}_1^\top \boldsymbol{x}_j = 0, j \neq i \in [m]$. Therefore,

$$
\boldsymbol{a}^\top \boldsymbol{b} = \sum_{t=1}^{T}(-\alpha_t(\boldsymbol{b}_K^{\text{tr}} - \boldsymbol{B}_K \boldsymbol{A} \boldsymbol{b}_K^{\text{tr}}))^\top \prod_{s=t+1}^{T} \left[\boldsymbol{I} - \alpha_s \boldsymbol{B}_K(\boldsymbol{A}\boldsymbol{B}_K - 2\boldsymbol{I})\right] \tau \boldsymbol{v}_1
$$

$$
+ \sum_{j \neq i}^{m} \sum_{t=1}^{T} \mathbb{1}[i_t = j] \left(-\alpha_t \boldsymbol{B}_K y_j \boldsymbol{x}_j\right)^\top \prod_{s=t+1}^{T} \left[\boldsymbol{I} - \alpha_s \boldsymbol{B}_K(\boldsymbol{A}\boldsymbol{B}_K - 2\boldsymbol{I})\right] \tau \boldsymbol{v}_1.
$$

As $\boldsymbol{B}_K := \eta \sum_{k=0}^{K-1}(\boldsymbol{I} - \eta\boldsymbol{A})^k$ and $\boldsymbol{v}_1$ is the eigenvector of $\boldsymbol{A}$, we have $\boldsymbol{a}^\top\boldsymbol{b} = 0$, i.e., $\boldsymbol{a} \perp \boldsymbol{b}$.

Now, we are ready to discuss $\mathcal{L}(\boldsymbol{\lambda}_T, \boldsymbol{z}) - \mathcal{L}(\tilde{\boldsymbol{\lambda}}_T, \boldsymbol{z})$.

$$
\begin{aligned}
&\mathcal{L}(\boldsymbol{\lambda}_T, \boldsymbol{z}) - \mathcal{L}(\tilde{\boldsymbol{\lambda}}_T, \boldsymbol{z}) \\
&= \frac{1}{2}\boldsymbol{\lambda}_T^\top\boldsymbol{B}_K(\boldsymbol{A}\boldsymbol{B}_K - 2\boldsymbol{I})\boldsymbol{\lambda}_T + (-\boldsymbol{b}_K^{\mathrm{tr}\top}\boldsymbol{A}\boldsymbol{B}_K + \boldsymbol{b}_K^{\mathrm{tr}\top} + y\boldsymbol{x}^\top\boldsymbol{B}_K)\boldsymbol{\lambda}_T \\
&\quad - \frac{1}{2}\tilde{\boldsymbol{\lambda}}_T^\top\boldsymbol{B}_K(\boldsymbol{A}\boldsymbol{B}_K - 2\boldsymbol{I})\tilde{\boldsymbol{\lambda}}_T - (-\boldsymbol{b}_K^{\mathrm{tr}\top}\boldsymbol{A}\boldsymbol{B}_K + \boldsymbol{b}_K^{\mathrm{tr}\top} + y\boldsymbol{x}^\top\boldsymbol{B}_K)\tilde{\boldsymbol{\lambda}}_T \\
&= \underbrace{\frac{1}{2}\boldsymbol{\lambda}_T^\top\boldsymbol{B}_K(\boldsymbol{A}\boldsymbol{B}_K - 2\boldsymbol{I})\boldsymbol{\lambda}_T - \frac{1}{2}\boldsymbol{\lambda}_T'^\top\boldsymbol{B}_K(\boldsymbol{A}\boldsymbol{B}_K - 2\boldsymbol{I})\tilde{\boldsymbol{\lambda}}_T}_{c} + (-\boldsymbol{b}_K^{\mathrm{tr}\top}\boldsymbol{A}\boldsymbol{B}_K + \boldsymbol{b}_K^{\mathrm{tr}\top} + y\boldsymbol{x}^\top\boldsymbol{B}_K)(\boldsymbol{\lambda}_T - \tilde{\boldsymbol{\lambda}}_T),
\end{aligned}
$$

where

$$
\begin{aligned}
\boldsymbol{c} &= \frac{1}{2}(\boldsymbol{a} - \boldsymbol{b})^\top\boldsymbol{B}_K(\boldsymbol{A}\boldsymbol{B}_K - 2\boldsymbol{I})(\boldsymbol{a} - \boldsymbol{b}) - \frac{1}{2}(\boldsymbol{a} + \boldsymbol{b})^\top\boldsymbol{B}_K(\boldsymbol{A}\boldsymbol{B}_K - 2\boldsymbol{I})(\boldsymbol{a} + \boldsymbol{b}) \\
&= -2\boldsymbol{a}^\top\boldsymbol{B}_K(\boldsymbol{A}\boldsymbol{B}_K - 2\boldsymbol{I})\boldsymbol{b} - 2\boldsymbol{b}^\top\boldsymbol{B}_K(\boldsymbol{A}\boldsymbol{B}_K - 2\boldsymbol{I})\boldsymbol{a} \\
&= -4\boldsymbol{a}^\top\boldsymbol{B}_K(\boldsymbol{A}\boldsymbol{B}_K - 2\boldsymbol{I})\boldsymbol{b} &\text{(by symmetric)} \\
&= 4\boldsymbol{a}^\top\boldsymbol{B}_K(\boldsymbol{A}\boldsymbol{B}_K - 2\boldsymbol{I})\tau\boldsymbol{v}_1 \\
&= 4(\gamma_1 L' - 2)\tau\boldsymbol{a}^\top\boldsymbol{v}_1 \\
&= \boldsymbol{0}.
\end{aligned}
$$

Therefore, $\mathcal{L}(\boldsymbol{\lambda}_T, \boldsymbol{z}) - \mathcal{L}(\tilde{\boldsymbol{\lambda}}_T, \boldsymbol{z})$ can be simplified as

$$
\begin{aligned}
\mathcal{L}(\boldsymbol{\lambda}_T, \boldsymbol{z}) - \mathcal{L}(\tilde{\boldsymbol{\lambda}}_T, \boldsymbol{z}) &= (-\boldsymbol{b}_K^{\mathrm{tr}\top}\boldsymbol{A}\boldsymbol{B}_K + \boldsymbol{b}_K^{\mathrm{tr}\top} + y\boldsymbol{x}^\top\boldsymbol{B}_K)(\boldsymbol{\lambda}_T - \tilde{\boldsymbol{\lambda}}_T) \\
&= -2(-\boldsymbol{b}_K^{\mathrm{tr}\top}\boldsymbol{A}\boldsymbol{B}_K + \boldsymbol{b}_K^{\mathrm{tr}\top} + y\boldsymbol{x}^\top\boldsymbol{B}_K)\boldsymbol{b} &(\boldsymbol{\lambda}_T - \tilde{\boldsymbol{\lambda}}_T = -2\boldsymbol{b}) \\
&= -2y\boldsymbol{x}^\top\boldsymbol{B}_K\boldsymbol{b} \\
&= -2yL'\tau\boldsymbol{x}^\top\boldsymbol{v}_1. &(\boldsymbol{b} = \tau\boldsymbol{v}_1)
\end{aligned}
$$

Let $\boldsymbol{z}^* = (\boldsymbol{v}_1, 1)$, we have

$$
|\mathcal{L}(\boldsymbol{\lambda}_T, \boldsymbol{z}^*) - \mathcal{L}(\tilde{\boldsymbol{\lambda}}_T, \boldsymbol{z}^*)| = 2L'\tau\|\boldsymbol{v}_1\|^2 = 2L'\tau = L'\|\boldsymbol{\lambda}_T - \tilde{\boldsymbol{\lambda}}_T\|. \qquad (\|\boldsymbol{\lambda}_T - \tilde{\boldsymbol{\lambda}}_T\| = 2\tau)
$$

Therefore, by the definition of $\epsilon_{\mathrm{stab}}$, we have

$$
\epsilon_{\mathrm{stab}} \geq |\mathcal{L}(\boldsymbol{\lambda}_T, \boldsymbol{z}^*) - \mathcal{L}(\tilde{\boldsymbol{\lambda}}_T, \boldsymbol{z}^*)| = L'\|\boldsymbol{\lambda}_T - \tilde{\boldsymbol{\lambda}}_T\| \geq \sum_{t=1}^{T}\prod_{s=t+1}^{T}\left(1 + \alpha_s(1 - 1/m)\gamma'\right)\frac{2\alpha_t L'}{m},
$$

where the last inequality holds for Eq. (8) in the proof of Theorem 5.1. Following the proof of Theorem B.6, we can further derive

$$
\epsilon_{\mathrm{stab}} \geq \frac{2cL'^2\left[\left(\frac{T+1}{2}\right)^{\ln\left(1+(1-1/m)c\gamma'\right)} - 1\right]}{m\ln\left(1 + (1 - 1/m)c\gamma'\right)}.
$$

Omitting the constants regarding $c$, $\gamma'$ and $L'$, we have $\epsilon_{\mathrm{stab}} \gtrsim \frac{T^{\ln\left(1+(1-\frac{1}{m})c\gamma'\right)}}{m}$, which completes the proof. $\qquad\square$

## C  Deferred results of IFT-based HO algorithm

Based on Example 5.3, we also investigate and establish a stability lower bound for the IFT-based algorithm. The stability analysis for the IFT-based algorithm is conducted following the same proof idea as the UD-based algorithm. Similarly to the analysis for the UD-based algorithm, we first obtain the expansive and divergent properties of the outer level in Proposition C.5 of Appendix C. These jointly lead to uniform argument stability bounds in Theorem C.6. For completeness, we also derive an upper bound for the IFT algorithm based on existing techniques [17], presented together as follows.

We first introduce several lemmas useful for the following proofs.

**Lemma C.1** (Lemma 2 in [17]). *Suppose $\Lambda$ and $\Theta$ are convex and compact with non-empty interiors, $\mathcal{Z}$ is compact, $\Lambda \times \Theta \times \mathcal{Z}$ is included in an open set $\Omega$ and $f(\boldsymbol{\lambda}, \boldsymbol{\theta}; \boldsymbol{z}) \in C^k(\Omega)$, then for all $i \le k-1$ order partial differential $h(\boldsymbol{\lambda}, \boldsymbol{\theta}; \boldsymbol{z})$ of $f(\boldsymbol{\lambda}, \boldsymbol{\theta}; \boldsymbol{z})$, we have $\sup_{\boldsymbol{\theta} \in \Theta, \boldsymbol{z} \in \mathcal{Z}} ||h(\boldsymbol{\lambda}, \boldsymbol{\theta}; \boldsymbol{z})||_{\boldsymbol{\lambda} \in \Lambda, Lip} < \infty$ and $\sup_{\boldsymbol{\lambda} \in \Lambda, \boldsymbol{z} \in \mathcal{Z}} ||h(\boldsymbol{\lambda}, \boldsymbol{\theta}; \boldsymbol{z})||_{\boldsymbol{\theta} \in \Theta, Lip} < \infty$.*

Lemma C.1 implies that any $i \le 1$ order partial differential of $\ell^{\mathrm{val}}(\boldsymbol{\lambda}, \boldsymbol{\theta}; \boldsymbol{z})$ is Lipschitz and any $i \le 2$ order partial differential of $\ell^{\mathrm{tr}}(\boldsymbol{\lambda}, \boldsymbol{\theta}, \boldsymbol{z})$ is Lipschitz continuous under Assumption B.1. We denote the maximal Lipschitz constants among them as $Q$.

**Lemma C.2** (Theorem 3 in [17]). *Denote $\boldsymbol{\theta}_K(\boldsymbol{\lambda})$ as $L^{\boldsymbol{\theta}_K}$-Lipschitz continuous, we have $L^{\boldsymbol{\theta}_K} \lesssim (1 + \eta\gamma^{\mathrm{tr}})^K$.*

**Lemma C.3.** *In the case of Example 5.3, the $\widehat{\nabla_{\boldsymbol{\lambda}}\boldsymbol{\theta}_K}(\boldsymbol{\lambda})$ calculated by the IFT-based algorithm is exactly $\nabla_{\boldsymbol{\lambda}}\boldsymbol{\theta}_K(\boldsymbol{\lambda})$.*

*Proof.*

$$\widehat{\nabla_{\boldsymbol{\lambda}}\boldsymbol{\theta}_K}(\boldsymbol{\lambda}) = -\nabla^2_{\boldsymbol{\theta}\boldsymbol{\lambda}}\ell^{\mathrm{tr}}(\boldsymbol{\lambda}, \boldsymbol{\theta}_K(\boldsymbol{\lambda}))\eta \sum_{k=0}^{K-1} \left[\boldsymbol{I} - \eta\nabla^2_{\boldsymbol{\theta}\boldsymbol{\theta}}\ell^{\mathrm{tr}}(\boldsymbol{\lambda}, \boldsymbol{\theta}_K(\boldsymbol{\lambda}))\right]^k$$

$$= -\eta \sum_{k=0}^{K-1} [\boldsymbol{I} - \eta\boldsymbol{A}]^k$$

$$= \nabla_{\boldsymbol{\lambda}}\boldsymbol{\theta}_K(\boldsymbol{\lambda}). \tag{Eq. (9)}$$

$\square$

Now, we are ready to prove Propositions C.4 and C.5.

**Proposition C.4** (Lipshchitz properties of IFT-based algorithm). *Suppose we solve Example 5.3 by IFT-based Algorithm 1 with constant inner step size $\eta$ where $1 - \eta\gamma_d \ge 0$ and outer step size $\alpha_t$. Then the composite validation loss $\mathcal{L}(\cdot; \boldsymbol{z})$ is L-Lipschitz continuous and $\gamma$-smooth for all $\boldsymbol{z} \in S^{\mathrm{val}}$, where*

$$L \lesssim (1 + \eta\gamma^{\mathrm{tr}})^K, \gamma \lesssim K(1 + \eta\gamma^{\mathrm{tr}})^{2K}.$$

*Proof.* According to Lemma 1 in [17], the Lipschitz continuous coefficient $L = \sup_{\boldsymbol{\lambda} \in \Lambda} \left\|\nabla_{\boldsymbol{\lambda}}\mathcal{L}(\boldsymbol{\lambda}; \boldsymbol{z})\right\|$. For all $\boldsymbol{\lambda} \in \Lambda$, we have

$$\left\|\nabla_{\boldsymbol{\lambda}}\mathcal{L}(\boldsymbol{\lambda}; \boldsymbol{z})\right\| = \left\|\nabla_{\boldsymbol{\lambda}}\ell^{\mathrm{val}}(\boldsymbol{\lambda}, \boldsymbol{\theta}_K(\boldsymbol{\lambda}); \boldsymbol{z}) + \widehat{\nabla_{\boldsymbol{\lambda}}\boldsymbol{\theta}_K}(\boldsymbol{\lambda})\nabla_{\boldsymbol{\theta}}\ell^{\mathrm{val}}(\boldsymbol{\lambda}, \boldsymbol{\theta}_K(\boldsymbol{\lambda}); \boldsymbol{z})\right\|$$

$$\le \left\|\nabla_{\boldsymbol{\lambda}}\ell^{\mathrm{val}}(\boldsymbol{\lambda}, \boldsymbol{\theta}_K(\boldsymbol{\lambda}); \boldsymbol{z})\right\| + \left\|\widehat{\nabla_{\boldsymbol{\lambda}}\boldsymbol{\theta}_K}(\boldsymbol{\lambda})\right\| \left\|\nabla_{\boldsymbol{\theta}}\ell^{\mathrm{val}}(\boldsymbol{\lambda}, \boldsymbol{\theta}_K(\boldsymbol{\lambda}); \boldsymbol{z})\right\|$$

$$\le Q + Q\left\|\widehat{\nabla_{\boldsymbol{\lambda}}\boldsymbol{\theta}_K}(\boldsymbol{\lambda})\right\|$$

$$= Q + Q\left\|\nabla^2_{\boldsymbol{\theta}\boldsymbol{\lambda}}\ell^{\mathrm{tr}}(\boldsymbol{\lambda}, \boldsymbol{\theta}_K(\boldsymbol{\lambda}))\eta \sum_{k=0}^{K-1}\left[\boldsymbol{I} - \eta\nabla^2_{\boldsymbol{\theta}\boldsymbol{\theta}}\ell^{\mathrm{tr}}(\boldsymbol{\lambda}, \boldsymbol{\theta}_K(\boldsymbol{\lambda}))\right]^k\right\|$$

$$\le Q + \eta Q\left\|\nabla^2_{\boldsymbol{\theta}\boldsymbol{\lambda}}\ell^{\mathrm{tr}}(\boldsymbol{\lambda}, \boldsymbol{\theta}_K(\boldsymbol{\lambda}))\right\| \sum_{k=0}^{K-1}\left\|\boldsymbol{I} - \eta\nabla^2_{\boldsymbol{\theta}\boldsymbol{\theta}}\ell^{\mathrm{tr}}(\boldsymbol{\lambda}, \boldsymbol{\theta}_K(\boldsymbol{\lambda}))\right\|^k$$

$$\le Q + \eta Q^2 \sum_{k=0}^{K-1}\left(1 + \eta\|\nabla^2_{\boldsymbol{\theta}\boldsymbol{\theta}}\ell^{\mathrm{tr}}(\boldsymbol{\lambda}, \boldsymbol{\theta}_K(\boldsymbol{\lambda}))\|\right)^k$$

$$\le Q + \eta Q^2 \sum_{k=0}^{K-1}\left(1 + \eta\gamma^{\mathrm{tr}}\right)^k$$

$$= Q + Q^2 \frac{\left(1 + \eta\gamma^{\mathrm{tr}}\right)^K - 1}{\gamma^{\mathrm{tr}}}.$$

Omitting the constants depending on $Q$, $\eta$, and $\gamma^{\mathrm{tr}}$, we get $L \lesssim \left( \left( 1 + \eta \gamma^{\mathrm{tr}} \right)^K \right)$.

To obtain the smoothness coefficient $\gamma$, we first discuss the Lipschitz continuty coefficient of $\widehat{\nabla_{\boldsymbol{\lambda}} \boldsymbol{\theta}_K}(\boldsymbol{\lambda})$. In the following, we use $\ell^{\mathrm{tr}}$ to represent $\ell^{\mathrm{tr}}(\boldsymbol{\lambda}, \boldsymbol{\theta}_K(\boldsymbol{\lambda}))$ when there is no ambiguity.

$$
\nabla_{\boldsymbol{\lambda}} \widehat{\nabla_{\boldsymbol{\lambda}} \boldsymbol{\theta}_K}(\boldsymbol{\lambda}) = \nabla_{\boldsymbol{\lambda}} \left( -\nabla_{\boldsymbol{\theta}\boldsymbol{\lambda}}^2 \ell^{\mathrm{tr}}(\boldsymbol{\lambda}, \boldsymbol{\theta}_K(\boldsymbol{\lambda})) \eta \sum_{k=0}^{K-1} \left[ \boldsymbol{I} - \eta \nabla_{\boldsymbol{\theta}\boldsymbol{\theta}}^2 \ell^{\mathrm{tr}}(\boldsymbol{\lambda}, \boldsymbol{\theta}_K(\boldsymbol{\lambda})) \right]^k \right)
$$

$$
= \underbrace{- \left( \left( \nabla_{\boldsymbol{\theta}\boldsymbol{\lambda}\boldsymbol{\lambda}}^3 \ell^{\mathrm{tr}} + \nabla_{\boldsymbol{\lambda}} \boldsymbol{\theta}_K(\boldsymbol{\lambda}) \nabla_{\boldsymbol{\theta}\boldsymbol{\lambda}\boldsymbol{\theta}}^3 \ell^{\mathrm{tr}} \right) \eta \sum_{k=0}^{K-1} \left[ \boldsymbol{I} - \eta \nabla_{\boldsymbol{\theta}\boldsymbol{\theta}}^2 \ell^{\mathrm{tr}} \right]^k \right)}_{\boldsymbol{B}_1}
$$

$$
\underbrace{- \left( \nabla_{\boldsymbol{\theta}\boldsymbol{\lambda}}^2 \ell^{\mathrm{tr}} \eta \sum_{k=1}^{K-1} \left( -\eta \left( \nabla_{\boldsymbol{\theta}\boldsymbol{\theta}\boldsymbol{\lambda}}^3 \ell^{\mathrm{tr}} + \nabla_{\boldsymbol{\lambda}} \boldsymbol{\theta}_K(\boldsymbol{\lambda}) \nabla_{\boldsymbol{\theta}\boldsymbol{\lambda}\boldsymbol{\theta}}^3 \ell^{\mathrm{tr}} \right) \right) k \left[ \boldsymbol{I} - \eta \nabla_{\boldsymbol{\theta}\boldsymbol{\theta}}^2 \ell^{\mathrm{tr}}(\boldsymbol{\lambda}, \boldsymbol{\theta}_K(\boldsymbol{\lambda})) \right]^{k-1} \right)}_{\boldsymbol{B}_2}.
$$

We bound the spectral norm of $\boldsymbol{B}_1$ and $\boldsymbol{B}_2$, respectively.

$$
\|\boldsymbol{B}_1\| = \left\| \left( \nabla_{\boldsymbol{\theta}\boldsymbol{\lambda}\boldsymbol{\lambda}}^3 \ell^{\mathrm{tr}} + \nabla_{\boldsymbol{\lambda}} \boldsymbol{\theta}_K(\boldsymbol{\lambda}) \nabla_{\boldsymbol{\theta}\boldsymbol{\lambda}\boldsymbol{\theta}}^3 \ell^{\mathrm{tr}} \right) \eta \sum_{k=0}^{K-1} \left[ \boldsymbol{I} - \eta \nabla_{\boldsymbol{\theta}\boldsymbol{\theta}}^2 \ell^{\mathrm{tr}} \right]^k \right\|
$$

$$
\leq \left( Q + Q \| \nabla_{\boldsymbol{\lambda}} \boldsymbol{\theta}_K(\boldsymbol{\lambda}) \| \right) \left( \eta \sum_{k=0}^{K-1} (1 + \eta \gamma^{\mathrm{tr}})^k \right)
$$

$$
\leq \left( Q + Q L^{\boldsymbol{\theta}_K} \right) \left( \eta \sum_{k=0}^{K-1} (1 + \eta \gamma^{\mathrm{tr}})^k \right)
$$

$$
\lesssim (1 + \eta \gamma^{\mathrm{tr}})^{2K}. \tag{Lemma C.2}
$$

In addition,

$$
\|\boldsymbol{B}_2\| = \left\| \nabla_{\boldsymbol{\theta}\boldsymbol{\lambda}}^2 \ell^{\mathrm{tr}} \eta \sum_{k=1}^{K-1} \left( -\eta \left( \nabla_{\boldsymbol{\theta}\boldsymbol{\theta}\boldsymbol{\lambda}}^3 \ell^{\mathrm{tr}} + \nabla_{\boldsymbol{\lambda}} \boldsymbol{\theta}_K(\boldsymbol{\lambda}) \nabla_{\boldsymbol{\theta}\boldsymbol{\lambda}\boldsymbol{\theta}}^3 \ell^{\mathrm{tr}} \right) \right) k \left[ \boldsymbol{I} - \eta \nabla_{\boldsymbol{\theta}\boldsymbol{\theta}}^2 \ell^{\mathrm{tr}}(\boldsymbol{\lambda}, \boldsymbol{\theta}_K(\boldsymbol{\lambda})) \right]^{k-1} \right\|
$$

$$
= \left\| \nabla_{\boldsymbol{\theta}\boldsymbol{\lambda}}^2 \ell^{\mathrm{tr}} \eta \left( -\eta \left( \nabla_{\boldsymbol{\theta}\boldsymbol{\theta}\boldsymbol{\lambda}}^3 \ell^{\mathrm{tr}} + \nabla_{\boldsymbol{\lambda}} \boldsymbol{\theta}_K(\boldsymbol{\lambda}) \nabla_{\boldsymbol{\theta}\boldsymbol{\lambda}\boldsymbol{\theta}}^3 \ell^{\mathrm{tr}} \right) \right) \sum_{k=1}^{K-1} k \left[ \boldsymbol{I} - \eta \nabla_{\boldsymbol{\theta}\boldsymbol{\theta}}^2 \ell^{\mathrm{tr}}(\boldsymbol{\lambda}, \boldsymbol{\theta}_K(\boldsymbol{\lambda})) \right]^{k-1} \right\|
$$

$$
\leq Q \eta^2 \left( Q + Q \| \nabla_{\boldsymbol{\lambda}} \boldsymbol{\theta}_K(\boldsymbol{\lambda}) \| \right) \left( \sum_{k=1}^{K-1} k (1 + \eta \gamma^{\mathrm{tr}})^{k-1} \right)
$$

$$
\leq Q \eta^2 \left( Q + Q L^{\boldsymbol{\theta}_K} \right) \left( K (1 + \eta \gamma^{\mathrm{tr}})^K - \frac{(1 + \eta \gamma^{\mathrm{tr}})^K - (1 + \eta \gamma^{\mathrm{tr}})}{\eta \gamma^{\mathrm{tr}}} - 1 \right)
$$

$$
\lesssim K (1 + \eta \gamma^{\mathrm{tr}})^{2K}.
$$

Denote $\widehat{\nabla_{\boldsymbol{\lambda}} \boldsymbol{\theta}_K}(\boldsymbol{\lambda})$ to be $L^{\widehat{\nabla_{\boldsymbol{\lambda}} \boldsymbol{\theta}_K}}$-Lipschitz continuous for all $\boldsymbol{\lambda} \in \Lambda$, then we have

$$
L^{\widehat{\nabla_{\boldsymbol{\lambda}} \boldsymbol{\theta}_K}} = \sup_{\boldsymbol{\lambda} \in \Lambda} \left\| \nabla_{\boldsymbol{\lambda}} \widehat{\nabla_{\boldsymbol{\lambda}} \boldsymbol{\theta}_K}(\boldsymbol{\lambda}) \right\| \leq \|\boldsymbol{B}_1\| + \|\boldsymbol{B}_2\| \lesssim K (1 + \eta \gamma^{\mathrm{tr}})^{2K}.
$$

With the above result and Lemma C.3, we have that for all $\boldsymbol{\lambda}, \boldsymbol{\lambda}' \in \Lambda$,

$$
\begin{aligned}
\left\|\nabla_{\boldsymbol{\lambda}}\mathcal{L}(\boldsymbol{\lambda};\boldsymbol{z}) - \nabla_{\boldsymbol{\lambda}}\mathcal{L}(\boldsymbol{\lambda}';\boldsymbol{z})\right\| \leq{}& \left\|\nabla_{\boldsymbol{\lambda}}\ell^{\mathrm{val}}(\boldsymbol{\lambda},\boldsymbol{\theta}_K(\boldsymbol{\lambda});\boldsymbol{z}) - \nabla_{\boldsymbol{\lambda}}\ell^{\mathrm{val}}(\boldsymbol{\lambda}',\boldsymbol{\theta}_K(\boldsymbol{\lambda}');\boldsymbol{z})\right\| \\
&+ \left\|\widehat{\nabla_{\boldsymbol{\lambda}}\boldsymbol{\theta}_K}(\boldsymbol{\lambda})\nabla_{\boldsymbol{\theta}}\ell^{\mathrm{val}}(\boldsymbol{\lambda},\boldsymbol{\theta}_K(\boldsymbol{\lambda});\boldsymbol{z}) - \widehat{\nabla_{\boldsymbol{\lambda}}\boldsymbol{\theta}_K}(\boldsymbol{\lambda}')\nabla_{\boldsymbol{\theta}}\ell^{\mathrm{val}}(\boldsymbol{\lambda}',\boldsymbol{\theta}_K(\boldsymbol{\lambda}');\boldsymbol{z})\right\| \\
\leq{}& \left\|\nabla_{\boldsymbol{\lambda}}\ell^{\mathrm{val}}(\boldsymbol{\lambda},\boldsymbol{\theta}_K(\boldsymbol{\lambda});\boldsymbol{z}) - \nabla_{\boldsymbol{\lambda}}\ell^{\mathrm{val}}(\boldsymbol{\lambda}',\boldsymbol{\theta}_K(\boldsymbol{\lambda});\boldsymbol{z})\right\| \\
&+ \left\|\nabla_{\boldsymbol{\lambda}}\ell^{\mathrm{val}}(\boldsymbol{\lambda}',\boldsymbol{\theta}_K(\boldsymbol{\lambda});\boldsymbol{z}) - \nabla_{\boldsymbol{\lambda}}\ell^{\mathrm{val}}(\boldsymbol{\lambda}',\boldsymbol{\theta}_K(\boldsymbol{\lambda}');\boldsymbol{z})\right\| \\
&+ \left\|\widehat{\nabla_{\boldsymbol{\lambda}}\boldsymbol{\theta}_K}(\boldsymbol{\lambda})\nabla_{\boldsymbol{\theta}}\ell^{\mathrm{val}}(\boldsymbol{\lambda},\boldsymbol{\theta}_K(\boldsymbol{\lambda});\boldsymbol{z}) - \widehat{\nabla_{\boldsymbol{\lambda}}\boldsymbol{\theta}_K}(\boldsymbol{\lambda})\nabla_{\boldsymbol{\theta}}\ell^{\mathrm{val}}(\boldsymbol{\lambda}',\boldsymbol{\theta}_K(\boldsymbol{\lambda}');\boldsymbol{z})\right\| \\
&+ \left\|\widehat{\nabla_{\boldsymbol{\lambda}}\boldsymbol{\theta}_K}(\boldsymbol{\lambda})\nabla_{\boldsymbol{\theta}}\ell^{\mathrm{val}}(\boldsymbol{\lambda}',\boldsymbol{\theta}_K(\boldsymbol{\lambda}');\boldsymbol{z}) - \widehat{\nabla_{\boldsymbol{\lambda}}\boldsymbol{\theta}_K}(\boldsymbol{\lambda}')\nabla_{\boldsymbol{\theta}}\ell^{\mathrm{val}}(\boldsymbol{\lambda}',\boldsymbol{\theta}_K(\boldsymbol{\lambda}');\boldsymbol{z})\right\| \\
\leq{}& Q\|\boldsymbol{\lambda} - \boldsymbol{\lambda}'\| + Q\|\boldsymbol{\theta}_K(\boldsymbol{\lambda}) - \boldsymbol{\theta}_K(\boldsymbol{\lambda}')\| \\
&+ \left\|\widehat{\nabla_{\boldsymbol{\lambda}}\boldsymbol{\theta}_K}(\boldsymbol{\lambda})\right\|\left(Q\|\boldsymbol{\lambda} - \boldsymbol{\lambda}'\| + Q\|\boldsymbol{\theta}_K(\boldsymbol{\lambda}) - \boldsymbol{\theta}_K(\boldsymbol{\lambda}')\|\right) \\
&+ Q\left\|\widehat{\nabla_{\boldsymbol{\lambda}}\boldsymbol{\theta}_K}(\boldsymbol{\lambda}) - \widehat{\nabla_{\boldsymbol{\lambda}}\boldsymbol{\theta}_K}(\boldsymbol{\lambda}')\right\| \\
\leq{}& Q\|\boldsymbol{\lambda} - \boldsymbol{\lambda}'\| + QL^{\boldsymbol{\theta}_K}\|\boldsymbol{\lambda} - \boldsymbol{\lambda}'\| \\
&+ L^{\boldsymbol{\theta}_K}\left(Q\|\boldsymbol{\lambda} - \boldsymbol{\lambda}'\| + QL^{\boldsymbol{\theta}_K}\|\boldsymbol{\lambda} - \boldsymbol{\lambda}'\|\right) \\
&+ QL^{\widehat{\nabla_{\boldsymbol{\lambda}}\boldsymbol{\theta}_K}}\|\boldsymbol{\lambda} - \boldsymbol{\lambda}'\| \\
\lesssim{}& K(1 + \eta\gamma^{\mathrm{tr}})^{2K}\|\boldsymbol{\lambda} - \boldsymbol{\lambda}'\|.
\end{aligned}
$$

which implies that $\gamma \lesssim K(1 + \eta\gamma^{\mathrm{tr}})^{2K}$. $\qquad\square$

**Proposition C.5** (Expansion properties of IFT-based algorithms). *Suppose we solve Example 5.3 by IFT-based Algorithm 1 with constant inner step size $\eta$ where $1 - \eta\gamma_d \geq 0$ and outer step size $\alpha_t$. Then (1) the outer update rules $G_{\boldsymbol{z}_i,\alpha_t}$ and $G_{\tilde{\boldsymbol{z}}_i,\alpha_t}$ are $2\alpha_t L'$- divergent along $\boldsymbol{v}_1$, and (2) the composite validation loss $\mathcal{L}(\cdot;\boldsymbol{z})$ is $\gamma'$-expansive along $\boldsymbol{v}_1$ for all $\boldsymbol{z} \in S^{\mathrm{val}}$, where*

$$
L' \gtrsim (1 + \eta\gamma^{\mathrm{tr}})^K, \gamma' \gtrsim (1 + \eta\gamma^{\mathrm{tr}})^{2K}.
$$

*Proof.* According to Lemma C.3, in the case of Example 5.3, the hypergradient calculated with the IFT-based algorithm is the same as the UD-based algorithm, which implies they achieve the same parameter divergence in this example. Therefore, we have the same result for $L'$ and $\gamma'$ as in Proposition 5.4 that $L' =\gtrsim (1 + \eta\gamma^{\mathrm{tr}})^K$ and $\gamma' \gtrsim (1 + \eta\gamma^{\mathrm{tr}})^{2K}$, which complete the proof. $\qquad\square$

### C.1 Stability bounds of IFT-based HO algorithm

Combining the lower bound in Theorem 5.1 with the upper bound in Equation (6), we instantly have

$$
\sum_{t=1}^{T}\prod_{s=t+1}^{T}\left(1 + \alpha_s(1 - 1/m)\gamma'\right)\frac{2\alpha_t L'}{m} \leq \epsilon_{\mathrm{arg}} \leq \sum_{t=1}^{T}\prod_{s=t+1}^{T}\left(1 + \alpha_s(1 - 1/m)\gamma\right)\frac{2\alpha_t L}{m}. \tag{13}
$$

**Theorem C.6** (Uniform argument stability of IFT-based algorithm, proof in Appendix C). *Solving Example 5.3 by IFT-based Algorithm 1 with constant inner step size $\eta$ where $1 - \eta\gamma_d \geq 0$ and decreasing outer step sizes $\alpha_t = c/t$ with $c$ as a positive constant $\dfrac{T^{\ln\left(1+(1-\frac{1}{m})c\gamma'\right)}}{m} \lesssim \epsilon_{\mathrm{arg}} \lesssim \dfrac{T^{(1-\frac{1}{m})c\gamma}}{m}$, where $\gamma \lesssim K(1 + \eta\gamma^{\mathrm{tr}})^{2K}, \gamma' \gtrsim (1 + \eta\gamma^{\mathrm{tr}})^{2K}$ as in Proposition C.5.*

The upper bound is not limited to Example 5.3, but holds in more general case with the same mild assumption in [17] (see Assumption B.1). Notably, in contrast to the outcomes observed with the UD-based algorithm, the upper bound incorporates an extra factor of $K$, leading to a larger upper bound for the IFT-based algorithm and a misalignment between the lower and upper bounds.

We can further establish a similar guarantee for the uniform stability $\epsilon_{\mathrm{stab}}$ detailed in Theorem C.7.

*Proof.* Based on properties in Propositions C.4 and C.5 the same proof as Theorem 5.5, we get the result. $\square$

**Theorem C.7** (Uniform stability of IFT-based algorithm). *Following the same condition as in Theorem C.6, and additionally, if the initial points $\boldsymbol{\theta}_0 = \boldsymbol{0}, \boldsymbol{\lambda}_0 = \boldsymbol{0}$, and $\boldsymbol{v}_1 \perp \boldsymbol{x}_j^{\mathrm{val}}$ for any $j \in [m]\backslash i$ and $\boldsymbol{v}_1 \perp \boldsymbol{x}_j^{\mathrm{tr}}$ for any $j \in [n]$, then the order of uniform stability $\epsilon_{\mathrm{stab}}$ w.r.t. $T$ satisfies $\frac{T^{\ln\left(1+(1-\frac{1}{m})c\gamma'\right)}}{m} \lesssim \epsilon_{\mathrm{stab}} \lesssim \frac{T^{(1-\frac{1}{m})c\gamma}}{m}$, where $\gamma$ and $\gamma'$ are the same as in Proposition C.5.*

*Proof.* With the same proof as Theorem 5.6, we can get the result. $\square$

## D   Deferred results of UD-based algorithm on (strongly) convex inner loss

Recalling that in Example 5.3, $\ell^{\mathrm{tr}}(\boldsymbol{\lambda}, \boldsymbol{\theta}; \boldsymbol{z}) = \frac{1}{2}\boldsymbol{\theta}^\top \boldsymbol{A}\boldsymbol{\theta} + \boldsymbol{\lambda}^\top \boldsymbol{\theta} - y\boldsymbol{x}^\top \boldsymbol{\theta}$, where the smallest eigenvalue of $\boldsymbol{A}$ is $\gamma_1$. Therefore, when $\gamma_1 \geq 0$ ($\gamma_1 > 0$), $\ell^{\mathrm{tr}}(\boldsymbol{\lambda}, \boldsymbol{\theta}; \boldsymbol{z})$ is convex (strongly convex) w.r.t. $\boldsymbol{\theta}$ for all $\boldsymbol{z} \in \mathcal{Z}$. Utilizing the case for $\gamma_1 \geq 0$ ($\gamma_1 > 0$) in Proposition B.5 and the same proof as in Theorem B.6 and Theorem 5.6, we can get the stability results for the convex (strongly convex) case in this section.

**Theorem D.1** (Uniform argument stability of UD-based algorithms for (strongly) convex $\ell^{\mathrm{tr}}$). *Solving Example 5.3 by UD-based Algorithm 1 with constant inner step size $\eta$ where $1 - \eta\gamma_d \geq 0$ and decreasing outer step sizes $\alpha_t = c/t$ with $c$ as a positive constant has uniform argument stability that*

$$\frac{T^{\ln\left(1+(1-\frac{1}{m})c\gamma'\right)}}{m} \lesssim \epsilon_{\mathrm{arg}} \lesssim \frac{T^{(1-\frac{1}{m})c\gamma}}{m},$$

*where $\gamma = \gamma' \asymp K$ when $\gamma_1 = 0$ and $\gamma = \gamma' \asymp 1$ when $\gamma_1 > 0$ as in Proposition B.5.*

*Proof.* Please refer to the proof of Proposition B.5 and Theorem B.6 where we generalize the results in the main paper for the (strongly) convex case. $\square$

**Theorem D.2** (Uniform stability of UD-based algorithms for (strongly) convex $\ell^{\mathrm{tr}}$). *Following the same condition as in Theorem D.1, and additionally, if the initial points $\boldsymbol{\theta}_0 = \boldsymbol{0}, \boldsymbol{\lambda}_0 = \boldsymbol{0}$, and $\boldsymbol{v}_1 \perp \boldsymbol{x}_j^{\mathrm{val}}$ for any $j \in [m]\backslash i$ and $\boldsymbol{v}_1 \perp \boldsymbol{x}_j^{\mathrm{tr}}$ for any $j \in [n]$, then Algorithm 1 has uniform stability that*

$$\frac{T^{\ln\left(1+(1-\frac{1}{m})c\gamma'\right)}}{m} \lesssim \epsilon_{\mathrm{stab}} \lesssim \frac{T^{(1-\frac{1}{m})c\gamma}}{m},$$

*where $\gamma = \gamma' \asymp K$ when $\gamma_1 = 0$ and $\gamma = \gamma' \asymp 1$ when $\gamma_1 > 0$ as in Proposition B.5.*

*Proof.* Please refer to the proof of Theorem 5.6. $\square$

## E   Deferred results of single-level SGD

As discussed in Section 4, deriving a stability lower bound entails constructing an example with maximum instability, and we need to study two aspects of the constructed example: (1) properties of the (compound) loss, and (2) stability behavior of the outer SGD corresponding to these properties. For (2), the outer level of gradient-based bilevel HO algorithms and the single-level SGD have equivalent formulation observing corresponding relations between $\boldsymbol{\lambda} \leftrightarrow \boldsymbol{w}$, $\mathcal{L} \leftrightarrow \ell$, $S^{\mathrm{val}} \leftrightarrow S$ and stability definitions Definition 3.1 $\leftrightarrow$ Eq. (15). As a result, given the smoothness constants $\gamma$ for $\mathcal{L}$ and $\ell$, the stability upper bounds under the bounded loss condition for the bilevel ($\epsilon_{\mathrm{stab}} \lesssim T^{\frac{(1-1/m)\gamma c}{(1-1/m)\gamma c+1}}/m$ in [17]) and single-level ($\epsilon_{\mathrm{stab}} \lesssim T^{\frac{\gamma c}{\gamma c+1}}/m$ in [19]) algorithms have similar results. Given those properties, their stability lower bounds can be analyzed in a general framework: construct a well-designed example, examine its key properties, and derive the stability lower bound in response to these properties.

Our proposed lower-bounded expansion properties in Section 4 and provable stability lower bound given these properties in Theorem 5.1 are generally applicable for both bilevel and single-level

analysis. Building upon these tools, we also establish stability lower bounds for single-level SGD in addition to our main results regarding bilevel algorithms. Notably, while the technique of stability analysis for the outer level of bilevel problems can be adapted to single-level ones, **the stability behavior of bilevel and single-level problems are not directly comparable**.

In this Section, we introduce basic concepts corresponding to stability analysis of single-level SGD in Appendix E.1 introduced by [19]. Based on this, Appendix E.2 leverages the lower-bounded expansion properties established in Section 4 to provide a stability lower bound for single-level SGD, which is tighter than the existing result in [34, Theorem 4]. An upper bound is established in Appendix E.3 for a fair comparison between the lower and upper bounds without the bounded loss condition. Detailed comparison with existing works is provided in Table 2.

---

**Algorithm 2** Single-level SGD

---

1: **Input:** Initialization $\boldsymbol{w}_0$; dataset $S$; step size scheme $\alpha$
2: **Output:** The parameter $\boldsymbol{w}_T$
3: **for** $t = 1$ **to** $T$ **do**
4:     uniformly sampling $i_t$ from $[m]$
5:     $\boldsymbol{g} \leftarrow \nabla\ell(\boldsymbol{w}_{t-1}; \boldsymbol{z}_{i_t})$
6:     $\boldsymbol{w}_t \leftarrow \boldsymbol{w}_{t-1} - \alpha_t \boldsymbol{g}$
7: **end for**
8: **return** $\boldsymbol{w}_T$

---

### E.1 Problem formulation for the stability analysis of single-level SGD

Suppose we are interested in the distribution $\mathcal{D}$ on data space $\mathcal{Z}$, from which we obtain a sample $S = \{\boldsymbol{z}_i\}_{i=1}^m \overset{\text{i.i.d.}}{\sim} \mathcal{D}^m$. Suppose $\boldsymbol{w}$ is the parameter to optimize in space $\Omega$, and its loss on an example $\boldsymbol{z}$ is $\ell(\boldsymbol{w}; \boldsymbol{z})$. The single-level SGD is shown in Algorithm 2. Following [19], the *generalization error* of single-level SGD is defined as

$$\epsilon_{\text{gen}} := \mathbb{E}_{\mathcal{A},S}\left[\mathbb{E}_{\boldsymbol{z}\sim\mathcal{D}}[\ell(\mathcal{A}(S); \boldsymbol{z})] - \frac{1}{m}\sum_{i=1}^m \ell(\mathcal{A}(S); \boldsymbol{z}_i)\right], \tag{14}$$

and we say a single-level stochastic algorithm $\mathcal{A}$ is $\epsilon_{\text{arg}}$-*uniformly argument stable* if,

$$\epsilon_{\text{arg}} = \sup_{S\simeq\tilde{S}\in\mathcal{Z}^m} \mathbb{E}_{\mathcal{A}}[\|\mathcal{A}(S) - \mathcal{A}(\tilde{S})\|]. \tag{15}$$

Based on these definitions, [19] has shown that stability guarantees generalization in single-level problems: if a stochastic algorithm $\mathcal{A}$ is $\epsilon_{\text{arg}}$-uniformly argument stable and the loss function $\ell(\boldsymbol{w}; \boldsymbol{z})$ is $L$-Lipschitz on $\Omega$ for all $\boldsymbol{z} \in \mathcal{Z}$, then we have

$$\epsilon_{\text{gen}} \leq \epsilon_{\text{stab}} \leq L\epsilon_{\text{arg}}. \tag{16}$$

### E.2 Proof of uniform stability lower bound

We first present a single-level example following [34].

**Example E.1.** Suppose $\Omega = \{\boldsymbol{w} : \|\boldsymbol{w}\| \leq W\}$ where $W > 0$, and $\mathcal{Z} = \mathcal{X} \times \mathcal{Y}$ where $\mathcal{X} = \{\boldsymbol{x} : \|\boldsymbol{x}\| \leq 1\}$ and $\mathcal{Y} = [-1, 1]$. Assume the loss function is $\ell(\boldsymbol{w}; \boldsymbol{z}) = \frac{1}{2}\boldsymbol{w}^\top \boldsymbol{A}\boldsymbol{w} - y\boldsymbol{x}^\top \boldsymbol{w}$, where $\boldsymbol{A} \in \mathbb{R}^{d\times d}$ is a symmetric matrix. Denote the eigenvalues of $\boldsymbol{A}$ as $\gamma_1 \leq \cdots \leq \gamma_d$, where $\gamma_1 < 0$ and $|\gamma_1| \geq |\gamma_d|$, and $\boldsymbol{v}_1$ as a unit eigenvector of $\boldsymbol{A}$ for $\gamma_1$. Additionally, suppose the twin datasets $S$ and $\tilde{S}$ are different at the $i$-th entry, where $\boldsymbol{z}_i = (\boldsymbol{x}_i, y_i) = (\boldsymbol{v}_1, 1), \tilde{\boldsymbol{z}}_i = (\tilde{\boldsymbol{x}}_i, \tilde{y}_i) = (-\boldsymbol{v}_1, 1)$.

**Proposition E.2** (Lipschitz continuity and smoothness coefficients). *In Example E.1, the loss function $\ell(\boldsymbol{w}; \boldsymbol{z})$ is $L$-Lipschitz continuous and $\gamma$-smooth on $\Omega$ for all $\boldsymbol{z} \in \mathcal{Z}$, where $L \leq |\gamma_1|W + 1$ and $\gamma = |\gamma_1|$.*

*Proof.* As $\ell$ is twice differentiable on $\Omega \times \mathcal{Z}$, we have

$$L = \sup_{\boldsymbol{z}\in\mathcal{Z}} \sup_{\boldsymbol{w}\in\Omega} \|\nabla\ell(\boldsymbol{w}; \boldsymbol{z})\| = \sup_{\boldsymbol{z}\in\mathcal{Z}} \sup_{\boldsymbol{w}\in\Omega} \|\boldsymbol{A}\boldsymbol{w} - y\boldsymbol{x}\| \leq |\gamma_1|W + 1,$$

$$\gamma = \sup_{\boldsymbol{z}\in\mathcal{Z}} \sup_{\boldsymbol{w}\in\Omega} \|\nabla^2\ell(\boldsymbol{w}; \boldsymbol{z})\| = \sup_{\boldsymbol{z}\in\mathcal{Z}} \sup_{\boldsymbol{w}\in\Omega} \|\boldsymbol{A}\| = |\gamma_1|.$$

$\square$

**Proposition E.3** (Divergent and expansive coefficients). *Suppose we solve Example E.1 by single-level SGD, then the gradient update rules $G_{\boldsymbol{z}_i, \alpha_t}$ and $G_{\tilde{\boldsymbol{z}}_i, \alpha_t}$ are $2\alpha_t L'$-divergent along $\boldsymbol{v}_1$ and $\ell(\boldsymbol{w}; \boldsymbol{z})$ is $\gamma'$-expansive along $\boldsymbol{v}_1$ on $\Omega$ for all $\boldsymbol{z} \in S$, where $L' = 1$ and $\gamma' = |\gamma_1|$.*

*Proof.* For all $\boldsymbol{w} \in \Omega$,

$$G_{\boldsymbol{z}_i, \alpha_t}(\boldsymbol{w}) - G_{\tilde{\boldsymbol{z}}_i, \alpha_t}(\boldsymbol{w}) = -\alpha_t\big(\nabla\ell(\boldsymbol{w}; \boldsymbol{z}_i) - \nabla\ell(\boldsymbol{w}; \tilde{\boldsymbol{z}}_i)\big) = -\alpha_t(y_i\boldsymbol{x}_i - \tilde{y}_i\tilde{\boldsymbol{x}}_i) = 2\alpha_t\boldsymbol{v}_1 \doteq \boldsymbol{v}_1.$$

Recalling Definition 4.1, this implies $G_{\boldsymbol{z}_i, \alpha_t}$ and $G_{\tilde{\boldsymbol{z}}_i, \alpha_t}$ are $2\alpha_t L'$-divergent where $L' = 1$. Additionally, for all $\boldsymbol{w}, \boldsymbol{w}' \in \Omega$ such that $\boldsymbol{w} - \boldsymbol{w}'$ collinear with $\boldsymbol{v}_1$ and any $\boldsymbol{z} \in \mathcal{Z}$, we have

$$\nabla\ell(\boldsymbol{w}; \boldsymbol{z}) - \nabla\ell(\boldsymbol{w}'; \boldsymbol{z}) = \boldsymbol{A}(\boldsymbol{w} - \boldsymbol{w}') = \gamma_1(\boldsymbol{w} - \boldsymbol{w}') = -|\gamma_1|(\boldsymbol{w} - \boldsymbol{w}').$$

Recalling Definition 4.3, this implies $\ell(\boldsymbol{w}; \boldsymbol{z})$ is $\gamma'$-expansive on $\Omega$ for all $\boldsymbol{z} \in \mathcal{Z}$ where $\gamma' = |\gamma_1|$. $\square$

Given Proposition E.3, we can directly leverage Theorem 5.1 to obtain a stability lower bound.

**Theorem E.4** (Lower bound of single-level SGD in recursion form). *In th case of Example E.1, running SGD for $T$ steps on a $\gamma$-smooth loss function has uniform argument stability with*

$$\epsilon_{\mathrm{arg}} \geq \sum_{t=1}^{T} \prod_{s=t+1}^{T} \big(1 + \alpha_s(1 - 1/m)\gamma\big)\frac{2\alpha_t}{m}.$$

*Proof.* Using Theorem 5.1 and Proposition E.3, we gets the result. $\square$

Following the proof of Theorem 5.5 we can deform the result in Theorem E.4 to display an explicit order under decreasing step sizes.

**Theorem E.5** (Lower bound of single-level SGD in deformed form). *In th case of Example E.1, running SGD for $T$ steps on a $\gamma$-smooth loss function with step sizes $\alpha_t = c/t$ has uniform argument stability with*

$$\epsilon_{\mathrm{arg}} \geq \frac{2c}{m\ln\big(1 + (1 - 1/m)c\gamma\big)}\left[\left(\frac{T+1}{2}\right)^{\ln\big(1 + (1-1/m)c\gamma\big)} - 1\right].$$

*Omitting constant factors that depends on $c$ and $\gamma$, we have $\epsilon_{\mathrm{arg}} \gtrsim \frac{T^{\ln\big(1 + (1-1/m)c\gamma\big)}}{m}$.*

*Proof.* The proof follows the scaling for the lower bound in Theorem 5.5. $\square$

*Remark.* Compared with Theorem 4 in [34] our result relaxes the condition and improves its order w.r.t. $m$. To see this, we first show that the step-size settings are equivalent. In particular, $\alpha_t = \frac{a}{0.99\gamma t}$ (Lemma 3, [34]) is equivalent to $\alpha_t = \frac{c}{t}$ (Theorem E.5, ours) with $c = \frac{a}{0.99\gamma}$. Based on this equivalence, we can rewrite the lower bound in [34] as $\epsilon_{\mathrm{arg}} \gtrsim \frac{T^{0.99c\gamma}}{m^{1+0.99c\gamma}}$ with assumptions $c = 1$, $0 < \gamma < \frac{0.1}{0.99}$ and $T > m$ (detailed in the proof of Theorem 4, [34]). In contrast, our lower bound of $\epsilon_{\mathrm{arg}} \gtrsim \frac{T^{\ln(1+(1-1/m)c\gamma)}}{m}$ in Theorem E.5 holds for any $c > 0$, $\gamma > 0$, $T \geq 1$, relaxing the conditions. Regarding the tightness of the lower bound, our result is sharper concerning $m$ given $\lim_{m\to\infty} \frac{\frac{T^{0.99c\gamma}}{m^{1+0.99c\gamma}}}{\frac{T^{\ln(1+(1-1/m)c\gamma)}}{m}} = 0$ for fixed $T$. In addition, concerning $T$, our result is comparable observing that the ratio of the powers on $T$ differ slightly, namely $0.96 \leq \frac{0.99c\gamma}{\ln(1+(1-1/m)c\gamma)} \leq 1.06$ for all $m \geq 100$, under the scope of application of their result (i.e., $c = 1$ and $0 < \gamma < \frac{0.1}{0.99}$). The superiority of our lower bound stems from a loose result in Lemma 3 in [34]. Denote $\Delta_t := \boldsymbol{w}_t - \tilde{\boldsymbol{w}}_t$. It states that $\mathbb{E}_{\mathcal{A}}[\|\Delta_T\|\,|\,\Delta_{t_0} \neq 0] \geq \frac{1}{2n}(\frac{T}{t_0})^{0.99c\gamma}$, while this can be improved into $\mathbb{E}_{\mathcal{A}}[\|\Delta_T\|\,|\,\Delta_{t_0} \neq 0] \geq \frac{1}{2m}(\frac{T}{t_0})^{0.99c\gamma} + (\frac{T+1}{t_0+1})^{0.99c\gamma}\Delta_{t_0}$, which will lead to a sharper lower bound.

### E.3 Proof of uniform stability upper bound

Denote $\delta_t := \|\boldsymbol{w}_t - \tilde{\boldsymbol{w}}_t\|$. The proof leverages an intermediate result of Theorem 3.12 in [19].

**Lemma E.6** ([19], Theorem 3.12). *Assume $\ell(\cdot; \boldsymbol{z})$ is L-Lipschitz and $\gamma$-smooth for all $\boldsymbol{z} \in \mathcal{Z}$. Running SGD with step sizes $\alpha_t$, for all $S \simeq \tilde{S} \in \mathcal{Z}^m$, we have the recurrence relation: $\forall 1 \le t \le T$,*

$$\mathbb{E}_{\mathcal{A}}[\delta_t] \le \left(1 - \frac{1}{m}\right)(1 + \alpha_t\gamma)\mathbb{E}_{\mathcal{A}}[\delta_{t-1}] + \frac{1}{m}\big(\mathbb{E}_{\mathcal{A}}[\delta_{t-1}] + 2\alpha_t L\big).$$

Unwinding the recursion we have the stability upper bound.

**Theorem E.7** (Upper bound of single-level SGD in recursion form). *Assume $\ell(\cdot; \boldsymbol{z})$ is L-Lipschitz and $\gamma$-smooth for all $\boldsymbol{z} \in \mathcal{Z}$. In th case of Example E.1, running SGD with step sizes $\alpha_t$ has uniform argument stability with*

$$\epsilon_{\text{arg}} \le \sum_{t=1}^{T} \prod_{s=t+1}^{T+1} \big(1 + (1 - 1/m)\alpha_s\gamma\big)\frac{2\alpha_t(\gamma W + 1)}{m}.$$

*Proof.* As defined, $\epsilon_{\text{arg}} = \sup_{S \simeq \tilde{S} \in \mathcal{Z}^m} \mathbb{E}_{\mathcal{A}}[\delta_T]$ and. Unwinding the recursion in Lemma E.6 and using the fact that $L \le \gamma W + 1$ in Proposition E.2, we get the result. $\qquad\square$

Here we set an additional $\alpha_{T+1} = 0$ for the expression neatness. Recalling Theorem E.4, the upper and lower bound are in exactly the same formulation with only difference in by a constant (i.e., $\gamma W + 1$), which means the lower and upper bound tightly match w.r.t. the key factors $T$ and $m$. Considering the case of constant step size, we get $\epsilon_{\text{arg}} \asymp \frac{\left(1+\left(1-1/m\right)\alpha\gamma\right)^T}{m}$, showing an exploding rate w.r.t. $T$. When adopting linearly decreasing step sizes $\alpha_t \le c/t$, the upper bound can also be deformed to reveal an explicit order w.r.t. key factors.

**Theorem E.8** (Upper bound of single-level SGD in defromed form). *Assume $\ell(\cdot; \boldsymbol{z})$ is L-Lipschitz and $\gamma$-smooth for all $\boldsymbol{z} \in \mathcal{Z}$. Running SGD for $T$ steps with step sizes $\alpha_t \le c/t$ has uniform argument stability of*

$$\epsilon_{\text{arg}} \le \frac{2(\gamma W + 1)}{(m-1)\gamma}\left[\big(1 + (1 - 1/m)\gamma c\big)T^{(1-1/m)c\gamma} - 1\right].$$

*Omitting constant factors that depends on c, $\gamma$ and W, we have $\epsilon_{\text{arg}} \lesssim \frac{T^{(1-1/m)c\gamma}}{m}$.*

*Proof.* The proof follows the scaling for the upper bound in Theorem 5.5. $\qquad\square$

Combining Theorem E.8 and Theorem E.5, we have $\frac{T^{\ln\left(1+(1-1/m)c\gamma\right)}}{m} \lesssim \epsilon_{\text{arg}} \lesssim \frac{T^{(1-1/m)c\gamma}}{m}$. Notably, the discrepancy between the lower and upper bounds is unavoidable, stemming from the scaling steps required to obtain an explicit order, and this gap becomes small when we have large $m$ and small $c\gamma$. Concerning the lower and upper bounds in recursion form in Theorem E.5 and Theorem E.7, our results are tightly matched.

*Remark.* Notice that Theorem 3.8 in [19] presents an upper bound of $\epsilon_{\text{arg}} \lesssim \frac{T^{\frac{\gamma c}{\gamma c+1}}}{m}$, which is tighter than our result of $\epsilon_{\text{arg}} \lesssim \frac{T^{(1-1/m)\gamma c}}{m}$ but with additional bounded loss assumption that $\ell(\boldsymbol{w}; \boldsymbol{z}) \in [0, 1]$. Both results are based on the recurrence relation in Lemma E.6. They derive the upper bound with a hitting time $t_0$ and bound the loss divergence after $t_0$ with the bounding loss constant (i.e., 1) and thus get a tighter upper bound. However, to derive lower bounds, we need to explicitly calculate the divergence between parameters and corresponding loss values, which will inevitably reveal all the terms in the recursion. In this case, the bounded loss assumption is not applicable and thus we present an upper bound without this condition as a fair and clear comparison with the lower bound.

We acknowledge that the bounded loss assumption is commonly adopted for upper-bound analysis in theoretical works. Despite [19], several following works also adopt this technique. [34] derive the upper bound of $\epsilon_{\text{arg}} \lesssim \frac{T^{\gamma c}}{m^{\gamma c+1}}$ in the nonconvex case with a similar approach by bounding the loss after hitting time $t_0$, with an additional setting for $t_0 = n$. However, there appears to be a misuse of Lemma 4 in their proof of Theorem 5, which leads to their result being tighter compared to [19] in

the case of $T^{\frac{c\gamma}{c\gamma+1}} \leq m$. Specifically, in the proof of Theorem 5, they decompose $\mathbb{E}_{\mathcal{A}}[\|\Delta_T]$ into two terms that $\mathbb{E}_{\mathcal{A}}[\|\Delta_T] \leq \mathbb{E}_{\mathcal{A}}[\|\Delta_T\|\|\Delta_n = 0]\mathbb{P}[\Delta_n = 0] + \mathbb{E}_{\mathcal{A}}[\|\Delta_T\|\|\Delta_n \neq 0]\mathbb{P}[\Delta_n \neq 0]$ to bound these two terms separately. For the second term, the union bound is used to get $\mathbb{E}_{\mathcal{A}}[\|\Delta_T\|\|\Delta_n \neq 0]\mathbb{P}[\Delta_n \neq 0] \leq \frac{1}{n}\sum_{t=1}^{n}\mathbb{E}_{\mathcal{A}}[\|\Delta_T\|\|H = t]$, where $H = t$ denotes that $t$ is the first time SGD pick the different entry in the twin datasets. $\mathbb{E}_{\mathcal{A}}[\|\Delta_T\|\|H = t]$ is further bounded using Lemma 4 that $\mathbb{E}_{\mathcal{A}}[\|\Delta_T\|\|\Delta_t = 0] \leq (\frac{T}{t})^a \frac{2L}{n}$ to get $\mathbb{E}_{\mathcal{A}}[\|\Delta_T\|\|H = t] \leq (\frac{T}{t})^a \frac{2L}{n}$. While this appears to be a misuse as $H = t$ can only imply $\Delta_{t-1} = 0$ and whether $\Delta_t = 0$ remains uncertain, where Lemma 4 is not applicable. Another work [49] use a large constant to bound the loss divergence from the start of the evolution of the parameter divergence, which leads to an upper bound of $\epsilon_{\mathrm{arg}} \lesssim \frac{T}{m}$ even with constant step sizes. A detailed comparison of existing results is listed in Table 2.

Table 2: A detailed comparison of existing results on uniform stability of single-level SGD. We unify the notations that the loss function is $\gamma$-smooth, the dataset is of size $m$ and SGD picks the different entry for the first time at $t_0$.

| Settings | Step size | Constant $\alpha_t = \alpha$ | Decreasing $\alpha_t \leq c/t$ | |
|---|---|---|---|---|
| | Range of iterations with bounding loss | $1 \leq t \leq T$ | $t_0 \leq t \leq T$ | - |
| Results | Upper bound | $\frac{T}{m}$ [49] | $\frac{T^{\frac{\gamma c}{\gamma c+1}}}{m}$ [19] | $\frac{T^{(1-1/m)\gamma c}}{m}$ (Ours) |
| | Lower bound | - | - | $\frac{T^{\ln(1+(1-1/m)c\gamma)}}{m}$ (Ours) $\frac{T^{0.99c\gamma}}{m^{1+0.99c\gamma}}$ [34] |

## F   Details of simulations

The implementing code is provided in the supplementary material. All simulations can be conducted on the CPU of a laptop.

### F.1   Hyperparameter distance and stability bounds

To examine the tightness and validity of the upper and lower bounds presented in Theorem 5.5, we implement UD-based Algorithm 1 on Example 5.3 with linearly decreasing step sizes and compare the practical output hyperparameter distances with our theoretical bounds under a range of outer iterations $T$.

Specifically, we set the loss functions and the twin validation sets as in Example 5.3 with $\mathbf{A} = \begin{bmatrix} -1 & 0 \\ 0 & 1 \end{bmatrix}$ and $\mathbf{v}_1 = \begin{bmatrix} 1 \\ 0 \end{bmatrix}$. The optimization is implemented with fixed $\gamma^{\mathrm{tr}} = 1$, $K = 100$, $m = 100$, $n = 100$, $\eta = 0.01$, and $c = 0.01$.

The comparison is shown in Fig. 2. We plot the output hyperparameter distances with increasing $T$ from 1000 to 5000 on the horizontal axis and the deformed lower bounds and upper bounds with corresponding $T$ on the vertical axis. The dashed lines are linear fittings of the hyperparameter distances and the upper/lower bounds, to examine the linear trends of their relative magnitude.

### F.2   Recursive stability bounds and deformed stability bounds

Here we implement additional simulations to examine the tightness between the recursive upper/lower bounds and deformed upper/lower bounds presented in Eq. (7) and Appendix B.6. Specifically, the recursive upper bound is calculated by $\sum_{t=1}^{T}\prod_{s=t+1}^{T}(1 + \alpha_s(1 - 1/m)\gamma)2\alpha_t L'/m$ and the recursive lower bound is calculated by $\sum_{t=1}^{T}\prod_{s=t+1}^{T}(1 + \alpha_s(1 - 1/m)\gamma)2\alpha_t L/m$ in Eq. (7). The deformed upper bound is calculated by $T^{(1-\frac{1}{m})c\gamma}/m$ and the deformed lower bound is calculated by $T^{\ln(1+(1-\frac{1}{m})c\gamma')}/m$. As for the coefficients: we set $\gamma^{\mathrm{tr}} = 1$ in Example 5.3. $L'$ is calculated with $[(1 + \eta\gamma^{\mathrm{tr}})^K - 1]/\gamma^{\mathrm{tr}}$ as in Eq. (10) and $L$ is calculated with $L = 0.1 + 1.1L'$ as they are of the same

order of magnitude. $\gamma = \gamma'$ is calculated with $\eta \sum_{k=0}^{K-1}(1 + \eta\gamma^{\mathrm{tr}})^k \left(2 + \eta\gamma^{\mathrm{tr}} \sum_{k=0}^{K-1}(1 + \eta\gamma^{\mathrm{tr}})^k\right)$ as in Eq. (11).

During the optimization, we fix $\eta = 0.01$, $n = 100$ and $c = 0.01$. For the simulation regarding $T$, we set $K = 100$ and $m = 100$ and plot the results of the recursive upper bounds for $T$ from 1000 to 5000 in the horizontal axis with the corresponding other three bounds in the vertical axis, shown in Fig.4. For the simulation regarding $K$, we set $T = 1000$ and $m = 100$ and plot the results of the recursive upper bounds for $K$ from 25 to 200 in the horizontal axis with the corresponding other three bounds in the vertical axis, shown in Fig.5. For the simulation regarding $m$, we set $T = 1000$ and $K = 100$ and plot the results of the recursive upper bounds for $m$ from 100 to 2000 in the horizontal axis with the corresponding other three bounds in the vertical axis, shown in Fig.6.

All curves exhibit linear trends, indicating these bounds are in the same order w.r.t. $T$, $K$, and $m$.

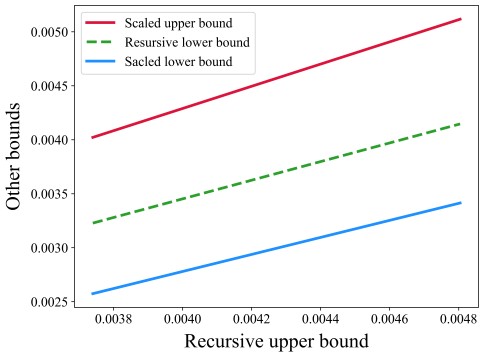

Figure 4: The relations of bounds for $T$ from 1000 to 5000.

Figure 5: The relations of bounds for $K$ from 25 to 200.

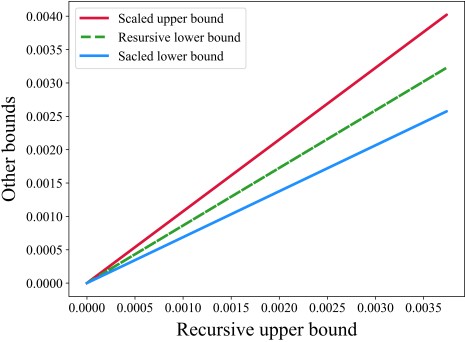

Figure 6: The relations of bounds for $K$ from 100 to 2000.

## G    Additional discussions

### G.1    Additional discussion for the definition of uniform stability on validation

In HO, the model is typically evaluated during the validation phase and is expected to generalize well on unseen test data based on its validation performance. Therefore, we focus on the impact of perturbations in the validation set in our current definition of generalization error and uniform argument stability. However, in the context of meta-learning where both datasets play crucial roles [50], considering the perturbations in the training set may provide additional insights for generalization analysis, which might be an interesting topic for future work.

## G.2 Additional discussion for expansiveness and existing concepts

We first clarify that the convex loss function corresponds to Definition 4.3 for the case when $\mu \leq 0$. When the loss function is convex, we have $\langle \nabla \ell(\boldsymbol{w}) - \nabla \ell(\boldsymbol{w}'), \boldsymbol{w} - \boldsymbol{w}' \rangle \geq 0$. According to Definition 4.3, if the loss is additionally $\mu$-expansive, there exist $\mu_{\boldsymbol{w}, \boldsymbol{w}'} \geq \mu$ such that

$$\langle \nabla \ell(\boldsymbol{w}) - \nabla \ell(\boldsymbol{w}'), \boldsymbol{w} - \boldsymbol{w}' \rangle = \langle -\mu_{\boldsymbol{w}, \boldsymbol{w}'}(\boldsymbol{w} - \boldsymbol{w}\prime), \boldsymbol{w} - \boldsymbol{w}' \rangle$$
$$= -\mu_{\boldsymbol{w}, \boldsymbol{w}'} \|\boldsymbol{w} - \boldsymbol{w}'\|^2$$
$$\geq 0,$$

thus we have $\mu \leq \mu_{\boldsymbol{w}, \boldsymbol{w}'} \leq 0$.

When $\mu > 0$, $\mu$-strongly concavity requires $\langle \nabla \ell(\boldsymbol{w}) - \nabla \ell(\boldsymbol{w}'), \boldsymbol{w} - \boldsymbol{w}' \rangle \leq -\mu \|\boldsymbol{w} - \boldsymbol{w}'\|^2$ for all $\boldsymbol{w}, \boldsymbol{w}' \in \Omega$, and Definition 4.3 restricts that for all $\boldsymbol{w}, \boldsymbol{w}' \in \Omega$ that $\boldsymbol{w} - \boldsymbol{w}'$ parallel with $\boldsymbol{v}$, there exists $\mu_{\boldsymbol{w}, \boldsymbol{w}'} \geq \mu$ such that

$$\langle \nabla \ell(\boldsymbol{w}) - \nabla \ell(\boldsymbol{w}'), \boldsymbol{w} - \boldsymbol{w}' \rangle = \langle -\mu_{\boldsymbol{w}, \boldsymbol{w}'}(\boldsymbol{w} - \boldsymbol{w}'), \boldsymbol{w} - \boldsymbol{w}' \rangle$$
$$= -\mu_{\boldsymbol{w}, \boldsymbol{w}'} \|\boldsymbol{w} - \boldsymbol{w}'\|^2$$
$$\leq -\mu \|\boldsymbol{w} - \boldsymbol{w}'\|^2.$$

Therefore, $\mu$-expansiveness along $\boldsymbol{v}$ implies concavity only for $\boldsymbol{w} - \boldsymbol{w}'$ parallel with $\boldsymbol{v}$.

On the other hand, if we have $\langle \nabla \ell(\boldsymbol{w}) - \nabla \ell(\boldsymbol{w}'), \boldsymbol{w} - \boldsymbol{w}' \rangle \leq -\mu \|\boldsymbol{w} - \boldsymbol{w}'\|^2$ for $\boldsymbol{w} - \boldsymbol{w}'$ parallel with $\boldsymbol{v}$, then

$$\langle \nabla \ell(\boldsymbol{w}) - \nabla \ell(\boldsymbol{w}'), \boldsymbol{w} - \boldsymbol{w}' \rangle \leq \|\nabla \ell(\boldsymbol{w}) - \nabla \ell(\boldsymbol{w}')\| \|\boldsymbol{w} - \boldsymbol{w}'\|$$
$$\leq -\mu \|\boldsymbol{w} - \boldsymbol{w}'\|^2,$$

and thus $\|\nabla \ell(\boldsymbol{w}) - \nabla \ell(\boldsymbol{w}')\| \leq -\mu \|\boldsymbol{w} - \boldsymbol{w}'\|$. Therefore, compared with strongly concavity on a single direction, $\mu$-expansiveness has an additional restriction for the colinearity of $\nabla \ell(\boldsymbol{w}) - \nabla \ell(\boldsymbol{w}')$ and $-(\boldsymbol{w} - \boldsymbol{w}')$.

Additionally, for the one-dimensional case, the condition $w - w'$ parallel with some scalar $v$ is equivalent to $\forall w, w' \in \Omega$. Therefore, $\mu$-expansiveness along $v$ implies $\mu$-strongly concavity. Conversely, $\mu$-strongly concavity implies that there exists a $\mu_{\boldsymbol{w}, \boldsymbol{w}'} \geq \mu$ such that

$$(\nabla \ell(\boldsymbol{w}) - \nabla \ell(\boldsymbol{w}'))(\boldsymbol{w} - \boldsymbol{w}') = -\mu_{\boldsymbol{w}, \boldsymbol{w}'}(\boldsymbol{w} - \boldsymbol{w}')^2$$
$$\leq -\mu (\boldsymbol{w} - \boldsymbol{w}')^2,$$

thus $\mu$-strongly concavity conversely also implies $\mu$-expansiveness along $v$. Therefore, these concepts are equivalent in the one-dimensional case.

## G.3 Technical challenges of stability lower bound analysis for bilevel algorithms

The nested optimization in bilevel algorithms poses challenges to the stability analysis as the instability and simplicity of the constructed example are both crucial for deriving a tight lower bound. Specifically, to examine the alignment of the lower and upper bounds, we need to precisely calculate the smooth coefficient $\gamma$ and expansive coefficient $\mu$ of the compound validation loss for the constructed example. While the implicit and intricate formulation of $\nabla \mathcal{L}(\boldsymbol{\lambda})$ in bilevel optimization makes $\gamma$ and $\mu$ difficult to obtain. In the following, we take ridge regression as an example to illustrate how the bilevel structure hinders the stability analysis.

**Example G.1** (Regularization coefficient in ridge regression). The validation loss and training loss are given by $\ell^{\mathrm{val}}(\lambda, \boldsymbol{\theta}) = \frac{1}{2}(y - \boldsymbol{\theta}^T \boldsymbol{x})^2$, $\ell^{\mathrm{tr}}(\lambda, \boldsymbol{\theta}) = \frac{1}{2}(y - \boldsymbol{\theta}^\top \boldsymbol{x})^2 + \frac{\lambda}{2}\boldsymbol{\theta}^\top \boldsymbol{\theta}$. Solving it with UD-based Algorithm 1, we have the inner output as $\boldsymbol{\theta}_K(\lambda) = \prod_{k=1}^K (\boldsymbol{I} - \eta\lambda\boldsymbol{I} - \eta\boldsymbol{x}_{j_k}\boldsymbol{x}_{j_k}^\top)\boldsymbol{\theta}_0 + \sum_{i=1}^K \prod_{l=k+1}^K (\boldsymbol{I} - \eta\lambda\boldsymbol{I} - \eta\boldsymbol{x}_{j_l}\boldsymbol{x}_{j_l}^\top)\eta y_{j_k}\boldsymbol{x}_{j_k}$ and a far more complex inner Jacobian $\nabla_\lambda \boldsymbol{\theta}_K(\lambda)$, resulting in a unmeasurable hypergradient $\nabla \mathcal{L}(\lambda) = \nabla_\lambda \boldsymbol{\theta}_K(\lambda)(y - \boldsymbol{\theta}_K(\lambda)^\top \boldsymbol{x})(-\boldsymbol{x})$.

These complexities obstacle us to precisely examine the divergence dynamics at each step. Therefore, we introduce expansion properties in Section 4 and the general lower bounded guarantees in Theorems 5.1 and 5.2 to jointly contribute to the careful construction of Example 5.3. As a result, Example 5.3 exhibits the maximum instability while having a relatively simple outer gradient update feasible for lower bound analysis, which will lead to tight stability lower bounds presented in Section 5.4.

# H Potential extension of our framework

## H.1 Extension on average stability lower bounds

Considering the similarities between average stability [32, Definition 2] and uniform stability, our techniques may be adapted to the data-dependent setting for average stability with some modifications. In this section, we present a preliminary proof sketch for establishing the stability lower bound of single-level SGD based on the variant Example E.1.

To account for the randomness in the sampled datasets, we first define some notations. Let $S_i$ denote a copy of $S$ with the $i$-th element replaced by $z_i'$, $G_{S,t}/G_{S_i,t}$ and $w_{S,t}/w_{S_i,t}$ denote the SGD update rules and the updated parameters optimized on $S$ and $S_i$ at the step $t$. With a slight adjustment Section 4.2 and a similar proof as in the current paper, Theorem 5.1 can be modified as: Suppose there exists a nonzero vector $v(S, S_i)$ along which $G_{S,t}$ and $G_{S_i,t}$ are $2\alpha_t L'(S, S_i)$-divergent and $\mathcal{L}(\cdot; z)$ is $\gamma_t'(S, S_i)$-expansive for all $w_{S,t} - w_{S_i,t}$ that parallel with $v(S, S_i)$. Then we have

$$\mathbb{E}_{\mathcal{A}}[\delta_t(S, S_i)] \geq [1 + \alpha_t(1 - 1/m)\gamma_t'(S, S_i)]\mathbb{E}_{\mathcal{A}}[\delta_{t-1}(S, S_i)] + 2\alpha_t L'(S, S_i)/m.$$

This recursion closely corresponds with the recursive upper bound in [32, Eq.(19)]:

$$\mathbb{E}_{\mathcal{A}}[\delta_t(S, S_i)] \leq [1 + \alpha_t(1 - 1/m)\psi_t(S, S_i)]\mathbb{E}_{\mathcal{A}}[\delta_{t-1}(S, S_i)] + 2\alpha_t L(S, S_i)/m,$$

and the matching of $\gamma_t'$ & $\psi_t$, $L'$ & $L$ will further guide the design of the constructed example.

Therefore, our analysis framework can be adapted for the average stability with suitable modifications. Specifically for average stability, a core challenge is calculating the expectation over $S$ and $S_i$ for the above recursive formula, which is beyond the scope of our paper. In the following, we provide a proof sketch for establishing the average argument stability lower bound to clarify a possible approach to extend our framework.

Assume that the data follows a distribution $\mathcal{D}$ that $p(z) = \begin{cases} 0.5 & \text{if } z = (v_1, 1), \\ 0.5 & \text{if } z = (-v_1, 1). \end{cases}$ $S \overset{\text{i.i.d.}}{\sim} \mathcal{D}^m$

and $z_i' \sim \mathcal{D}$ are independent of each other. The validation loss and training loss follow Example E.1. Under these assumptions, we derive that $L'(S, S_i) = \|y_i x_i - y_i' x_i'\|/2$ and $\mu_t(S, S_i) = 0$ for $z_i = z_i'$, $\mu_t(S, S_i) = |\gamma_1|$ for $z_i \neq z_i'$. This leads to the average argument stability lower bound:

$$\epsilon_{\text{arg}} \geq \sum_{t=1}^{T} \prod_{s=t+1}^{T} (1 + \alpha_s(1 - 1/m|\gamma_1|))\alpha_t/m.$$

## H.2 Extension on generalization lower bounds

In this section, we discuss a possible approach to extend our framework on the analysis of generalization lower bounds. We first present a lemma clarifying the fundamental equivalence between generalization and stability. Then a lower bound on the expected hyperparameter divergence is established by slightly modifying Example 5.3 with additional design on the data distribution, which will imply the generalization lower bound under certain conditions.

**Lemma H.1.** *Let $S^{\text{val}} = (z_1^{\text{val}}, \ldots, z_m^{\text{val}})$ and $S^{\text{val}'} = (z_1^{\text{val}'}, \ldots, z_m^{\text{val}'})$ be two independent samples drawn i.i.d. from $\mathcal{D}^{\text{val}}$. Let $\tilde{S}_i^{\text{val}} = (z_1^{\text{val}}, \ldots, z_i^{\text{val}'}, \ldots, z_m^{\text{val}})$ denote the twin validation set of $S^{\text{val}}$ differing in the $i$-th example. Consider $\epsilon_{\text{gen}}$ as the generalization error defined in Eq. (4). Then, we have*

$$\epsilon_{\text{gen}} = \frac{1}{m} \sum_{i=1}^{m} \mathbb{E}_{\mathcal{A}, S^{\text{tr}}, S^{\text{val}}, z_i^{\text{val}'}}[\mathcal{L}(\mathcal{A}(S^{\text{val}}, S^{\text{tr}}); z_i^{\text{val}'}) - \mathcal{L}(\mathcal{A}(\tilde{S}_i^{\text{val}}, S^{\text{tr}}); z_i^{\text{val}'})].$$

*Proof.* According to the definition of generalization error in Eq. (4) and the linearity of expectation,

$$\epsilon_{\text{gen}} = \underbrace{\mathbb{E}_{\mathcal{A}, S^{\text{val}}, S^{\text{tr}}} \left[\mathbb{E}_{z \sim \mathcal{D}^{\text{test}}}[\mathcal{L}(\mathcal{A}(S^{\text{val}}, S^{\text{tr}}); z)]\right]}_{(a)} - \underbrace{\frac{1}{m} \sum_{i=1}^{m} \mathbb{E}_{\mathcal{A}, S^{\text{val}}, S^{\text{tr}}} \left[\mathcal{L}(\mathcal{A}(S^{\text{val}}, S^{\text{tr}}); z_i^{\text{val}})\right]}_{(b)}.$$

As it is assumed $\mathcal{D}^{\text{val}} = \mathcal{D}^{\text{test}}$ and $S^{\text{val}'}$ is i.i.d. sampled from $\mathcal{D}^{\text{val}}$ independent from $S^{\text{val}}$, term (a) can be rewritten as

$$(\text{a}) = \mathbb{E}_{\mathcal{A}, S^{\text{val}}, S^{\text{tr}}} \left[ \frac{1}{m} \sum_{i=1}^{m} \mathbb{E}_{\boldsymbol{z}_i^{\text{val}'}} [\mathcal{L}(\mathcal{A}(S^{\text{val}}, S^{\text{tr}}); \boldsymbol{z}_i^{\text{val}'})] \right].$$

Under the expectation, $S^{\text{val}}$ and $\tilde{S}_i^{\text{val}}$ is exchangeable, then term (b) can be rewritten as

$$(\text{b}) = \frac{1}{m} \sum_{i=1}^{m} \mathbb{E}_{\mathcal{A}, \tilde{S}_i^{\text{val}}, S^{\text{tr}}} \left[ \mathcal{L}(\mathcal{A}(\tilde{S}_i^{\text{val}}, S^{\text{tr}}); \boldsymbol{z}_i^{\text{val}'}) \right] = \frac{1}{m} \sum_{i=1}^{m} \mathbb{E}_{\mathcal{A}, S^{\text{val}}, \boldsymbol{z}_i^{\text{val}'}, S^{\text{tr}}} \left[ \mathcal{L}(\mathcal{A}(\tilde{S}_i^{\text{val}}, S^{\text{tr}}); \boldsymbol{z}_i^{\text{val}'}) \right].$$

Combining (a) and (b) leads to the equation in the theorem. $\qquad\square$

Lemma H.1 shows a fundamental relation between generalization and stability: The generalization error equals the expected loss divergence when replacing a single example in the validation set. Stability-based generalization analysis typically takes the supremum on $S^{\text{tr}}, S^{\text{val}}, \boldsymbol{z}_i^{\text{val}'}$ to obtain a distribution-agnostic upper bound of generalization error as

$$\epsilon_{\text{gen}} \leq \epsilon_{\text{stab}} := \sup_{S^{\text{tr}}, S^{\text{val}}, \boldsymbol{z}_i^{\text{val}'}} \mathbb{E}_{\mathcal{A}} [\mathcal{L}(\mathcal{A}(S^{\text{val}}, S^{\text{tr}}); \boldsymbol{z}_i^{\text{val}'}) - \mathcal{L}(\mathcal{A}(\tilde{S}_i^{\text{val}}, S^{\text{tr}}); \boldsymbol{z}_i^{\text{val}'})],$$

where $\epsilon_{\text{stab}}$ is commonly upper bounded assuming $L$-Lipschitz of the loss as

$$\epsilon_{\text{stab}} \leq L\epsilon_{\text{arg}} := L \sup_{S^{\text{tr}}, S^{\text{val}} \simeq \tilde{S}^{\text{val}}} \mathbb{E}_{\mathcal{A}} [\|\mathcal{A}(S^{\text{val}}, S^{\text{tr}}) - \mathcal{A}(\tilde{S}^{\text{val}}, S^{\text{tr}})\|].$$

This paper attempts to derive lower bounds of $\epsilon_{\text{arg}}$ which will not directly imply the generalization lower bounds because $\epsilon_{\text{arg}}$ is fundamentally a distribution-agnostic upper bound for the generalization error. In order to obtain a generalization lower bound, a promising way is to extend Example 5.3 with additional assumption on the validation distribution rather than directly specifying $S^{\text{val}}$ and $\tilde{S}^{\text{val}}$.

We present a primary result below for the lower bound of the expected hyperparameter divergence, which indicates a way to extend our methods on the analysis of generalization lower bounds.

**Example H.2.** We introduce an HO problem as follows. Let the validation loss and the training loss be:

$$\ell^{\text{val}}(\boldsymbol{\lambda}, \boldsymbol{\theta}; \boldsymbol{z}) = \ell^{\text{tr}}(\boldsymbol{\lambda}, \boldsymbol{\theta}; \boldsymbol{z}) = \frac{1}{2}\boldsymbol{\theta}^\top \boldsymbol{A} \boldsymbol{\theta} + \boldsymbol{\lambda}^\top \boldsymbol{\theta} - y\boldsymbol{x}^\top \boldsymbol{\theta},$$

where $\boldsymbol{A} \in \mathbb{R}^{d \times d}$ is symmetric. Denote the eigenvalues of $\boldsymbol{A}$ as $\gamma_1 \leq \cdots \leq \gamma_d$. Let $\gamma_1 < 0, \gamma_d \leq 0$, and $\boldsymbol{v}_1$ be a unit eigenvector for $\gamma_1$. Suppose the validation distribution follows:

$$p(\boldsymbol{z}) = \begin{cases} 0.5 & \text{if } \boldsymbol{z} = (\boldsymbol{v}_1, 1), \\ 0.5 & \text{if } \boldsymbol{z} = (-\boldsymbol{v}_1, 1). \end{cases}$$

**Theorem H.3.** *Suppose we solve Example H.2 by UD-based Algorithm 1 with constant inner step size $\eta$ where $1 - \eta\gamma_d \geq 0$ and outer step size $\alpha_t$. Denote that the expected hyperparameter divergence as $\epsilon_{\text{gen,arg}} := \frac{1}{m} \sum_{i=1}^{m} \mathbb{E}_{\mathcal{A}, S^{\text{tr}}, S^{\text{val}}, \boldsymbol{z}_i^{\text{val}'}} [\|\mathcal{A}(S^{\text{val}}, S^{\text{tr}}) - \mathcal{A}(\tilde{S}_i^{\text{val}}, S^{\text{tr}})\|]$. Then, we have*

$$\epsilon_{\text{gen,arg}} \geq \sum_{t=1}^{T} \prod_{s=t+1}^{T} \left(1 + \alpha_s(1 - 1/m)\gamma\right) \frac{\alpha_t L'}{m},$$

*where*

$$L \asymp L' \asymp (1 + \eta\gamma^{\text{tr}})^K, \gamma = \gamma' \asymp (1 + \eta\gamma^{\text{tr}})^{2K}.$$

*Proof.* We first decompose $\epsilon_{\mathrm{gen,arg}}$ conditioned on the difference of $\boldsymbol{z}_i^{\mathrm{val}}$ and $\boldsymbol{z}_i^{\mathrm{val}'}$ as

$$
\begin{aligned}
&\epsilon_{\mathrm{gen,arg}}\\
&=\frac{1}{m}\sum_{i=1}^{m}\mathbb{P}[\boldsymbol{z}_i^{\mathrm{val}}=\boldsymbol{z}_i^{\mathrm{val}'}]\mathbb{E}_{\mathcal{A},S^{\mathrm{tr}},S^{\mathrm{val}},\boldsymbol{z}_i^{\mathrm{val}'}}[\|\mathcal{A}(S^{\mathrm{val}},S^{\mathrm{tr}});-\mathcal{A}(\tilde{S}_i^{\mathrm{val}},S^{\mathrm{tr}})\||\boldsymbol{z}_i^{\mathrm{val}}=\boldsymbol{z}_i^{\mathrm{val}'}]\\
&\quad+\frac{1}{m}\sum_{i=1}^{m}\mathbb{P}[\boldsymbol{z}_i^{\mathrm{val}}\neq\boldsymbol{z}_i^{\mathrm{val}'}]\mathbb{E}_{\mathcal{A},S^{\mathrm{tr}},S^{\mathrm{val}},\boldsymbol{z}_i^{\mathrm{val}'}}[\|\mathcal{A}(S^{\mathrm{val}},S^{\mathrm{tr}})-\mathcal{A}(\tilde{S}_i^{\mathrm{val}},S^{\mathrm{tr}})\||\boldsymbol{z}_i^{\mathrm{val}}\neq\boldsymbol{z}_i^{\mathrm{val}'}]\\
&=\frac{1}{m}\sum_{i=1}^{m}\mathbb{P}[\boldsymbol{z}_i^{\mathrm{val}}\neq\boldsymbol{z}_i^{\mathrm{val}'}]\mathbb{E}_{\mathcal{A},S^{\mathrm{tr}},S^{\mathrm{val}},\boldsymbol{z}_i^{\mathrm{val}'}}[\|\mathcal{A}(S^{\mathrm{val}},S^{\mathrm{tr}})-\mathcal{A}(\tilde{S}_i^{\mathrm{val}},S^{\mathrm{tr}})\||\boldsymbol{z}_i^{\mathrm{val}}\neq\boldsymbol{z}_i^{\mathrm{val}'}]\\
&=\frac{1}{2m}\sum_{i=1}^{m}\mathbb{E}_{\mathcal{A},S^{\mathrm{tr}},S^{\mathrm{val}},\boldsymbol{z}_i^{\mathrm{val}'}}[\|\mathcal{A}(S^{\mathrm{val}},S^{\mathrm{tr}})-\mathcal{A}(\tilde{S}_i^{\mathrm{val}},S^{\mathrm{tr}})\||\boldsymbol{z}_i^{\mathrm{val}}\neq\boldsymbol{z}_i^{\mathrm{val}'}].
\end{aligned}
$$

The last equation holds for that as $\boldsymbol{z}_i^{\mathrm{val}}$ and $\boldsymbol{z}_i^{\mathrm{val}'}$ are sampled from $\mathcal{D}^{\mathrm{val}}$ specified in Example H.2 independently, which leads to $\mathbb{P}[\boldsymbol{z}_i^{\mathrm{val}}\neq\boldsymbol{z}_i^{\mathrm{val}'}]=1/2$.

According to Proposition 5.4 and Theorem 5.1, for any $S^{\mathrm{tr}}$, $i\in[m]$, and $S^{\mathrm{val}}\simeq\tilde{S}_i^{\mathrm{val}}$ where $\boldsymbol{z}_i^{\mathrm{val}}\neq\boldsymbol{z}_i^{\mathrm{val}'}\in\{(-\boldsymbol{v}_1,1),(\boldsymbol{v}_1,1)\}$, we have

$$
\|\mathcal{A}(S^{\mathrm{val}},S^{\mathrm{tr}})-\mathcal{A}(\tilde{S}_i^{\mathrm{val}},S^{\mathrm{tr}})\|\geq\sum_{t=1}^{T}\prod_{s=t+1}^{T}\left(1+\alpha_s(1-1/m)\gamma\right)\frac{2\alpha_t L'}{m}.
$$

Therefore, it holds that

$$
\epsilon_{\mathrm{gen,arg}}\geq\frac{1}{2m}\sum_{i=1}^{m}\sum_{t=1}^{T}\prod_{s=t+1}^{T}\left(1+\alpha_s(1-1/m)\gamma\right)\frac{2\alpha_t L'}{m}=\sum_{t=1}^{T}\prod_{s=t+1}^{T}\left(1+\alpha_s(1-1/m)\gamma\right)\frac{\alpha_t L'}{m}.
$$

$\square$

This result sheds light on the analysis of the generalization lower bound by establishing the lower bound on the expected hyperparameter divergence since it will induce a generalization lower bound if there exists a positive real constant $\underline{L}$ such that $|\epsilon_{\mathrm{gen}}|\geq\underline{L}\epsilon_{\mathrm{gen,arg}}$. One possible situation is that the designed compound validation loss satisfies for all $\boldsymbol{z}\in\mathcal{Z}$, $|\mathcal{L}(\boldsymbol{\lambda};\boldsymbol{z})-\mathcal{L}(\boldsymbol{\lambda}';\boldsymbol{z})|\geq\underline{L}\|\boldsymbol{\lambda}-\boldsymbol{\lambda}'\|$. As the generalization lower bound is beyond the main scope of this paper, further design and derivation may be left for future research.

