# OpenReview forum: "Lower Bounds of Uniform Stability in Gradient-Based Bilevel Algorithms for Hyperparameter Optimization"
_NeurIPS.cc/2024/Conference — NeurIPS 2024 poster_

### Official Review · Reviewer_3C12 · 2024-06-18

**Soundness:** 2
**Presentation:** 2
**Contribution:** 2
**Rating:** 5
**Confidence:** 3

**Summary:**

The paper provides new lower bounds for the uniform stability of UD-based and IFT-based algorithms in hyperparameter optimization.

**Strengths:**

1. The general setting is described clearly.
2. The uniform stability lower bounds presented in the paper are novel and clear.

**Weaknesses:**

1. In my view, the technique employed by the authors appears to be relatively straightforward. For the lower bound, they demonstrate that gradient algorithms may be costly when applied to a non-convex quadratic loss function. While this is not particularly surprising, it can result in a significant distance between the iterates of the two algorithms.

2. The lower bound presented by the authors does not necessarily imply a generalization lower bound. This might be acceptable, given that there are existing works in the literature that show a lower bound for uniform stability (e.g., [30]). However, in relation to the previous point, I find the result to be less insightful.

**Questions:**

1. Are the bounds the authors present in theorems 5.6,5.7 tight?

When denoting $ x = \gamma \left(1 - \frac{1}{m}\right) $ for $ \gamma = \gamma' $, the upper bound scales like $T^{cx} $ and the lower bound scales like $ T^{\log(1+cx)} $. The authors state that when $ c $ is a small constant, ``the explicit lower bound closely aligns with the explicit upper bound'' (line 310). However, since  c appears in the exponent of T, for a multiplicative constant difference between the bounds,  c needs to depend on T.

2. Does the technique of the paper can lead also to lower bounds for the average stability?

**Limitations:**

Yes.

---

> ### Author Rebuttal · Authors · 2024-08-07
>
> # Response to Reviewer 3C12
> We thank Reviewer 3C12 for the thoughtful and constructive comments.
> ## W1: Technical challenges of establishing the lower bounds
> We agree that the instability of gradient-based algorithms on non-convex loss functions is intuitively expected, but rigorously establishing (nearly) tight lower bounds is a non-trivial task, especially for bilevel algorithms.
>
> For bilevel algorithms, the stability bounds must consider key factors from both the outer and inner optimization processes (m, T, K), which poses challenges to achieving the tightness of the lower bounds. Moreover, the update of the outer parameters with the hypergradient (see Eq. (1), (2), (3), and the discussion for Example 5.4) becomes particularly complex due to the involvement of inner optimization, necessitating a careful design of the loss function beyond merely being non-convex.
>
> ## W2: The implication of a stability lower bound.
> Indeed, as discussed in Section 6, a stability lower bound does not directly imply a generalization lower bound. However, our work focuses on characterizing the limit of the uniform stability on the generalization analysis, and we have presented (nearly) tight lower bounds for the UD-based algorithms which is currently missing in the literature and technically challenging as discussed in the response for W1. Moreover, our methods may inspire the establishment of the generalization lower bound as discussed in the following response for Q2 regarding single-level SGD.
>
> ## Q1: The tightness of the lower bounds in Theorems 5.6 and 5.7.
> **The recursive lower and upper bounds in Eq.(7) are tight as claimed in Line 300,** while the adjusted bounds in Theorems 5.6 and 5.7 are not tight due to **unavoidable scaling steps needed to reveal an explicit order w.r.t. $T$**.
>
> For a constant step size, i.e., $\alpha_t = c$, Eq.(7) directly provides tight bounds that $[(1 + c(1 - 1/m)\gamma)^T - 1]{2cL^\prime}/{m} \leq \epsilon_{\text{arg}} \leq [(1 + c(1 - 1/m)\gamma)^T - 1]{2cL}/{m}.$ However, when decreasing step sizes are adopted, additional scaling steps based on Eq.(7) must be applied to reveal an explicit order of the bounds w.r.t. $T$, resulting in the discrepancy between the bounds in Theorems 5.6 and 5.7.
>
> Regarding our statement of the close alignment, considering Thms 5.6 and 5.7, the discrepancy between $T^{cx}$ and $T^{\log(1+cx)}$ is $T^{cx}/T^{\log(1+cx)}=T^{cx-\log(1+cx)}$ and thus depend on $T$. Following your suggestion, we will revise relevant statements to clarify that: the discrepancy between the bounds is $T^{cx}/T^{\log(1+cx)}=T^{cx - \log(1+cx)}$.
>
> ## Q2: Extend the technique to the average stability.
> **Considering the similarities between average and uniform stability, we believe our techniques can be adapted to the data-dependent setting for average stability with some modifications, and we established a preliminary lower bound for single-level SGD using our Example E.1, which is comparable to the corresponding upper bound in [Theorem 4, 31].**
>
> To account for the randomness in the sampled datasets, we first refine some notations. Let $S^{(i)}$ denote a copy of $S$ with the $i$-th element replaced by $z_i'$, $G_{S,t}$/$G_{S^{(i)},t}$ and $w_{S,t}$/$w_{S^{(i)},t}$ denote the SGD update rules and the updated parameters on $S$ and $S^{(i)}$ at the step $t$. With a similar proof as in the current paper and a slight adjustment on the Definition 4.3, Theorem 5.1 in our paper can be modified as: Suppose there exists a nonzero vector $v(S, S^{(i)})$ along which $G_{S,t}$ and $G_{S^{(i)},t}$ are $2\alpha_tL'(S, S^{(i)})$-divergent and $L(\cdot;z)$ is $\gamma_t'(S, S^{(i)})$-expansive for all $w_{S,t} - w_{S^{(i)},t}$ that parallel with $v(S, S^{(i)})$. Then we have $$E_A[\delta_t(S, S^{(i)})]
> \geq [1+\alpha_t(1-1/m)]\gamma_t'(S, S^{(i)})E_A[\delta_{t-1}(S, S^{(i)})]+2\alpha_tL'(S, S^{(i)})/m.$$
> This recursion closely corresponds with the one in [Eq.(19), 31]: $$E_A[\delta_t(S, S^{(i)})]
> \leq [1+\alpha_t(1-1/m)]\psi_t'(S, S^{(i)})E_A[\delta_{t-1}(S, S^{(i)})]+2\alpha_tL(S, S^{(i)})/m,$$ and the matching of $\gamma_t'\\&\psi_t, L'\\&L$ will further guide the design of the constructed example.
>
> Therefore, our analysis framework can be adapted for the average stability with suitable modifications. However, specifically for average stability, a core challenge is calculating the expectation over $S$ and $S^{(i)}$ for the above recursive formula, which is beyond our ability to fully resolve within the short rebuttal period. Nevertheless, we can present a preliminary lower bound using Example E.1 with a simple assumption on the data distribution.
>
> Assume that the data follows a distribution $D$ that $p(z)=0.5$ for $z=(v_1, 1)$ and $z=(-v_1, 1)$ respectively, $S \overset{i.i.d.}{\sim} D^m$, $z_i' \sim D$, and other conditions follow Example E.1 in our paper. Under these assumptions, we derive $L'(S, S^{(i)})=\Vert y_ix_i-y_i'x_i'\Vert /2$ and $\mu_t(S, S^{(i)})=0$ for $z_i=z_i'$, $\mu_t(S, S^{(i)})=\vert\gamma_1\vert$ for $z_i\neq z_i'$. This leads to the average stability lower bound: $\epsilon\geq\sum_{t=1}^T\prod_{s=t+1}^T(1+\alpha_s(1-1/m\vert\gamma_1\vert))\alpha_t/m$.
>
> Moreover, for our example, Eq. (2) in Theorem 4 in [31] takes $\gamma=\vert \gamma_1\vert$. When adopting decreasing step sizes and removing the bounded loss assumption to get an adjusted upper bound as discussed in Lines 822-829, we further have: $$T^{\ln(1+(1-1/m))c\vert\gamma_1\vert}/m\lesssim\epsilon\lesssim T^{(1-1/m)c\vert\gamma_1\vert}/m.$$
>
> Furthermore, related to the above W2, this lower bound also applies to another definition of average stability [Definition 5, 21] which is shown to be equivalent to the expected generalization error [Lemma 11, 21]. Therefore this lower bound is also a generalization lower bound.
>
> We will add the above discussion and detailed proofs in the revised version. Thanks again for your valuable comments. We welcome any additional feedback to address any further questions.

---

> > ### Comment · Reviewer_3C12 · 2024-08-08
> >
> > Thank you for the detailed response!
> >
> > Although the reviewers addressed each of my questions individually, I want to highlight my main concern, which relates to the combined implications of all their answers.
> >
> > The technique used by the authors to bound uniform stability involves lower bounding the expansion of the algorithm. As the authors mentioned in their response, this approach has an inherent limitation due to the "unavoidable scaling steps needed to reveal an explicit order with respect to $T$." Consequently, unless the step sizes of the algorithm are $o\left(\frac{1}{t}\right)$ (which I believe are not typically used in practice), the lower bounds provided by the authors are not tight and exhibit a polynomial difference (with a constant degree) with respect to $T$ when compared to the best-known stability upper bounds. Together with the fact that a lower bound for uniform stability does not necessarily imply a lower bound for generalization, this weakens the significance of the results.

---

> > > ### Author Response · Authors · 2024-08-08
> > > **Potential misunderstanding regarding the tightness of our results**
> > >
> > > Thank you for your response.
> > >
> > > There might be a potential misunderstanding that: the recursive lower and upper bounds in Eq.(7) are indeed tight, and **for a constant step size** where $\alpha_t = c$ for all $t$ (which is commonly adopted in practice), Eq.(7) directly provides **tight** bounds: $\left[(1 + c(1 - 1/m)\gamma)^T - 1\right] \frac{2cL'}{m} \leq \epsilon_{\text{arg}} \leq \left[(1 + c(1 - 1/m)\gamma)^T - 1\right] \frac{2cL}{m}$. Therefore, we argue for the practical implications of our results.
> > >
> > > In the submission, to align with the setting where $\alpha_t \leq \frac{c}{t}$ (which is widely discussed in theoretical works such as [17] and [18]), we present lower bounds in Theorems 5.6 and 5.7.
> > >
> > > We will add the discussion in the final version.

---

> > > ### Author Response · Authors · 2024-08-09
> > > **Expanding our methods to generalization lower bounds**
> > >
> > > We want to clarify that **our methods, specifically the constructed examples and the analysis framework, can inspire the establishment of generalization lower bounds.** As a result, **we present a preliminary generalization lower bound** for bilevel (UD and IFT-based) HO algorithms using Example 5.3 and for single-level SGD using Example E.1 with additional assumptions on the data distribution.
> > >
> > > In particular for bilevel HO algorithms, further assume that the data follows a distribution $D$ where $p(z)=0.5$ for $z=(v_1, 1)$ and $(-v_1, 1)$ respectively, $S^{val}\overset{i.i.d.}{\sim} D^m$, ${z^{val}}'_ i \sim D$, and other conditions follow Example 5.3 in the paper. We derive $L'(S, S^{(i)})=\eta\Vert \sum_{k=1}^{K-1}(I-\eta A)^k(y_ix_i-y_i'x_i')\Vert /2$ and $\mu_t(S, S^{(i)})=0$ for $z_i=z_i'$, $\mu_t(S, S^{(i)})=\gamma'$ (as in Lines 586-588) for $z_i\neq z_i'$. This leads to a generalization lower bound: $$\epsilon_ {gen}\geq\sum_ {t=1}^T\prod_ {s=t+1}^T(1+\alpha_ s(1-1/m\gamma'))\alpha_tL'/m,$$ where $L' \asymp L \asymp (1+\eta\gamma^{tr})^{K}$ and $\gamma' = \gamma \asymp (1+\eta\gamma^{tr})^{2K}$. When adopting decreasing step sizes, we further have: $$\epsilon_{gen}\gtrsim T^{\ln(1+(1-1/m))c\gamma'}/m.$$ More detailed illustrations and results for single-level SGD have been presented in the above response for Q2 from Reviewer 3C12.
> > >
> > > We will add relevant discussions and detailed proofs in the final version.

---

> > > > ### Comment · Reviewer_3C12 · 2024-08-11
> > > >
> > > > Thank you for the clarifications. I have updated my score.

---

> > > > > ### Author Response · Authors · 2024-08-11
> > > > >
> > > > > Thanks for the helpful comments and for increasing the score. We are glad that we have addressed the reviewer's main concerns.

---

### Official Review · Reviewer_bA9a · 2024-07-11

**Soundness:** 4
**Presentation:** 4
**Contribution:** 4
**Rating:** 8
**Confidence:** 2

**Summary:**

The authors consider gradient-based algorithms for hyperparameter selection. They derive lower stability bounds for UD and IFT-based algorithms that solve this problem. To get this lower bound, the authors introduce the notion of lower-bounded exansion property of the update rule of the hyperparameters.

**Strengths:**

- **S1**: This paper is very well written and easy to follow.

- **S2**: The paper establishes lower bound on the stability of gradient-based methods for bilevel optimization. By doing so, they demonstrate the tighness of existing stability bounds for UD-based algorithms.

- **S3**: To my knowledge, the use of the lower-bounded expansion property to prove lower stability bounds is novel.

**Weaknesses:**

- **W1**: The most efficient gradient-based algorithms use warm-starting strategy, meaning that the inner and the inverse Hessian vector product solvers are not reinitialized at each iteration [1]. It is not clear if these algorithms are covered by the analysis.

**Questions:**

- **Q1**: Can the lower bound be extended to algorithms that use warm-starting strategy?

---

> ### Author Rebuttal · Authors · 2024-08-07
>
> # Response to Reviewer bA9a
> ## W&Q: Extension of the lower bound analysis to algorithms with warm-starting strategy.
>
> We thank Reviewer bA9a for the positive comments and thoughtful questions.
>
> We are uncertain about the precise meaning of the "warm-starting strategy" you mentioned, as we do not find related definitions in the reference [1] *Gradient-based optimization of hyperparameters*. Please correct us if there is any misunderstanding in our response.
>
> In our understanding, the "warm-starting strategy" may have two meanings. First, the inner parameter $\theta$ is not reinitialized at each outer step. Second, the inverse Hessian vector product solver used to calculate the hypergradient is not reinitialized at each outer step.
>
> For the inner warm starting, our lower bound analysis can be extended with necessary modifications. In Section 4, we define the expansion properties assuming that the update rule is a mapping from the parameter's current state to its next state, independent of its previous states. However, with the warm-starting strategy, the hyperparameter update depends on all its previous states. Therefore, our proposed definitions and corresponding analysis need to be adjusted to characterize this dependence. Specifically, Definition 4.2 can be modified as: An update rule $G$ is $\\{\rho_ s\\}_ {s=1}^S$-growing along $v$ if for all
> $\\{w_ s,w_ s'\\}_ {s=1}^S$ such that $w_ s-w_ s'$ parallel with $v$, $G(w_ S)-G(w_ S) \circeq w_ S-w_ S', \Vert G(w_ S)-G(w_ S)\Vert\geq \sum_ {s=1}^S\rho_ s \Vert w_ s-w_ s'\Vert.$ Consequently, the recursion in Theorem 5.1 will take the form: $E_A[\delta_t]\geq a_tE_A[\delta_{t-1}]+a_{t-1}E_A[\delta_{t-2}]+\dots+a_2E_A[\delta_{1}]+2\alpha_tL'/m$ in contrast of the current form: $E_A[\delta_t]\geq a_tE_A[\delta_{t-1}]+2\alpha_tL'/m$. The stability lower bound will be derived by unrolling this revised recursion.
>
> Regarding the inverse Hessian-vector product solvers, we clarify that the algorithms presented in our paper (Algorithms 1, Eq.(2) and Eq.(3)) do not calculate the inver Hessian and thus do not use such numerical solver. While we recognize that for some IFT-based algorithms where the hypergradient is decomposed as $\nabla_ {\lambda}L = \nabla_ {\lambda}\ell^{val} -\underbrace{\nabla_ {\theta}\ell^{val}(\nabla^2_ {\theta\theta}\ell^{tr})^{-1}}_ {\text{inverse Hessian vector product}}\nabla^2_ {\theta\lambda}\ell^{tr}$, the reinitialization of the solver might be an important factor to consider. We suggest a similar modification as for the inner warm-starting strategy, involving additional dependency on parameters, to extend our methods to this case.
>
> Thanks again for your insightful questions. Warm starting is indeed a commonly used practice to enhance the efficiency of algorithms. We are not able to present a precise result on the lower bounds within this short rebuttal period, but we believe our work can inspire future attempts in these settings as described above. We will add relevant discussions in the revised version, and we believe this will improve the quality of our paper. We hope our response addresses your concerns and welcome any further feedback.

---

### Official Review · Reviewer_NBuR · 2024-07-23

**Soundness:** 3
**Presentation:** 3
**Contribution:** 2
**Rating:** 5
**Confidence:** 3

**Summary:**

This paper studies the generalization bound of the hyper-parameter optimization problem. It provides a uniform lower-bound for the validation argument stability, which proves that the upper-bound of the validation argument stability in existing works is tight.

**Strengths:**

The paper is well-written and easy to follow.

This paper studies the hyper-parameter optimization problem, which is of great importance to bilevel optimization community.

The paper' theoretical result on the lower-bound of validation argument stability is sound and novel. It provides good insight into the stability of the hyper-parameter optimization problem.

**Weaknesses:**

How does the current generalization bound depend on the number of data $n$ in training dataset? According to the current bound, it is beneficial to just divide more data into the validation set to increase $m$. It seems there is nothing that quantifies the negative impact of having a smaller training dataset. This could lead to unreasonable interpretation of the bound: suppose there is a dataset of size $N$ where we assign $m$ data into validation and $n=N-m$ into training dataset. According to the current bound, the generalization bound decreases in an order of $\mathcal{O}(1/m)$, which suggests one should just use a largest $m$ possible, which is counter intuitive.

**Questions:**

My question is described in the weakness section. It is my major question, and I will consider raising the score if it is well answered.

**Limitations:**

No potential social impact.

---

> ### Author Rebuttal · Authors · 2024-08-07
>
> # Response to Reviewer NBuR
> ## W&Q: Interpretation of the current generalization bound w.r.t. validation and training sample sizes.
>
> We thank Reviewer NBuR for the insightful question.
>
> Intuitively, the expected population risk can be divided into the generalization error and the empirical validation risk as $$E_ {S^{val},S^{tr}}[R(A(S^{val},S^{tr}))]=\underbrace{E_ {S^{val},S^{tr}}[R(A(S^{val},S^{tr}))-\hat{R}_ {S^{val}}(A(S^{val},S^{tr}))]}_ {\text{(I): generalization error (Eq.(4) in our paper)}}+\underbrace{E_ {S^{val},S^{tr}}[\hat{R}_ {S^{val}}(A(S^{val},S^{tr}))]}_ {\text{(II): empirical validation risk}}.$$  The first term is explicitly bounded w.r.t. the validation sample size $m$ in our paper ($\epsilon_{gen}\leq \epsilon_{stab}=\Theta(1/m)$, see Line 140 and Theorem 5.7.). The second term is implicitly bounded w.r.t. the training sample size $n$ since a larger training set generally leads to better validation performance. Therefore, **the assignment of validation and training sets is a trade-off between $\text{(I)}$ and $\text{(II)}$, which aligns with common discussions on generalization in validation, as seen in [Page 70, 4]  for cross-validation.**
>
> Specifically, consider assigning $m=aN$ examples to $S^{val}$ and $n=(1-a)N$ examples to $S^{tr}$ where $a\in(0,1)$. **On the one hand**, the generalization bound $\epsilon_{gen}\leq \epsilon_{stab}=\Theta(1/aN)$ (see Line 140 and Theorem 5.7) presented in our paper implies that $a$ needs to be sufficiently large for the term $\text{(I)}$ to be small. **On the other hand**, as a larger training set generally leads to better validation performance, $a$ also needs to be sufficiently small to get a low validation risk in the term $\text{(II)}$. Therefore, **the choice of $a$ represents a trade-off to achieve a favorable population risk**.
>
> Technically, the absence of $n$ in the generalization bound arises from the formulation and decomposition of the generalization error. As defined, $\epsilon_{gen}$ can be written as $E_{S^{tr}}[E_{S^{val}}[R(A(S^{val},S^{tr}))-\hat{R}_{S^{val}}(A(S^{val},S^{tr}))]]$. Given a fixed $S^{tr}$ and the independence of $S^{val}$ and $S^{tr}$, the inner term becomes a difference between the expected and empirical validation risk, which is typically bounded with convergence analysis (e.g, stability, uniform convergence, e.t.c.). The outer expectation on $S^{tr}$ is often scaled as a supremum over $S^{tr}$. These technical analyses lead to a generalization bound solely on $m$.
>
> We will add the above discussion in the revised version, and we believe this will improve our paper's clarity.

---

### Decision · Program_Chairs · 2024-09-25

**Decision:**

Accept (poster)

**Comment:**

The paper makes an interesting contribution to the lower bounds on generalization error in a bi-level ERM setup. The authors are highly encouraged to address ReviewerNBuR's question on "how the generalization bound depends on the training dataset size" with a concrete bound (as opposed to a qualitative explanation) in the revision.